# Theoretical Characterization of Neural Network Generalization with Group Imbalance

## Abstract

Group imbalance has been a known problem in empirical risk minimization (ERM), where the achieved high *average* accuracy is accompanied by low accuracy in a *minority* group. Despite algorithmic efforts to improve the minority group accuracy, a theoretical generalization analysis of ERM on individual groups remains elusive. By formulating the group imbalance problem with the Gaussian Mixture Model, this paper quantifies the impact of individual groups on the sample complexity, the convergence rate, and the average and group-level testing performance. Although our theoretical framework is centered on binary classification using a one-hidden-layer neural network, to the best of our knowledge, we provide the first theoretical analysis of the group-level generalization of ERM in addition to the commonly studied average generalization performance. Sample insights of our theoretical results include that when all group-level co-variance is in the medium regime and all mean are close to zero, the learning performance is most desirable in the sense of a small sample complexity, a fast training rate, and a high average and group-level testing accuracy. Moreover, we show that increasing the fraction of the minority group in the training data does not necessarily improve the generalization performance of the minority group. Our theoretical results are validated on both synthetic and empirical datasets such as CelebA and CIFAR-10 in image classification.

## 1 Introduction

Training neural networks with empirical risk minimization (ERM) is a common practice to reduce the average loss of a machine learning task evaluated on a dataset. However, recent findings (Blodgett et al., 2016; Tatman, 2017; Hashimoto et al., 2018; Buolamwini & Gebru, 2018; McCoy et al., 2019; Sagawa et al., 2020; Sagawa* et al., 2020; Mehrabi et al., 2021) have shown empirical evidence about a critical challenge of ERM, known as *group imbalance*, where a well-trained model that has high average accuracy may have significant errors on the minority group that infrequently appears in the data. Moreover, the group attributes that determine the majority and minority groups are usually hidden and unknown during training. The training set can be augmented by data augmentation methods (Shorten & Khoshgoftaar, 2019) with varying performance, such as cropping and rotation (Krizhevsky et al., 2012), noise injection (Moreno-Barea et al., 2018), and generative adversarial network (GAN)-based methods (Goodfellow et al., 2014; Bowles et al., 2018; Radford et al., 2016).

As ERM is a prominent method and enjoys great empirical success, it is important to characterize the impact of ERM on group imbalance theoretically. However, the technical difficulty of analyzing the nonconvex ERM problem of neural networks results from the concatenation of nonlinear functions across layers, and the existing generalization analyses of ERM often make overly simplistic assumptions and only focus on the average generalization performance. For example, the neural tangent kernel type of analysis (Arora et al., 2019; Allen-Zhu et al., 2019b;a; Cao & Gu, 2019; Chen et al., 2020; Du et al., 2019; Jacot et al., 2018; Zou et al., 2020; Zou & Gu, 2019) linearizes the neural network around the random initialization to remove the nonconvex interactions across layers. The generalization bounds are independent of the feature distribution and cannot be exploited to analyze the impact of individual groups. Li & Liang (2018) provides the sample complexity analysis when the data comes from the mixtures of well-separated distributions but still cannot characterize the learning performance of individual groups. Another line of works (Du et al., 2018a; Ghorbani et al., 2020; Goldt et al., 2020; Li & Liang, 2018; Mei et al., 2018; Mignacco et al., 2020; Yoshida & Okada, 2019) considers one-hidden-layer neural networks because the ERM problem is already

Figure 1: Group imbalance experiment. (a) Binary classification on CelebA dataset using Gaussian augmentation to control the minority group co-variance. (b) Test accuracy against the augmented noise level.

highly nonconvex, and the analytical complexity increases tremendously when the number of hidden layers increases. In these works, the input features are usually assumed to be i.i.d. samples drawn from the standard Gaussian distribution, and this data model cannot differentiate the majority and minority groups.

**Contribution**: To the best of our knowledge, *this paper provides the first theoretical characterization of both the average and group-level generalization of a one-hidden-layer neural network trained by ERM on data generated from a mixture of distributions.* This paper considers the binary classification problem with the cross entropy loss function, with training data generated by a ground-truth neural network with known architecture and unknown weights. The optimization problem is challenging due to a high non-convexity from the multi-neuron architecture and the non-linear sigmoid activation.

Assuming the features follow a Gaussian Mixture Model (GMM), where samples of each group are generated from a Gaussian distribution with an arbitrary mean vector and co-variance matrix, this paper quantifies the impact of individual groups on the sample complexity, the training convergence rate, and the average and group-level test error. The training algorithm is the gradient descent following a tensor initialization and converges linearly. Our key results include

(1) *Medium-range group-level co-variance enhances the learning performance*. When a group-level co-variance deviates from the medium regime, the learning performance degrades in terms of higher sample complexity, slower convergence in training, and worse average and group-level generalization performance. As shown in Figure 1(a), we introduce Gaussian augmentation to control the co-variance level of the minority group in the CelebA dataset (Liu et al., 2015). The learned model achieves the highest test accuracy when the co-variance is at the medium level, see Figure 1(b). Another implication is that the diverse performance of different data augmentation methods might partially result from the different group-level co-variance introduced by these methods. Furthermore, although our setup does not directly model the batch normalization approach (Ioffe & Szegedy, 2015; Bjorck et al., 2018; Chai et al., 2020; Santurkar et al., 2018) that modifies the mean and variance in each layer to achieve fast and stable convergence, our result provides a theoretical insight that co-variance indeed affects the learning performance.

(2) *Group-level mean shifts from zero hurt the learning performance*. When a group-level mean deviates from zero, the sample complexity increases, the algorithm converges slower, and both the average and group-level test error increases. Thus, the learning performance is improved if each distribution is zero-mean. This paper provides a similar theoretical insight to practical tricks such as whitening LeCun et al. (1998), subgroup shift (Koch et al., 2022; Ma et al., 2021), population shift (Biswas & Mukherjee, 2021; Giguere et al., 2022) and the pre-processing of making data zero-mean (Lecun et al., 1998), that data mean affects the learning performance.

(3) *Increasing the fraction of the minority group in the training data does not always improve its generalization performance*. The generalization performance is also affected by the mean and co-variance of individual groups. In fact, increasing the fraction of the minority group in the training data can have a completely opposite impact in different datasets.

## 2 BACKGROUND AND RELATED WORK

**Improving the minority-group performance with known group attributes**. With known group attributes, distributionally robust optimization (DRO) (Sagawa* et al., 2020) minimizes the worst-group training loss instead of solving ERM. DRO is more computationally expensive than ERM and does not always outperform ERM in the minority-group test error. Spurious correlations (Sagawa et al., 2020) can be viewed as one reason of group imbalance, where strong associations between labels and irrelevant features exist in training samples. Different from the approaches that address spurious correlations, such as down-sampling the majority (Japkowicz & Stephen, 2002; Haixiang et al., 2017; Buda et al., 2018), up-weight the minority group (Shimodaira, 2000; Byrd & Lipton, 2019), and removing spurious features (Garg et al., 2019; Elhabian et al., 2008; Zemel et al., 2013), this paper does not require the special model of spurious correlations and any group attribute information.

**Fairness in machine learning** has received a lot of interest recently (Barocas & Selbst, 2016), and a substantial body of work has been developed to enhance the fairness under various notions (Dwork et al., 2012; Feldman et al., 2015; Hardt et al., 2016; Kleinberg et al., 2017; Kearns et al., 2018; Chen et al., 2018; Makhlouf et al., 2021; Li et al., 2021). For example, DRO maximizes the welfare of the worst group, satisfying the fairness notion of (Rawls, 2001). Different from the majority of these works, this paper solves ERM directly without group attribute information. Moreover, this paper focuses on characterizing the generalization performance of ERM as a function of the input distribution but does not attempt to evaluate fairness across groups.

**Generalization performance with the standard Gaussian input for one-hidden-layer neural networks.** (Brutzkus & Globerson, 2017; Du et al., 2018b; Ge et al., 2018; Liang et al., 2018; Li & Yuan, 2017; Shamir, 2018; Safran & Shamir, 2018; Tian, 2017) consider infinite training samples. (Zhong et al., 2017b;a) characterize the sample complexity of fully connected neural networks with smooth activation functions. Zhang et al. (2019; 2020b) extend to the non-smooth ReLU activation for fully-connected and convolutional neural networks, respectively. Fu et al. (2020) analyzes the cross entropy loss function for binary classification problems. Zhang et al. (2020a) analyzes the generalizability of graph neural networks for both regression and binary classification problems.

**Theoretical characterization of learning performance from other input distributions for one-hidden-layer neural networks.** Yoshida & Okada (2019) analyzes the training loss with a single Gaussian with an arbitrary co-variance. Mignacco et al. (2020) quantifies the SGD evolution trained on the Gaussian mixture model. When the hidden layer only contains one neuron, Du et al. (2018a) analyzes rotationally invariant distributions. With an infinite number of neurons and an infinite input dimension, Mei et al. (2018) analyzes the generalization error based on the mean-field analysis for distributions like Gaussian Mixture with the same mean. Ghorbani et al. (2020) considers inputs with low-dimensional structures. No sample complexity is provided in all these works.

**Notations**: $\boldsymbol{Z}$ is a matrix, and $\boldsymbol{z}$ is a vector. $z_i$ denotes the $i$-th entry of $\boldsymbol{z}$, and $Z_{i,j}$ denotes the $(i,j)$-th entry of $\boldsymbol{Z}$. $[K]$ denotes the set including integers from 1 to $K$. $\boldsymbol{I}_d$ and $\boldsymbol{e}_i$ represent the identity matrix in $\mathbb{R}^{d \times d}$ and the $i$-th standard basis vector, respectively. $\delta_i(\boldsymbol{Z})$ denotes the $i$-th largest singular value of $\boldsymbol{Z}$. The matrix norm $\|\boldsymbol{Z}\| = \delta_1(\boldsymbol{Z})$. $f(x) = O(g(x))$ (or $\Omega(g(x))$, $\Theta(g(x))$) means that $f(x)$ increases at most, at least, or in the order of $g(x)$, respectively.

## 3 PROBLEM FORMULATION AND ALGORITHM

We consider the classification problem with an unbalanced dataset using fully connected neural networks over $n$ independent training examples $\{(\boldsymbol{x}_i, y_i)\}_{i=1}^N$ from a data distribution. The learning algorithm is to minimize the empirical risk function via gradient descent (GD). In what follows, we will present the data model and neural network model considered in this paper.

**Data Model**. Let $\boldsymbol{x} \in \mathbb{R}^d$ and $y \in \mathbb{R}$ denote the input feature and label, respectively. We consider an unbalanced dataset that consists of $L$ ($L \geq 2$) groups of data, where the feature $\boldsymbol{x}$ in the group $l$ ($l \in [L]$) is drawn from a multi-variate Gaussian distribution with mean $\boldsymbol{\mu}_l \in \mathbb{R}^d$, and covariance $\boldsymbol{\Sigma}_l \in \mathbb{R}^{d \times d}$. Specifically, $\boldsymbol{x}$ follows the Gaussian mixture model (GMM) (Pearson, 1894; Titterington et al., 1985; Hsu & Kakade, 2013; Vempala & Wang, 2004; Moitra & Valiant, 2010; Regev & Vijayaraghavan, 2017), denoted as $\boldsymbol{x} \sim \sum_{l=1}^L \lambda_l \mathcal{N}(\boldsymbol{\mu}_l, \boldsymbol{\Sigma}_l)$. $\lambda_l \in (0, 1)$ is the probability of sampling from distribution-$l$ and represents the expected fraction of group-$l$ data. $\sum_{l=1}^L \lambda_l = 1$.

Group $l$ is defined as a minority group if $\lambda_l$ is less than $1/L$. We use $\Psi = \{\lambda_l, \boldsymbol{\mu}_l, \boldsymbol{\Sigma}_l, \forall l\}$ to denote all parameters of the mixture model[1]. We consider binary classification with label $y$ generated by a ground-truth neural network with unknown weights $\boldsymbol{W}^* = [\boldsymbol{w}_1^*, ..., \boldsymbol{w}_K^*] \in \mathbb{R}^{d \times K}$ and sigmoid activation[2]. function $\phi(x) = \frac{1}{1+\exp(-x)}$, where

$$\mathbb{P}(y = 1 | \boldsymbol{x}) = H(\boldsymbol{W}^*, \boldsymbol{x}) := \frac{1}{K} \sum_{j=1}^{K} \phi(\boldsymbol{w}_j^{*\top} \boldsymbol{x}). \tag{1}$$

**Learning model**. Learning is performed over a neural network that has the same architecture as in (1), which is a one-hidden-layer fully connected neural network[3] with its weights denoted by $\boldsymbol{W} \in \mathbb{R}^{d \times K}$. Given $n$ training samples $\{\boldsymbol{x}_i, y_i\}_{i=1}^{n}$ where $\boldsymbol{x}_i$ follows the GMM model, and $y_i$ is from (1), we aim to find the model weights via minimizing the nonconvex empirical risk $f_n(\boldsymbol{W})$ as

$$\min_{\boldsymbol{W} \in \mathbb{R}^{d \times K}} f_n(\boldsymbol{W}) := \frac{1}{n} \sum_{i=1}^{n} \ell(\boldsymbol{W}; \boldsymbol{x}_i, y_i), \tag{2}$$

where $\ell(\boldsymbol{W}; \boldsymbol{x}_i, y_i)$ is the cross-entropy loss function, i.e.,

$$\ell(\boldsymbol{W}; \boldsymbol{x}_i, y_i) = -y_i \cdot \log(H(\boldsymbol{W}, \boldsymbol{x}_i)) - (1 - y_i) \cdot \log(1 - H(\boldsymbol{W}, \boldsymbol{x}_i)). \tag{3}$$

Note that for any permutation matrix $\boldsymbol{P}$, $\boldsymbol{W}\boldsymbol{P}$ corresponds permuting neurons of a network with weights $\boldsymbol{W}$. Therefore, $H(\boldsymbol{W}, \boldsymbol{x}) = H(\boldsymbol{W}\boldsymbol{P}, \boldsymbol{x})$, and $f_n(\boldsymbol{W}\boldsymbol{P}) = f_n(\boldsymbol{W})$. The estimation is considered successful if one finds any column permutation of $\boldsymbol{W}^*$.

The average generalization performance of a learned model $\boldsymbol{W}$ is evaluated by the average risk

$$\bar{f}(\boldsymbol{W}) = \mathbb{E}_{\boldsymbol{x} \sim \sum_{l=1}^{L} \lambda_l \mathcal{N}(\boldsymbol{\mu}_l, \boldsymbol{\Sigma}_l)} \ell(\boldsymbol{W}; \boldsymbol{x}_i, y_i), \tag{4}$$

and the generalization performance on group $l$ is evaluated by the group-$l$ risk

$$\bar{f}_l(\boldsymbol{W}) = \mathbb{E}_{\boldsymbol{x} \sim \mathcal{N}(\boldsymbol{\mu}_l, \boldsymbol{\Sigma}_l)} \ell(\boldsymbol{W}; \boldsymbol{x}_i, y_i). \tag{5}$$

**Training Algorithm**. Our algorithm starts from an initialization $\boldsymbol{W}_0 \in \mathbb{R}^{d \times K}$ computed based on the tensor initialization method (Subroutine 1 in Section B.1) and then updates the iterates $\boldsymbol{W}_t$ using gradient descent with the step size[4] $\eta_0$. The computational complexity of tensor initialization is $O(Knd)$. The per-iteration complexity of the gradient step is $O(Knd)$. We defer the details of Algorithm 1 in Section B of the supplementary material.

## 4 MAIN THEORETICAL RESULTS

We will formally present our main theory below, and the insights are summarized in Section 4.1. For the convenience of presentation, some quantities are defined here, and all of them can be viewed as constant. Define $\sigma_{\max} = \max_{l \in [L]} \{\|\boldsymbol{\Sigma}_l\|^{\frac{1}{2}}\}$, $\sigma_{\min} = \min_{l \in [L]} \{\|\boldsymbol{\Sigma}_l^{-1}\|^{-\frac{1}{2}}\}$. Let $\tau = \frac{\sigma_{\max}}{\sigma_{\min}}$. We assume $\tau = \Theta(1)$, indicating that $\sigma_{\max}$ and $\sigma_{\min}$ are in the same order. Let $\delta_i(\boldsymbol{W}^*)$ denote the $i$-th largest singular value of $\boldsymbol{W}^*$. Let $\kappa = \frac{\delta_1(\boldsymbol{W}^*)}{\delta_K(\boldsymbol{W}^*)}$, and define $\eta = \prod_{i=1}^{K} \frac{\delta_i(\boldsymbol{W}^*)}{\delta_K(\boldsymbol{W}^*)}$.

---

[1]In practice, $\Psi$ can be estimated by the EM algorithm (Redner & Walker, 1984) and the moment-based method (Hsu & Kakade, 2013). The EM algorithm returns model parameters within Euclidean distance $O((\frac{d}{n})^{\frac{1}{2}})$ when the number of mixture components $L$ is known. When $L$ is unknown, one usually over-specifies an estimate $\bar{L} > L$, then the estimation error by the EM algorithm scales as $O((\frac{d}{n})^{\frac{1}{4}})$. Please refer to (Ho & Nguyen, 2016; Ho et al., 2020; Dwivedi et al., 2020a;b) for details.

[2]The results can be generalized to any activation function $\phi$ with bounded $\phi$, $\phi'$ and $\phi''$, where $\phi'$ is even. Examples include $\tanh$ and $\mathrm{erf}$.

[3]All the weights in the second layer are assumed to be fixed to facilitate the analysis. This is a standard assumption in theoretical generalization analysis (Zhang et al., 2019; Fu et al., 2020; Zhang et al., 2020a).

[4]Algorithm 1 employs a constant step size. One can potentially speed up the convergence, i.e., reduce $v$, by using a variable step size. We leave the corresponding theoretical analysis for future work.

**Theorem 1.** *There exist $\epsilon_0 \in (0, \frac{1}{4})$ and positive value functions $\mathcal{B}(\Psi)$ (sample complexity parameter), $q(\Psi)$ (convergence rate parameter), and $\mathcal{E}_w(\Psi), \mathcal{E}(\Psi), \mathcal{E}_l(\Psi)$ (generalization parameters) such that as long as the sample size $n$ satisfies*

$$n \geq n_{sc} := poly(\epsilon_0^{-1}, \kappa, \eta, \tau, K, \delta_1(\boldsymbol{W}^*))\mathcal{B}(\Psi)d\log^2 d, \tag{6}$$

*we have that with probability at least $1 - d^{-10}$, the iterates $\{\boldsymbol{W}_t\}_{t=1}^T$ returned by Algorithm 1 with step size $\eta_0 = O\left(\left(\sum_{l=1}^L \lambda_l(\|\boldsymbol{\mu}_l\| + \|\boldsymbol{\Sigma}_l\|^{\frac{1}{2}})^2\right)^{-1}\right)$ converge linearly with a statistical error to a critical point $\widehat{\boldsymbol{W}}_n$ with the rate of convergence $v$, i.e.,*

$$\|\boldsymbol{W}_t - \widehat{\boldsymbol{W}}_n\|_F \leq v(\Psi)^t \|\boldsymbol{W}_0 - \widehat{\boldsymbol{W}}_n\|_F + \frac{\eta_0 \xi}{1 - v(\Psi)}\sqrt{dK\log n/n}, \tag{7}$$

$$v(\Psi) = 1 - K^{-2}q(\Psi), \tag{8}$$

*where $\xi \geq 0$ is the upper bound of the entry-wise additive noise in the gradient computation.*

*Moreover, there exists a permutation matrix $\boldsymbol{P}^*$ such that*

$$\|\widehat{\boldsymbol{W}}_n - \boldsymbol{W}^*\boldsymbol{P}^*\|_F \leq \mathcal{E}_w(\Psi) \cdot poly(\kappa, \eta, \tau, \delta_1(\boldsymbol{W}^*))\Theta\left(K^{\frac{5}{2}}(1+\xi) \cdot \sqrt{d\log n/n}\right). \tag{9}$$

*The average population risk $\bar{f}$ and the group-$l$ risk $\bar{f}_l$ satisfy*

$$\bar{f} \leq \mathcal{E}(\Psi) \cdot poly(\kappa, \eta, \tau, \delta_1(\boldsymbol{W}^*))\Theta\left(K^{\frac{5}{2}}(1+\xi) \cdot \sqrt{d\log n/n}\right) \tag{10}$$

$$\bar{f}_l \leq \mathcal{E}_l(\Psi) \cdot poly(\kappa, \eta, \tau, \delta_1(\boldsymbol{W}^*))\Theta\left(K^{\frac{5}{2}}(1+\xi) \cdot \sqrt{d\log n/n}\right) \tag{11}$$

The closed-form expressions of $\mathcal{B}, q, \mathcal{E}_w, \mathcal{E}$, and $\mathcal{E}_l$ are in Section D of the supplementary material and skipped here. The quantitative impact of the GMM model parameters $\Psi$ on the learning performance varies in different regimes and can be derived from Theorem 1. The following corollary summarizes the impact of $\Psi$ on the learning performance in some sample regimes.

Table 1: Impact of GMM parameters on the learning performance in sample regimes

| | $\boldsymbol{\Sigma}_l$ changes | | $\boldsymbol{\mu}_l$ changes | $\lambda_l$ changes, constant $\|\boldsymbol{\Sigma}_j\|$'s, equal $\|\boldsymbol{\mu}_j\|$'s | |
|---|---|---|---|---|---|
| | $\|\boldsymbol{\Sigma}_l\| = o(1)$ | $\|\boldsymbol{\Sigma}_l\| = \Omega(1)$ | | if $\|\boldsymbol{\Sigma}_l\| = \sigma_{\min}^2$ | if $\|\boldsymbol{\Sigma}_l\| = \sigma_{\max}^2$ |
| $\mathcal{B}(\Psi)$, sample compl. $n_{sc}$ | $O(\|\boldsymbol{\Sigma}_l\|^{-3})$ | $O\|\boldsymbol{\Sigma}_l\|^3)$ | $O(poly(\|\boldsymbol{\mu}_l\|))^5$ | $O(\frac{1}{(1+\lambda_l)^2})$ | $O(1) - \frac{\Theta(1)}{(1+\lambda_l)^2}$ |
| conv. rate $v(\Psi) \propto -q(\Psi)$ | $1 - \Theta(\|\boldsymbol{\Sigma}_l\|^3)$ | $1 - \Theta(\frac{1}{1+\|\boldsymbol{\Sigma}_l\|})$ | $1 - \Theta(\frac{1}{\|\boldsymbol{\mu}_l\|^2 + 1})$ | $\Theta(\frac{1}{1+\lambda_l})$ | $1 - \Theta(\frac{1}{1+\lambda_l})$ |
| $\mathcal{E}_w(\Psi), \|\widehat{\boldsymbol{W}}_n - \boldsymbol{W}^*\boldsymbol{P}\|_F$ | $O(1) - \Theta(\|\boldsymbol{\Sigma}_l\|^3)$ | $O(\sqrt{\|\boldsymbol{\Sigma}_l\|})$ | $O(1 + \|\boldsymbol{\mu}_l\|)$ | $O(\frac{1}{1+\sqrt{\lambda_l}})$ | $O(1 + \sqrt{\lambda_l})$ |
| $\mathcal{E}(\Psi)$, average risk $\bar{f}$ | $O(1) - \Theta(\|\boldsymbol{\Sigma}_l\|^3)$ | $O(\|\boldsymbol{\Sigma}_l\|)$ | $O(1 + \|\boldsymbol{\mu}_l\|^2)$ | $O(\frac{1}{1+\lambda_l})$ | $O(1) - \frac{\Theta(1)}{1+\lambda_l}$ |
| $\mathcal{E}_l(\Psi)$, group-$l$ risk $\bar{f}_l$ | $O(1) - \Theta(\|\boldsymbol{\Sigma}_l\|^3)$ | $O(\|\boldsymbol{\Sigma}_l\|)$ | $O(1 + \|\boldsymbol{\mu}_l\|^2)$ | $O(\frac{1}{1+\sqrt{\lambda_l}})$ | $O(1 + \sqrt{\lambda_l})$ |

**Corollary 1.** *When we vary one parameter of group $l$ for any $l \in [L]$ of the GMM model $\Psi$ and fix all the others, the learning performance degrades in the sense that the sample complexity $n_{sc}$, the convergence rate $v$, $\|\widehat{\boldsymbol{W}}_n - \boldsymbol{W}^*\boldsymbol{P}\|_F$, average risk $\bar{f}$ and group-$l$ risk $\bar{f}_l$ all increase (details summarized in Table 1), as long as any of the following conditions happens,*

*(i) $\|\boldsymbol{\Sigma}_l\|$ approaches 0;   (ii) $\|\boldsymbol{\Sigma}_l\|$ increases from some constant;   (iii) $\|\boldsymbol{\mu}_l\|$ increases from 0,*

*(iv) $\lambda_l$ decreases, provided that $\|\boldsymbol{\Sigma}_l\| = \sigma_{\min}^2$, i.e., group $l$ has the smallest group-level co-variance, where $\|\boldsymbol{\Sigma}_j\|$ are all constants, and $\|\boldsymbol{\mu}_i\| = \|\boldsymbol{\mu}_j\|$ for all $i, j \in [L]$.*

*(v) $\lambda_l$ increases, provided that $\|\boldsymbol{\Sigma}_l\| = \sigma_{\max}^2$, i.e., group $l$ has the largest group-level co-variance, where $\|\boldsymbol{\Sigma}_j\|$ are all constants, and $\|\boldsymbol{\mu}_i\| = \|\boldsymbol{\mu}_j\|$ for all $i, j \in [L]$.*

To the best of our knowledge, Theorem 1 provides the first characterization of the sample complexity, learning rate, and generalization performance under the Gaussian mixture model. It also firstly characterizes the per-group generalization performance in addition to the average generalization.

---

[5] $poly(\|\boldsymbol{\mu}_l\|)$ is $\|\boldsymbol{\mu}_l\|^4$ for $\|\boldsymbol{\mu}_l\| \leq 1$, and $\|\boldsymbol{\mu}_l\|^{12}$ for $\|\boldsymbol{\mu}_l\| > 1$.

### 4.1 THEORETICAL INSIGHTS

We summarize the crucial implications of Theorem 1 and Corollary 1 as follows.

**(P1). Training convergence and generalization guarantee**. The iterates $\boldsymbol{W}_t$ converge to a critical point $\widehat{\boldsymbol{W}}_n$ linearly, and the distance between $\widehat{\boldsymbol{W}}_n$ and $\boldsymbol{W}^*\boldsymbol{P}^*$ is $O(\sqrt{d \log n/n})$ for a certain permutation matrix $\boldsymbol{P}^*$. When the computed gradients contain noise, there is an additional error term of $O(\xi\sqrt{d \log n/n})$, where $\xi$ is the noise level ($\xi = 0$ for noiseless case). Moreover, the average risk of all groups and the risk of each individual group are both $O((1+\xi)\sqrt{d \log n/n})$.

**(P2). Sample complexity.** For a given GMM, the sample complexity is $\Theta(d \log^2 d)$, where $d$ is the feature dimension. This result is in the same order as the sample complexity for the standard Gaussian input in (Fu et al., 2020) and (Zhong et al., 2017b). Our bound is almost order-wise optimal with respect to $d$ because the degree of freedom is $dK$. The additional multiplier of $\log^2 d$ results from the concentration bound in the proof technique. We focus on the dependence on the feature dimension $d$ and treat the network width $K$ as constant. The sample complexity in (Fu et al., 2020) and (Zhong et al., 2017b) is also $d \cdot \text{poly}(K, \log d)$.

**(P3). Learning performance is improved at a medium regime of group-level co-variance**. On the one hand, when $\|\boldsymbol{\Sigma}_l\|$ is $\Omega(1)$, the learning performance degrades as $\|\boldsymbol{\Sigma}_l\|$ increases in the sense that the sample complexity $n_{sc}$, the convergence rate $v$, the estimation error of $\boldsymbol{W}^*$, the average risk $\bar{f}$, and the group-$l$ risk $\bar{f}_l$ all increase. This is due to the saturation of the loss and gradient when the samples have a large magnitude. On the other hand, when $\|\boldsymbol{\Sigma}_l\|$ is $o(1)$, the learning performance also degrades when $\|\boldsymbol{\Sigma}_l\|$ approaches zero. The intuition is that in this regime, the input data are concentrated on a few vectors, and the optimization problem does not have a benign landscape.

**(P4). Increasing the fraction of the minority group data does not always improve the generalization**, while the performance also depends on the mean and co-variance of individual groups. Take $\|\boldsymbol{\Sigma}_j\| = \Theta(1)$ for all group $j$, and $\|\boldsymbol{\mu}_j\|$ is the same for all $j$ as an example (columns 5 and 6 of Table 1). When $\|\boldsymbol{\Sigma}_l\|$ is the smallest among all groups, increasing $\lambda_l$ improves the learning performance. When $\|\boldsymbol{\Sigma}_l\|$ is the largest among all groups, increasing $\lambda_l$ actually degrades the performance. The intuition is that from (P3), the learning performance is enhanced at a medium regime of group-level co-variance. Thus, increasing the fraction of a group with a medium level of co-variance improves the performance, while increasing the fraction of a group with large co-variance degrades the learning performance. Similarly, when augmenting the training data, an argumentation method that introduces medium variance could improve the learning performance, while an argumentation method that introduces a significant level of variance could hurt the learning performance.

**(P5). Group-level mean shifts from zero degrade the learning performance**. The learning performance degrades as $\|\boldsymbol{\mu}_l\|$ increases. An intuitive explanation of the degradation is that some training samples have a significant large magnitude such that the sigmoid function saturates.

### 4.2 PROOF IDEA AND TECHNICAL NOVELTY

Different from the analysis based on generalized linear models, our paper deals with more technical challenges of nonconvex optimization due to the multi-neuron architecture, the GMM model, and a more complicated activation and loss. The main idea of proof is to show that the nonconvex empirical risk $f_n(\boldsymbol{W})$ in a small neighborhood around $\boldsymbol{W}^*$ (or any permutation $\boldsymbol{W}^*\boldsymbol{P}$) is almost convex with a sufficiently large $n$. Then if $\boldsymbol{W}_0$ can be initialized in any of these local regions, gradient-based iterates can be proved to converge to $\boldsymbol{W}^*$ (or $\boldsymbol{W}^*\boldsymbol{P}$). The idea of tensor initialization is to first find quantities (see $\boldsymbol{Q}_j$ in (14) in the supplementary material) which are proven to be functions of tensors of $\boldsymbol{w}_i^*$. Then the method approximates these quantities numerically using training samples and then applies the tensor decomposition method on the estimated quantities to obtain $\boldsymbol{W}_0$, which is an estimation of $\boldsymbol{W}^*$. With a large number of training samples $n$, the estimation $\boldsymbol{W}_0$ can be proved to be in the local convex region. The full proof is in Section D of the supplementary material.

Our algorithmic and analytical framework is built upon some recent works on the generalization analysis of one-hidden-layer neural networks, see, e.g., (Zhong et al., 2017b; Zhang et al., 2019; Fu et al., 2020; Zhang et al., 2020a; 2021b), which assume that $\boldsymbol{x}_i$ follows the standard Gaussian distribution and cannot be directly extended to GMM. This paper makes new technical contributions from the following aspects. First, we characterize the local convex region near $\boldsymbol{W}^*$ for the GMM

model, while existing results only hold for standard Gaussian data. Second, new tools including matrix concentration bounds are develped to explicitly quantify the impact of $\Psi$ on the sample comeplxity. Third, we design and analyze new tensors for the mixture model to initialize properly, while the previous tensor methods in (Zhong et al., 2017b; Zhang et al., 2019; Fu et al., 2020; Zhang et al., 2020a) utilize the rotation invariant property that only holds for zero mean Gaussian.

## 5 NUMERICAL EXPERIMENTS

### 5.1 EXPERIMENTS ON SYNTHETIC DATASETS

We first verify the theoretical bounds in Theorem 1 on synthetic data. Each entry of $\boldsymbol{W}^* \in \mathbb{R}^{d \times K}$ is generated from $\mathcal{N}(0, 1)$. The training data $\{\boldsymbol{x}_i, y_i\}_{i=1}^n$ is generated using the GMM model and (1). If not otherwise specified, $L = 2$, $d = 5$, and $K = 3$[6]. To reduce the computational time, we randomly initialize near $\boldsymbol{W}^*$ instead of computing the tensor initialization[7].

**Sample complexity**. We first study the impact of $d$ on the sample complexity. Let $\boldsymbol{\mu}_1 = \boldsymbol{1}$ in $\mathbb{R}^d$ and let $\boldsymbol{\mu}_2 = \boldsymbol{0}$. $\boldsymbol{\Sigma}_1 = \boldsymbol{\Sigma}_2 = \boldsymbol{I}$. $\lambda_1 = \lambda_2 = 0.5$. We randomly initialize $M$ times and let $\widehat{\boldsymbol{W}}_n^{(m)}$ denote the output of Algorithm 1 in the $m$th trail. Let $\bar{\boldsymbol{W}}_n$ denote the mean values of all $\widehat{\boldsymbol{W}}_n^{(m)}$, and let $V_W = \sqrt{\sum_{m=1}^M ||\widehat{\boldsymbol{w}}_n^m - \bar{\boldsymbol{W}}_n||^2/M}$ denote the variance. An experiment is successful if $V_W \leq 10^{-3}$ and fails otherwise. $M$ is set to 20. For each pair of $d$ and $n$, 20 independent sets of $\boldsymbol{W}^*$ and the corresponding training samples are generated. Figure 2 (a) shows the success rate of these independent experiments. A black block means that all the experiments fail. A white block means that they all succeed. The sample complexity is indeed almost linear in $d$, as predicted by (6).

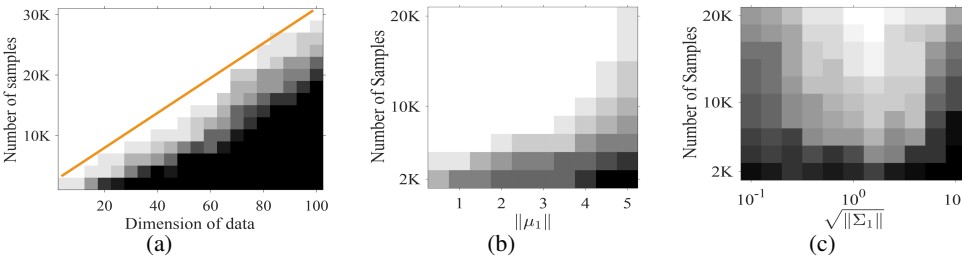

Figure 2: The sample complexity (a) when the feature dimension changes, (b) when one mean changes, (c) when one co-variance changes.

We next study the impact on the sample complexity of the GMM model. In Figure 2 (b), $\boldsymbol{\Sigma}_1 = \boldsymbol{\Sigma}_2 = \boldsymbol{I}$, and let $\boldsymbol{\mu}_1 = \mu \cdot \boldsymbol{1}$, $\boldsymbol{\mu}_2 = -\boldsymbol{1}$. $\|\boldsymbol{\mu}_1\|$ varies from 0 to 5. Figure 2(b) shows that when the mean increases, the sample complexity increases. In Figure 2 (c), we fix $\boldsymbol{\mu}_1 = \boldsymbol{1}$, $\boldsymbol{\mu}_2 = -\boldsymbol{1}$, and let $\boldsymbol{\Sigma}_1 = \sigma^2 \boldsymbol{I}$ and $\boldsymbol{\Sigma}_2 = \boldsymbol{I}$. $\sigma$ varies from $10^{-1}$ to $10^1$. The sample complexity increases both when $\|\boldsymbol{\Sigma}_1\|$ increases and when $\|\boldsymbol{\Sigma}_1\|$ approaches zero. All results match predictions in Corollary 1.

**Convergence analysis**. We next study the convergence rate of Algorithm 1. Figure 3(a) shows the impact of $\|\boldsymbol{\mu}_l\|$. $\lambda_1 = \lambda_2 = 0.5$, $\boldsymbol{\mu}_1 = -\boldsymbol{\mu}_2 = C \cdot \boldsymbol{1}$ for a positive $C$, and $\boldsymbol{\Sigma}_1 = \boldsymbol{\Sigma}_2 = \boldsymbol{\Lambda}^\top \boldsymbol{D} \boldsymbol{\Lambda}$. Here $\boldsymbol{\Lambda}$ is generated by computing the left-singular vectors of a $d \times d$ random matrix from the Gaussian distribution. $\boldsymbol{D} = \text{diag}(1, 1.1, 1.2, 1.3, 1.4)$. $n = 1 \times 10^4$. Algorithm 1 always converges linearly when $\|\boldsymbol{\mu}_1\|$ changes. Moreover, as $\|\boldsymbol{\mu}_1\|$ increases, Algorithm 1 converges slower. Figure 3 (b) shows the impact of the variance of the Gaussian mixture model. $\lambda_1 = \lambda_2 = 0.5$, $\boldsymbol{\mu}_1 = \boldsymbol{1}$, $\boldsymbol{\mu}_2 = -\boldsymbol{1}$, $\boldsymbol{\Sigma}_1 = \boldsymbol{\Sigma}_2 = \boldsymbol{\Sigma} = \sigma^2 \cdot \boldsymbol{\Lambda}^\top \boldsymbol{D} \boldsymbol{\Lambda}$. $n = 5 \times 10^4$. We change $\|\boldsymbol{\Sigma}\|$ by changing $\sigma$. Among

---

[6]Like Zhong et al. (2017b); Zhang et al. (2019); Fu et al. (2020), we consider a small-sized network in synthetic experiments to reduce the computational time, especially for computing the sample complexity in Figure 2. Our results hold for large networks too.

[7]The existing methods based on tensor initialization all use random initialization in synthetic experiments to reduce the computational time. See Fu et al. (2020); Zhang et al. (2019; 2020a; 2021b;a) as examples. We compare tensor initialization and local random initialization numerically in Section B.1 of the supplementary material and show that they have the same performance.

the values we test, Algorithm 1 converges fastest when $\|\Sigma\| = 1$. The convergence rate slows down when $\|\Sigma\|$ increases or decreases from 1. All results are consistent with the predictions in Corollary 1. We then study the impact of $K$ on the convergence rate. $\lambda_1 = \lambda_2 = 0.5$, $\boldsymbol{\mu}_1 = \mathbf{1}$, $\boldsymbol{\mu}_2 = -\mathbf{1}$, $\Sigma_1 = \Sigma_2 = \boldsymbol{I}$. Figure 2 (c) shows that, as predicted by (8), the convergence rate is linear in $-1/K^2$.

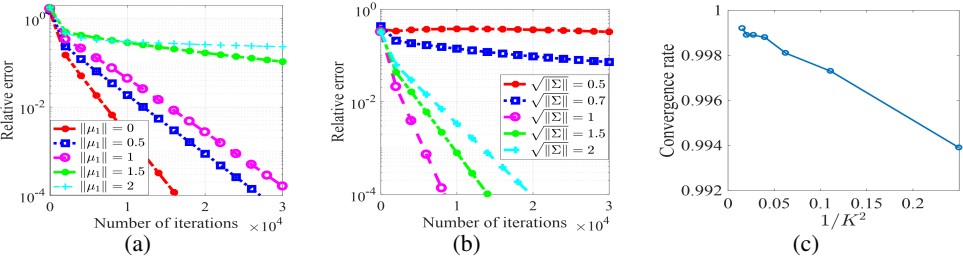

Figure 3: (a) The convergence rate with different $\boldsymbol{\mu}_1$. (b) The convergence rate with different $\Sigma$. (c) Convergence rate when the number of neurons $K$ changes.

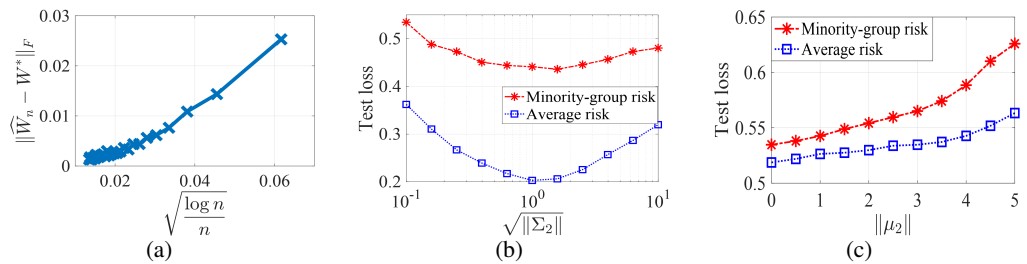

Figure 4: (a) The relative error of the learned model when $n$ changes (b) The cross-entropy test loss when the co-variance of the minority group changes. (c) The cross-entropy test loss when the mean of the minority group changes.

**Average and group-level generalization performance**. The distance between $\widehat{\boldsymbol{W}}_n$ returned by Algorithm 1 and $\boldsymbol{W}^*$ is measured by $\|\widehat{\boldsymbol{W}}_n - \boldsymbol{W}^*\|_F$. $n$ ranges from $2 \times 10^3$ to $6 \times 10^4$. $\Sigma_1 = \Sigma_2 = 9\boldsymbol{I}$, $\boldsymbol{\mu}_1 = \mathbf{1}$, $\boldsymbol{\mu}_2 = -\mathbf{1}$. Each point in Figure 4 (a) is averaged over 20 experiments of different $\boldsymbol{W}^*$ and training set. The error is indeed linear in $\sqrt{\log(n)/n}$, as predicted by (7).

We evaluate the impact of one mean/co-variance of the minority group on the generalization. $n = 2 \times 10^4$. Let $\lambda_1 = 0.8$, $\lambda_2 = 0.2$, $\boldsymbol{\mu}_1 = -\mathbf{1}$, $\Sigma_1 = \boldsymbol{I}$. First, we let $\boldsymbol{\mu}_2 = \mu_2 \cdot \mathbf{1}$ and $\Sigma_2 = \boldsymbol{I}$. Figure 4 (c) shows that both the average risk and the group-2 risk increase as $\mu_2$ increases, consistent with (P5). Then we set $\boldsymbol{\mu}_2 = 2 \cdot \mathbf{1}$, $\Sigma_2 = \sigma_2^2 \cdot \boldsymbol{I}$. Figure 4 (b) indicates that both the average and the group-2 risk will first decrease and then increase as the $\sigma_2$ increases, consistent with (P3).

Next, we study the impact of increasing the fraction of the minority group. $\boldsymbol{\mu}_1 = \boldsymbol{\mu}_2 = 0$. Let group 2 be the minority group. In Figure 5 (a), $\Sigma_1 = 10 \cdot \boldsymbol{I}$ and $\Sigma_2 = \boldsymbol{I}$, the minority group has a smaller level of co-variance. Then when $\lambda_2$ increases from 0 to 0.5, both the average and group-2 risk decease. In Figure 5 (b), $\Sigma_1 = \boldsymbol{I}$ and $\Sigma_2 = 10 \cdot \boldsymbol{I}$, and the minority group has a higher-level of co-variance. Then when $\lambda_2$ increases from 0 to 0.3, both the average and group-2 risk increase. As predicted by insight (P4), increasing $\lambda_2$ does not necessarily improve the generalization of group 2.

## 5.2 IMAGE CLASSIFICATION ON DATASET CELEBA

We choose the attribute "blonde hair" as the binary classification label. ResNet 9 He et al. (2016) is selected to be the learning model here because it was applied in many simple computer vision tasks Wu et al. (2018); Dutta et al. (2020). To study the impact of co-variance, we pick 4000 female (majority) and 1000 male (minority) images and implement Gaussian data augmentation to create additional 300 images for the male group. Specifically, we select 300 out of 1000 male images and add i.i.d. noise drawn from $\mathcal{N}(0, \delta^2)$ to every entry. The test set includes 500 male and 500

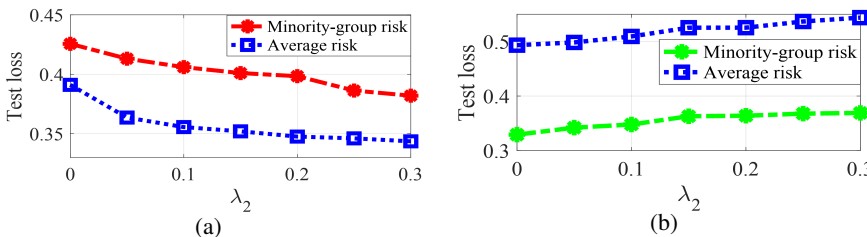

Figure 5: The test loss (cross entropy loss) of synthetic data with different $\lambda_2$ values. (a) Group 2 has a smaller level of co-variance. (b) Group 2 has a larger level of co-variance.

female images. Figure 1 shows that when $\delta^2$ increases, i.e., when the co-variance of the minority group increases, both the minority-group and average test accuracy increase first and then decrease, coinciding with our insight (P3).

Then we fix the total number of training data to be $5000$ and vary the fractions of the two groups. From Figure 6(a)[8] and (b), we observe opposite trends if we increase the fraction of the minority group in the training data with the male being the minority and the female being the minority. This is consistent with Insight (P4). Due to space limit, our results on the CIFAR10 dataset are deferred to Section A in the supplementary material.

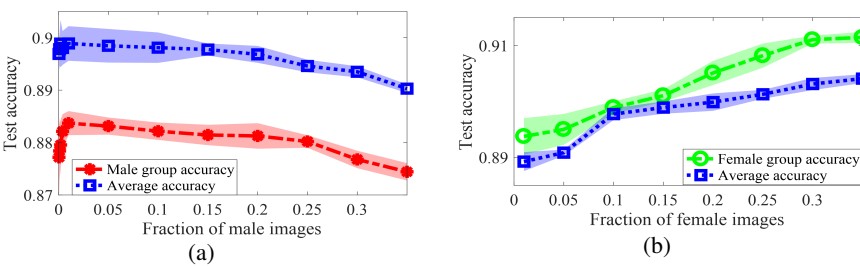

Figure 6: The test accuracy on CelebA dataset has opposite trends when the minority group fraction increases. (a) Male group is the minority (b) Female group is the minority

# 6 CONCLUSIONS, LIMITATIONS AND FUTURE DIRECTIONS

This paper provides a novel theoretical framework for characterizing neural network generalization with group imbalance. The group imbalance is formulated using the Gaussian mixture model, and this paper explicitly quantifies the impact of each group on the sample complexity, convergence rate, and the average and the group-level generalization. The learning performance is enhanced when the group-level covariance is at a medium regime, and the group-level mean is close to zero. Moreover, increasing the fraction of minority group does not guarantee improved group-level generalization.

Our results are limited to one-hidden-layer neural networks for binary classification problems. One future direction is to extend the analysis to multiple-hidden-layer neural networks and multi-class classification. Because of the concatenation of nonlinear activation functions, the analysis of the landscape of the empirical risk and the design of a proper initialization is more challenging and requires the development of new tools. Like many existing works, our sample complexity analysis is also based on the sufficient condition for training success, although it is already almost order-wise optimal. Another future direction is to formally characterize the information-theoretic lower bound of the sample complexity. We see no ethical or immediate negative societal consequence of our work.

---

[8]In Figure 6(a), when the minority fraction is less than $0.01$, the minority group distribution is almost removed from the Gaussian mixture model in the analysis. Then the $O(1)$ constants in the last column of Table 1 have some minor changes, and the order-wise analyses do not reflect the minor fluctuations in this regime.

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

We begin our Appendix here.

**Section A** provides more experiment results as a supplement of Section 5.

**Section B** introduces the algorithm, especially the tensor initialization in detail.

**Section C** includes some definitions and properties as a preliminary to our proof.

**Section D** shows the proof of Theorem 1 and Corollary 1, followed by **Section E, F, and G** as the proof of three key Lemmas about local convexity, linear convergence and tensor initialization, respectively.

## A    MORE EXPERIMENT RESULTS

We present our experiment resultson empirical datasets CelebA (Liu et al., 2015) and CIFAR-10 [9] in this section. To be more specific, we evaluate the impact of the variance levels introduced by different data augmentation methods on the learning performance. We also evaluate the impact of the minority group fraction in the training data on the learning performance. All the experiments are reported in a format of "mean$\pm 2 \times$standard deviation" with a random seed equal to $10$. We implement our experiments on an NVIDIA GeForce RTX 2070 super GPU and a work station with 8 cores of 3.40GHz Intel i7 CPU.

### A.1    TESTS ON CELEBA

In addition to the Gaussian augmentation method in Figure 7 (a) and Figure 1 (b). We also evaluate the performance of data augmentation by cropping. The setup is exactly the same as that for Gaussian augmentation, expect that we augment the data by cropping instead of adding Gaussian noise. Specifically, to generate an augmented image, we randomly crop an image with a size $w \times w \times 3$ and then resize back to $224 \times 224 \times 3$. One can observe that the minority-group and average test accuracy first increase and then decrease as $w$ increases, which is in accordance with Insight (P3).

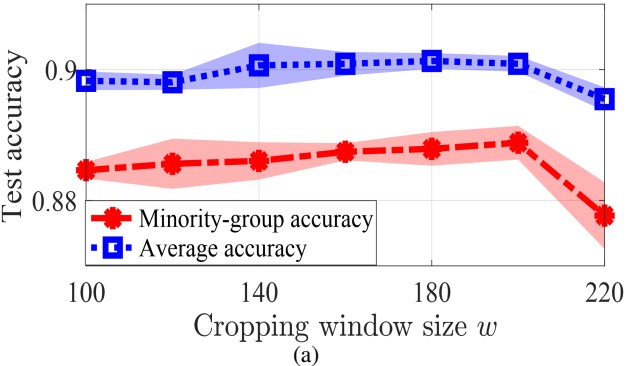

Figure 7: The test accuracy of CelebA dataset with the data augmentation method of cropping.

### A.2    TESTS ON CIFAR-10

Group 1 contains images with attributes "bird", "cat", "deer", "dog", "frog" and "horse." Group 2 contains "airplane" images. In this setting, Group 1 has a larger variance. Because each image in CIFAR-10 only has one attribute, we consider the binary classification setting where all images in Group 1 are labeled as "animal" and all images are labeled as "airplane." This is a special scenario that the group label is also the classification label. Note that our results hold for general setups where group labels and classification labels are irrelevant, like our previous results on CelebA. LeNet 5 Lecun et al. (1998) is selected to be the learning model.

---

[9]Alex Krizhevsky, Vinod Nair, and Geoffrey Hinton. The CIFAR-10 dataset. `www.cs.toronto.edu/~kriz/cifar.html`

We first pick 8000 animal images (majority) and 2000 airplane images (minority). We select 1000 out of 2000 airplane images to implement data augmentation, including both Gaussian augmentation and random cropping. For Gaussian augmentation, we add i.i.d. Gaussian noise drawn from $\mathcal{N}(0, \delta^2)$ to each entry[10]. For random cropping, we randomly crop the image with a certain size $w \times w \times 3$ and then resizing back to $32 \times 32 \times 3$. Figure 8 shows that when $\delta$ or $w$ increase, i.e., the variance introduced by either augmentation method increases, both the minority-group and average test accuracy increase first and then decrease, which is consistent with our Insight (P3).

Then we fix the total number of training data to be 5000 and vary the fractions of the two groups. One can see opposite trends in Figure 9 if we increase the fraction of the minority group with the airplane being the minority and the animal being the minority, which reflects our Insight (P4).

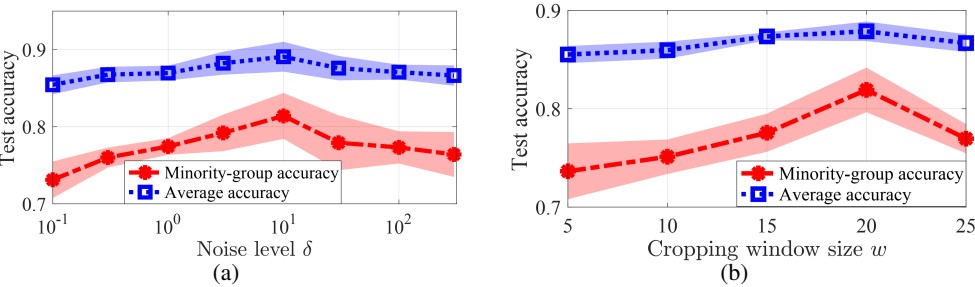

Figure 8: The test accuracy of CIFAR-10 dataset with different data augmentation methods (a) Gaussian noise (b) cropping.

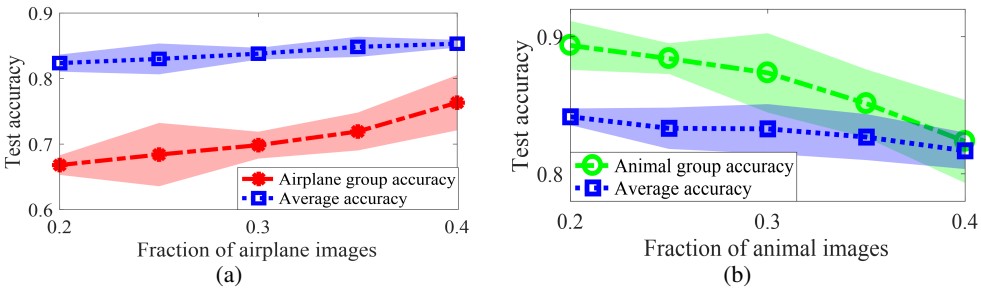

Figure 9: The test accuracy of CIFAR-10 dataset has opposite trends when the minority group fraction increases (a) Airplane group is the minority. (b) Animal group is the minority.

## B  ALGORITHM

We first introduce new notations to be used in this part and summarize key notions in Table 2. We write $f(x) \lesssim (\gtrsim)g(x)$ if $f(x) \leq (\geq)\Theta(g(x))$. The gradient and the Hessian of a function $f(\boldsymbol{W})$ are denoted by $\nabla f(\boldsymbol{W})$ and $\nabla^2 f(\boldsymbol{W})$, respectively. $\boldsymbol{A} \succeq 0$ means $\boldsymbol{A}$ is a positive semi-definite (PSD) matrix. $\boldsymbol{A}^{\frac{1}{2}}$ means that $\boldsymbol{A} = (\boldsymbol{A}^{\frac{1}{2}})^2$. The outer product of vectors $\boldsymbol{z}_i \in \mathbb{R}^{n_i}$, $i \in [l]$, is defined as $\boldsymbol{T} = \boldsymbol{z}_1 \otimes \cdots \otimes \boldsymbol{z}_l \in \mathbb{R}^{n_1 \times \cdots \times n_l}$ with $\boldsymbol{T}_{j_1 \cdots j_l} = (\boldsymbol{z}_1)_{j_1} \cdots (\boldsymbol{z}_l)_{j_l}$. Given a tensor $\boldsymbol{T} \in \mathbb{R}^{n_1 \times n_2 \times n_3}$ and matrices $\boldsymbol{A} \in \mathbb{R}^{n_1 \times d_1}$, $\boldsymbol{B} \in \mathbb{R}^{n_2 \times d_2}$, $\boldsymbol{C} \in \mathbb{R}^{n_3 \times d_3}$, the $(i_1, i_2, i_3)$-th entry of the tensor $\boldsymbol{T}(\boldsymbol{A}, \boldsymbol{B}, \boldsymbol{C})$ is given by

$$\sum_{i_1'}^{n_1} \sum_{i_2'}^{n_2} \sum_{i_3'}^{n_3} \boldsymbol{T}_{i_1', i_2', i_3'} \boldsymbol{A}_{i_1', i_1} \boldsymbol{B}_{i_2', i_2} \boldsymbol{C}_{i_3', i_3}. \tag{12}$$

---

[10]In this experiment, the noise is added to the raw image where the pixel value ranges from 0 to 255, while in the experiment of CelebA (Figure 1 (b)), the noise is added to the image after normalization where the pixel value ranges from 0 to 1.

Table 2: Summary of notations

| | |
|---|---|
| $\lambda_l$, $\boldsymbol{\mu}_l$, $\boldsymbol{\Sigma}_l$, $l \in [L]$ | The fraction, mean, and covariance of the $l$-th component in the Gaussian mixture distribution, respectively. |
| $d$, $n$, $K$ | The feature dimension, the number of training samples, and the number of neurons, respectively. |
| $\boldsymbol{W}^*$, $\boldsymbol{W}_t$ | $\boldsymbol{W}^*$ is the ground truth weight. $\boldsymbol{W}_t$ is the updated weight in the $t$-th iteration. |
| $f_n$, $\bar{f}$, $\ell$ | $f_n$ is the empirical risk function. $\bar{f}$ is the average risk or the population risk function. $\ell$ is the cross-entropy loss function. |
| $\Psi$, $\sigma_{\max}$, $\sigma_{\min}$, $\tau$ | $\Psi$ denotes our Gaussian mixture model $(\lambda_l, \boldsymbol{\mu}_l, \boldsymbol{\Sigma}_l, \forall l)$. $\sigma_{\max} = \max_{l \in [L]}\{\|\boldsymbol{\Sigma}_l\|^{\frac{1}{2}}\}$. $\sigma_{\min} = \min_{l \in [L]}\{\|\boldsymbol{\Sigma}_l^{-1}\|^{-\frac{1}{2}}\}$. $\tau = \sigma_{\max}/\sigma_{\min}$. |
| $\delta_i(\boldsymbol{W}^*)$, $\eta$, $\kappa$, $i \in [K]$ | $\delta_i(\boldsymbol{W}^*)$ is the $i$-th largest singular value of $\boldsymbol{W}^*$. $\eta$ and $\kappa$ are two functions of $\boldsymbol{W}^*$. |
| $\rho(\boldsymbol{u}, \sigma)$, $\Gamma(\Psi)$, $D_m(\Psi)$ | These items are functions of the Gaussian mixture distribution $\Psi$ used to develop our Theorem 1. |
| $\boldsymbol{\nu}_i$, $\xi$ | $\boldsymbol{\nu}_i$ is the gradient noise. $\xi$ is the upper bound of the noise level. |
| $\boldsymbol{Q}_j$, $j = 1, 2, 3$ | $\boldsymbol{Q}_j$'s are tensors used in the initialization. |
| $\mathcal{B}(\Psi)$ | A parameter appeared in the sample complexity bound (6). |
| $v(\Psi)$, $q(\Psi)$ | $v(\Psi)$ is the convergence rate (7). $q(\Psi)$ is a parameter in the definition of $v(\Psi)$ (8). |
| $\mathcal{E}_w(\Psi)$, $\mathcal{E}$, $\mathcal{E}_l$ | Generalization parameters. $\mathcal{E}_w(\Psi)$ appears in the error bound of the model (9). $\mathcal{E}(\Psi)$ and $\mathcal{E}_l(\Psi)$ are to characterize the average risk (10) and the group-l risk (11), respectively. |

The method starts from an initialization $\boldsymbol{W}_0 \in \mathbb{R}^{d \times K}$ computed based on the tensor initialization method (Subroutine 1) and then updates the iterates $\boldsymbol{W}_t$ using gradient descent with the step size $\eta_0$. To model the inaccuracy in computing the gradient, an i.i.d. zero-mean noise $\{\nu_i\}_{i=1}^n \in \mathbb{R}^{d \times K}$ with bounded magnitude $|(\nu_i)_{jk}| \leq \xi$ ($j \in [d], k \in [K]$) for some $\xi \geq 0$ are added in (13) when computing the gradient of the loss in (3).

---

**Algorithm 1** Our proposed learning algorithm

1: **Input:** Training data $\{(\boldsymbol{x}_i, y_i)\}_{i=1}^n$, the step size $\eta_0 = O\left(\left(\sum_{l=1}^L \lambda_l(\|\tilde{\boldsymbol{\mu}}_l\|_\infty + \|\boldsymbol{\Sigma}_l^{\frac{1}{2}}\|)^2\right)^{-1}\right)$, the total number of iterations $T$

2: **Initialization:** $\boldsymbol{W}_0 \leftarrow$ Tensor initialization method via Subroutine 1

3: **Gradient Descent:** for $t = 0, 1, \cdots, T-1$

$$\boldsymbol{W}_{t+1} = \boldsymbol{W}_t - \eta_0 \cdot \frac{1}{n}\sum_{i=1}^n (\nabla l(\boldsymbol{W}, \boldsymbol{x}_i, y_i) + \nu_i)$$

$$= \boldsymbol{W}_t - \eta_0 \left(\nabla f_n(\boldsymbol{W}) + \frac{1}{n}\sum_{i=1}^n \nu_i\right) \tag{13}$$

4: **Output:** $\boldsymbol{W}_T$

---

Our tensor initialization method in Subroutine 1 is extended from Janzamin et al. (2014) and Zhong et al. (2017b). The idea is to compute quantities ($\boldsymbol{Q}_j$ in (14)) that are tensors of $\boldsymbol{w}_i^*$ and then apply the tensor decomposition method to estimate $\boldsymbol{w}_i^*$. Because $\boldsymbol{Q}_j$ can only be estimated from training samples, tensor decomposition does not return $\boldsymbol{w}_i^*$ exactly but provides a close approximation, and this approximation is used as the initialization for Algorithm 1. Because the existing method on tensor construction only applies to the standard Gaussian distribution, we exploit the relationship between probability density functions and tensor expressions developed in Janzamin et al. (2014) to design tensors suitable for the Gaussian mixture model. Formally,

**Definition 1.** *For $j = 1, 2, 3$, we define*

$$\boldsymbol{Q}_j := \mathbb{E}_{\boldsymbol{x} \sim \sum_{l=1}^L \lambda_l \mathcal{N}(\boldsymbol{\mu}_l, \boldsymbol{\Sigma}_l)}[y \cdot (-1)^j p^{-1}(\boldsymbol{x}) \nabla^{(j)} p(\boldsymbol{x})], \tag{14}$$

*where $p(\boldsymbol{x})$, the probability density function of GMM is defined as*

$$p(\boldsymbol{x}) = \sum_{l=1}^{L} \lambda_l (2\pi)^{-\frac{d}{2}} |\boldsymbol{\Sigma}_l|^{-\frac{1}{2}} \exp\big(-\frac{1}{2}(\boldsymbol{x} - \boldsymbol{\mu}_l)\boldsymbol{\Sigma}_l^{-1}(\boldsymbol{x} - \boldsymbol{\mu}_l)\big) \tag{15}$$

*If the Gaussian mixture model is symmetric, the symmetric distribution can be written as*

$$\boldsymbol{x} \sim \begin{cases} \sum\limits_{l=1}^{\frac{L}{2}} \lambda_l \big(\mathcal{N}(\boldsymbol{\mu}_l, \boldsymbol{\Sigma}_l) + \mathcal{N}(-\boldsymbol{\mu}_l, \boldsymbol{\Sigma}_l)\big) & L \text{ is even} \\ \lambda_1 \mathcal{N}(\boldsymbol{0}, \boldsymbol{\Sigma}_1) + \sum\limits_{l=2}^{\frac{L-1}{2}} \lambda_l \big(\mathcal{N}(\boldsymbol{\mu}_l, \boldsymbol{\Sigma}_l) + \mathcal{N}(-\boldsymbol{\mu}_l, \boldsymbol{\Sigma}_l)\big) & L \text{ is odd} \end{cases} \tag{16}$$

$\boldsymbol{Q}_j$ is a $j$th-order tensor of $\boldsymbol{w}_i^*$, e.g., $\boldsymbol{Q}_3 = \frac{1}{K} \sum_{i=1}^{K} \mathbb{E}_{\boldsymbol{x} \sim \sum_{l=1}^{L} \lambda_l \mathcal{N}(\boldsymbol{\mu}_l, \boldsymbol{\Sigma}_l)}[\phi'''(\boldsymbol{w}_i^{*\top}\boldsymbol{x})]\boldsymbol{w}_i^{*\otimes 3}$. These quantifies cannot be directly computed from (14) but can be estimated by sample means, denoted by $\widehat{\boldsymbol{Q}}_j$ ($j = 1, 2, 3$), from samples $\{\boldsymbol{x}_i, y_i\}_{i=1}^{n}$. The following assumption guarantees that these tensors are nonzero and can thus be leveraged to estimate $\boldsymbol{W}^*$.

**Assumption 1.** *The Gaussian Mixture Model in (16) satisfies the following conditions:*

1. *$\boldsymbol{Q}_1$ and $\boldsymbol{Q}_3$ are nonzero.*

2. *If the distribution is not symmetric, then $\boldsymbol{Q}_2$ is nonzero.*

Assumption 1 is a very mild assumption[11]. Moreover, as indicated in Janzamin et al. (2014), in the rare case that some quantities $\boldsymbol{Q}_i$ ($i = 1, 2, 3$) are zero, one can construct higher-order tensors in a similar way as in Definition 1 and then estimate $\boldsymbol{W}^*$ from higher-order tensors.

Subroutine 1 describes the tensor initialization method, which estimates the direction and magnitude of $\boldsymbol{w}_j^*, j \in [K]$, separately. The direction vectors are denoted as $\bar{\boldsymbol{w}}_j^* = \boldsymbol{w}_j^*/\|\boldsymbol{w}_j^*\|$ and the magnitude $\|\boldsymbol{w}_j^*\|$ is denoted as $z_j$. Lines 2-6 estimate the subspace $\widehat{\boldsymbol{U}}$ spanned by $\{\boldsymbol{w}_1^*, \cdots, \boldsymbol{w}_K^*\}$ using $\widehat{\boldsymbol{Q}}_2$ or, in the case that $\boldsymbol{Q}_2 = 0$, a second-order tensor projected by $\widehat{\boldsymbol{Q}}_3$. Lines 7-8 estimate $\bar{\boldsymbol{w}}_j^*$ by employing the KCL algorithm Kuleshov et al. (2015). Lines 9-10 estimate the magnitude $z_j$. Finally, the returned estimation of $\boldsymbol{W}^*$ is used as an initialization $\boldsymbol{W}_0$ for Algorithm 1. The computational complexity of Subroutine 1 is $O(Knd)$ based on similar calculations as those in Zhong et al. (2017b).

### B.1 NUMERICAL EVALUATION OF TENSOR INITIALIZATION

Figure 10 shows the accuracy of the returned model by Algorithm 1. Here $n = 2 \times 10^5$, $d = 50$, $K = 2$, $\lambda_1 = \lambda_2 = 0.5$, $\boldsymbol{\mu}_1 = -0.3 \cdot \boldsymbol{1}$ and $\boldsymbol{\mu}_2 = \boldsymbol{0}$. We compare the tensor initialization with a random initialization in a local region $\{\boldsymbol{W} \in \mathbb{R}^{d \times K} : \|\boldsymbol{W} - \boldsymbol{W}^*\|_F \leq \epsilon\}$. Each entry of $\boldsymbol{W}^*$ is selected from $[-0.1, 0.1]$ uniformly. Tensor initialization in Subroutine 1 returns an initial point close to one permutation of $\boldsymbol{W}^*$, with a relative error of 0.65. If the random initialization is also close to $\boldsymbol{W}^*$, e.g., $\epsilon = 0.1$, then the gradient descent algorithm converges to a critical point from both initializations, and the linear convergence rate is the same. We also test a random initialization with each entry drawn from $\mathcal{N}(0, 25)$. The initialization is sufficiently far from $\boldsymbol{W}^*$, and the algorithm does not converge. On a MacBook Pro with Intel(R) Core(TM) i5-7360U CPU at 2.30GHz and MATLAB 2017a, it takes 5.52 seconds to compute the tensor initialization. Thus, to reduce the computational time, we consider a random initialization with $\epsilon = 0.1$ in the experiments instead of computing tensor initialization.

## C PRELIMINARIES OF THE MAIN PROOF

In this section, we introduce some **definitions** and **properties** that will be used to prove the main results.

First, we define the sub-Gaussian random variable and sub-Gaussian norm.

---

[11]By mild, we mean given $L$, if Assumption 1 is not met for some $\Psi_0$, there exists an infinite number of $\Psi'$ in any neighborhood of $\Psi_0$ such that Assumption 1 holds for $\Psi'$,

---

**Subroutine 1** Tensor Initialization Method

1: **Input:** Partition $n$ pairs of data $\{(\boldsymbol{x}_i, y_i)\}_{i=1}^n$ into three disjoint subsets $\mathcal{D}_1, \mathcal{D}_2, \mathcal{D}_3$
2: **if** the Gaussian Mixture distribution is not symmetric **then**
3:     Compute $\widehat{\boldsymbol{Q}}_2$ using $\mathcal{D}_1$. Estimate the subspace $\widehat{\boldsymbol{U}}$ by orthogonalizing the eigenvectors with respect to the $K$ largest eigenvalues of $\widehat{\boldsymbol{Q}}_2$
4: **else**
5:     Pick an arbitrary vector $\boldsymbol{\alpha} \in \mathbb{R}^d$, and use $\mathcal{D}_1$ to compute $\widehat{\boldsymbol{Q}}_3(\boldsymbol{I}_d, \boldsymbol{I}_d, \boldsymbol{\alpha})$. Estimate $\widehat{\boldsymbol{U}}$ by orthogonalizing the eigenvectors with respect to the $K$ largest eigenvalues of $\widehat{\boldsymbol{Q}}_3(\boldsymbol{I}_d, \boldsymbol{I}_d, \boldsymbol{\alpha})$.
6: **end if**
7: Compute $\widehat{\boldsymbol{R}}_3 = \widehat{\boldsymbol{Q}}_3(\widehat{\boldsymbol{U}}, \widehat{\boldsymbol{U}}, \widehat{\boldsymbol{U}})$ from data set $\mathcal{D}_2$
8: Employ the KCL algorithm to compute vectors $\{\hat{\boldsymbol{v}}_i\}_{i \in [K]}$, which are the estimates of $\{\widehat{\boldsymbol{U}}^\top \bar{\boldsymbol{w}}_i^*\}_{i=1}^K$. Then the direction vectors $\{\bar{\boldsymbol{w}}_i^*\}_{i=1}^K$ can be approximated by $\{\widehat{\boldsymbol{U}}\hat{\boldsymbol{v}}_i\}_{i=1}^K$.
9: Compute $\widehat{\boldsymbol{Q}}_1$ from data set $\mathcal{D}_3$.
10: Estimate the magnitude $\hat{\boldsymbol{z}}$ by solving the optimization problem

$$\hat{\boldsymbol{z}} = \arg \min_{\boldsymbol{\alpha} \in \mathbb{R}^K} \frac{1}{2}\|\widehat{\boldsymbol{Q}}_1 - \sum_{j=1}^K \alpha_j \bar{\boldsymbol{w}}_j^*\|^2 \tag{17}$$

11: **Return:** Use $\hat{z}_j\widehat{\boldsymbol{U}}\hat{\boldsymbol{v}}_j$ as the $j$th column of $\boldsymbol{W}_0$, $j \in [K]$.

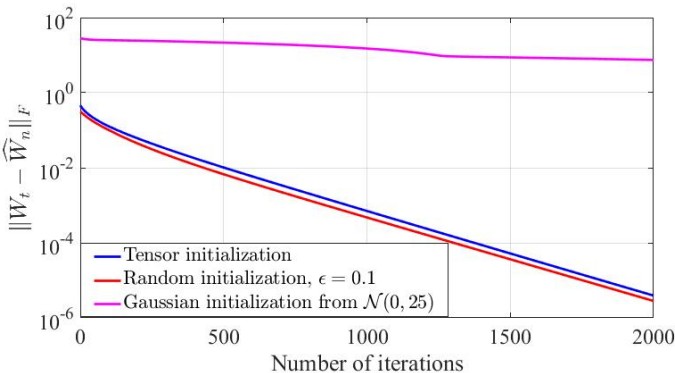

Figure 10: Comparison between tensor initialization, a random initialization near $\boldsymbol{W}^*$, and an arbitrary random initialization

**Definition 2.** *We say $X$ is a sub-Gaussian random variable with sub-Gaussian norm $K > 0$, if $(\mathbb{E}|X|^p)^{\frac{1}{p}} \leq K\sqrt{p}$ for all $p \geq 1$. In addition, the sub-Gaussian norm of X, denoted $\|X\|_{\psi_2}$, is defined as $\|X\|_{\psi_2} = \sup_{p \geq 1} p^{-\frac{1}{2}}(\mathbb{E}|X|^p)^{\frac{1}{p}}$.*

Then we define the following three quantities. $\rho(\boldsymbol{\mu}, \sigma)$ is motivated by the $\rho$ parameter for the standard Gaussian distribution in Zhong et al. (2017b), and we generalize it to a Gaussian with an arbitrary mean and variance. We define the new quantities $\Gamma(\Psi)$ and $D_m(\Psi)$ for the Gaussian mixture model.

**Definition 3.** *($\rho$-function). Let $\boldsymbol{z} \sim \mathcal{N}(\boldsymbol{u}, \boldsymbol{I}_d) \in \mathbb{R}^d$. Define $\alpha_q(i, \boldsymbol{u}, \sigma) = \mathbb{E}_{z_i \sim \mathcal{N}(u_i, 1)}[\phi'(\sigma \cdot z_i)z_i^q]$ and $\beta_q(i, \boldsymbol{u}, \sigma) = \mathbb{E}_{z_i \sim \mathcal{N}(u_i, 1)}[\phi'^2(\sigma \cdot z_i)z_i^q], \forall q \in \{0, 1, 2\}$, where $z_i$ and $u_i$ is the i-th entry of $\boldsymbol{z}$ and $\boldsymbol{u}$, respectively. Define $\rho(\boldsymbol{u}, \sigma)$ as*

$$\rho(\boldsymbol{u}, \sigma) = \min_{i,j \in [d], j \neq i}\{(u_j^2 + 1)(\beta_0(i, \boldsymbol{u}, \sigma) - \alpha_0(i, \boldsymbol{u}, \sigma)^2), \beta_2(i, \boldsymbol{u}, \sigma) - \frac{\alpha_2(i, \boldsymbol{u}, \sigma)^2}{u_i^2 + 1}\} \tag{18}$$

**Definition 4.** *($\Gamma$-function). With (18) and $\kappa, \eta$ defined in Section 3, we define*

$$\Gamma(\Psi) = \sum_{l=1}^L \frac{\lambda_l}{\tau^K \kappa^2 \eta} \frac{\|\boldsymbol{\Sigma}_l^{-1}\|^{-1}}{\sigma_{\max}^2}\rho(\frac{\boldsymbol{W}^{*\top}\boldsymbol{\mu}_l}{\delta_K(\boldsymbol{W}^*)\|\boldsymbol{\Sigma}_l^{-1}\|^{-\frac{1}{2}}}, \delta_K(\boldsymbol{W}^*)\|\boldsymbol{\Sigma}_l^{-1}\|^{-\frac{1}{2}}) \tag{19}$$

**Definition 5.** *(D-function). Given the Gaussian Mixture Model and any positive integer $m$, define $D_m(\Psi)$ as*

$$D_m(\Psi) = \sum_{l=1}^{L} \lambda_l \left( \frac{\|\boldsymbol{\mu}_l\|}{\|\boldsymbol{\Sigma}_l^{-1}\|^{-\frac{1}{2}}} + 1 \right)^m, \tag{20}$$

$\rho$-function is defined to compute the lower bound of the Hessian of the population risk with Gaussian input. $\Gamma$ function is the weighted sum of $\rho$-function under mixture Gaussian distribution. This function is positive and upper bounded by a small value. $\Gamma$ goes to zero if all $\|\boldsymbol{\mu}_l\|$ or all $\sigma_l$ goes to infinity. $D$-function is a normalized parameter for the means and variances. It is lower bounded by 1. $D$-function is an increasing function of $\|\boldsymbol{\mu}_l\|$ and a decreasing function of $\sigma_l$.

**Property 1.** *Given $\boldsymbol{W}^* = \boldsymbol{U}\boldsymbol{V} \in \mathbb{R}^{d \times k}$, where $\boldsymbol{U} \in \mathbb{R}^{d \times K}$ is the orthogonal basis of $\boldsymbol{W}^*$. For any $\boldsymbol{\mu} \in \mathbb{R}^d$, we can find an orthogonal decomposition of $\boldsymbol{\mu}$ based on the colomn space of $\boldsymbol{W}^*$, i.e. $\boldsymbol{\mu} = \boldsymbol{\mu}_U + \boldsymbol{\mu}_{U_\perp}$. If we consider the recovery problem of FCN with a dataset of Gaussian Mixture Model, in which $\boldsymbol{x}_i \sim \mathcal{N}(\boldsymbol{\mu}_h, \boldsymbol{\Sigma}_h)$ for some $h \in [L]$, the problem is equivalent to the problem of FCN with $\boldsymbol{x}_i \sim \mathcal{N}(\boldsymbol{\mu}_{U h}, \boldsymbol{\Sigma}_h)$. Hence, we can assume without loss of generality that $\boldsymbol{\mu}_l$ belongs to the column space of $\boldsymbol{W}^*$ for all $l \in [L]$.*

**Proof:**
From (1) and (3), the recovery problem can be formulated as

$$\min_{\boldsymbol{W}^*} g(\boldsymbol{W}^{*\top} \boldsymbol{x}_i, y_i)$$

For any $\boldsymbol{x}_i \sim \mathcal{N}(\boldsymbol{\mu}_h, \boldsymbol{\Sigma}_h)$, $\boldsymbol{x}_i$ can be written as

$$\boldsymbol{x}_i = \boldsymbol{z} + \boldsymbol{\mu}_h$$

where $\boldsymbol{z} \sim \mathcal{N}(\boldsymbol{0}, \boldsymbol{\Sigma}_h)$. Therefore,

$$\boldsymbol{W}^{*\top} \boldsymbol{x}_i = \boldsymbol{W}^{*\top}(\boldsymbol{z} + \boldsymbol{\mu}_h) = \boldsymbol{W}^{*\top}(\boldsymbol{z} + \boldsymbol{\mu}_{U h} + \boldsymbol{\mu}_{U_\perp h}) = \boldsymbol{W}^{*\top}(\boldsymbol{z} + \boldsymbol{\mu}_{U h})$$

The final step is because $\boldsymbol{W}^{*\top} \boldsymbol{\mu}_{U_\perp} = \boldsymbol{0}$. So the problem is equivalent to the recovery problem of FCN with $\boldsymbol{x}_i \sim \mathcal{N}(\boldsymbol{\mu}_{U h}, \boldsymbol{\Sigma}_h)$.

Recall that the gradient noise $\boldsymbol{\nu}_i \in \mathbb{R}^{d \times K}$ is zero-mean, and each of its entry is upper bounded by $\xi > 0$.

**Property 2.** *We have that $\|\boldsymbol{\nu}_i\|_F$ is a sub-Gaussian random variable with its sub-Gaussian norm bounded bu $\xi\sqrt{dK}$.*

**Proof:**

$$(\mathbb{E}\|\boldsymbol{\nu}_i\|_F^p)^{\frac{1}{p}} \le (\mathbb{E}|\sqrt{dK}\xi|^p)^{\frac{1}{p}} \le \xi\sqrt{dK} \tag{21}$$

We state some general properties of the $\rho$ function defined in Definition 3 in the following.

**Property 3.** *$\rho(\boldsymbol{u}, \sigma)$ in Definition 3 satisfies the following properties,*

1. *(**Positive**) $\rho(\boldsymbol{u}, \sigma) > 0$ for any $\boldsymbol{u} \in \mathbb{R}^d$ and $\sigma \neq 0$.*

2. *(**Finite limit point for zero mean**) $\rho(\boldsymbol{u}, \sigma)$ converges to a positive value function of $\sigma$ as $u_i$ goes to 0, i.e. $\lim_{u_i \to 0} \rho(\boldsymbol{u}, \sigma) := \mathcal{C}_m(\sigma)$.*

3. *(**Finite limit point for zero variance**) When all $u_i \neq 0$ ($i \in [d]$), $\rho(\frac{\boldsymbol{u}}{\sigma}, \sigma)$ converges to a strictly positive real function of $\boldsymbol{u}$ as $\sigma$ goes to 0, i.e. $\lim_{\sigma \to 0} \rho(\frac{\boldsymbol{u}}{\sigma}, \sigma) := \mathcal{C}_s(\boldsymbol{u})$. When $u_i = 0$ for some $i \in [d]$, $\lim_{\sigma \to 0} \rho(\frac{\boldsymbol{u}}{\sigma}, \sigma) = 0$.*

4. *(**Lower bound function of the mean**) When everything else except $|u_i|$ is fixed, $\rho(\frac{\boldsymbol{W}^{*\top}\boldsymbol{u}}{\sigma\delta_K(\boldsymbol{W}^*)}, \sigma\delta_K(\boldsymbol{W}^*))$ is lower bounded by a strictly positive real function, $\mathcal{L}_m(\frac{(\boldsymbol{\Lambda}\boldsymbol{W}^*)^\top \boldsymbol{\Lambda}\boldsymbol{u}}{\sigma\delta_K(\boldsymbol{W}^*)}, \sigma\delta_K(\boldsymbol{W}^*))$, which is monotonically decreasing as $|u_i|$ increases.*

5. (**Lower bound function of the variance**) *When everything else except $\sigma$ is fixed, $\rho(\frac{\boldsymbol{W}^{*\top}\boldsymbol{u}}{\sigma\delta_K(\boldsymbol{W}^*)}, \sigma\delta_K(\boldsymbol{W}^*))$ is lower bounded by a strictly positive real function, $\mathcal{L}_s(\frac{\boldsymbol{W}^{*\top}\boldsymbol{u}}{\sigma\delta_K(\boldsymbol{W}^*)}, \sigma\delta_K(\boldsymbol{W}^*))$, which satisfies the following conditions: (a) there exists $\zeta_{s'} > 0$, such that $\sigma^{-1}\mathcal{L}_s(\frac{\boldsymbol{W}^{*\top}\boldsymbol{u}}{\sigma\delta_K(\boldsymbol{W}^*)}, \sigma\delta_K(\boldsymbol{W}^*))$ is an increasing function of $\sigma$ when $\sigma \in (0, \zeta_{s'})$; (b) there exists $\zeta_s > 0$ such that $\mathcal{L}_s(\frac{\boldsymbol{W}^{*\top}\boldsymbol{u}}{\sigma\delta_K(\boldsymbol{W}^*)}, \sigma\delta_K(\boldsymbol{W}^*))$ is a decreasing function of $\sigma$ when $\sigma \in (\zeta_s, +\infty)$.*

**Proof:**

(1) From Cauchy Schwarz's inequality, we have

$$\mathbb{E}_{z_i \sim \mathcal{N}(u_i, 1)}[\phi'(\sigma \cdot z_i)] \leq \sqrt{\mathbb{E}_{z_i \sim \mathcal{N}(u_i, 1)}[\phi'^2(\sigma \cdot z_i)]} \tag{22}$$

$$\mathbb{E}_{z_i \sim \mathcal{N}(u_i, 1)}[\phi'(\sigma \cdot z_i)z_i \cdot z_i] \leq \sqrt{\mathbb{E}_{z_i \sim \mathcal{N}(u_i, 1)}[\phi'^2(\sigma \cdot z_i)z_i^2]} \cdot \sqrt{\mathbb{E}_{z_i \sim \mathcal{N}(u_i, 1)}[z_i^2]}$$
$$= \sqrt{\mathbb{E}_{z_i \sim \mathcal{N}(u_i, 1)}[\phi'^2(\sigma \cdot z_i)z_i^2]} \cdot \sqrt{u_i^2 + 1} \tag{23}$$

The equalities of the (22) and (23) hold if and only if $\phi'$ is a constant function. Since that $\phi$ is the sigmoid function, the equalities of (22) and (23) cannot hold.

By the definition of $\rho(\boldsymbol{u}, \sigma)$ in Definition 3, we have

$$\beta_0(i, \boldsymbol{u}, \sigma) - \alpha_0^2(i, \boldsymbol{u}, \sigma) > 0, \tag{24}$$

$$\beta_2(i, \boldsymbol{u}, \sigma) - \frac{\alpha_2^2(i, \boldsymbol{u}, \sigma)}{u_i^2 + 1} > 0. \tag{25}$$

Therefore,

$$\rho(\boldsymbol{u}, \sigma) > 0 \tag{26}$$

(2) We can derive that

$$\lim_{u_i \to 0}(\frac{u_j^2}{\sigma^2} + 1)\big(\beta_0(i, \boldsymbol{u}, \sigma) - \alpha_0^2(i, \boldsymbol{u}, \sigma)\big)$$
$$= \lim_{u_i \to 0}(\frac{u_j^2}{\sigma^2} + 1)\big(\int_{-\infty}^{\infty}\phi'^2(\sigma \cdot z_i)(2\pi)^{-\frac{1}{2}}\exp(-\frac{\|z_i - u_i\|^2}{2})dz_i$$
$$- (\int_{-\infty}^{\infty}\phi'(\sigma \cdot z_i)(2\pi)^{-\frac{1}{2}}\exp(-\frac{\|z_i - u_i\|^2}{2})dz_i)^2\big)$$
$$= (\frac{u_j^2}{\sigma^2} + 1)\big(\int_{-\infty}^{\infty}\phi'^2(\sigma \cdot z_i)(2\pi)^{-\frac{1}{2}}\exp(-\frac{\|z_i\|^2}{2})dz_i - (\int_{-\infty}^{\infty}\phi'(\sigma \cdot z_i)(2\pi)^{-\frac{1}{2}}\exp(-\frac{\|z_i\|^2}{2})dz_i)^2\big), \tag{27}$$

where the first step is by Definition 3, and the second step comes from the limit laws. Similarly, we also have

$$\lim_{u_i \to 0}\big(\beta_2(i, \boldsymbol{u}, \sigma) - \frac{1}{u_i^2 + 1}\alpha_2^2(i, \boldsymbol{u}, \sigma)\big)$$
$$= \lim_{u_i \to 0}\int_{-\infty}^{\infty}\phi'^2(\sigma \cdot z_i)z_i^2(2\pi)^{-\frac{1}{2}}\exp(-\frac{\|z_i - u_i\|^2}{2})dz_i$$
$$- (\frac{1}{u_i^2 + 1}\int_{-\infty}^{\infty}\phi'(\sigma \cdot z_i)z_i^2(2\pi)^{-\frac{1}{2}}\exp(-\frac{\|z_i - u_i\|^2}{2})dz_i)^2$$
$$= \int_{-\infty}^{\infty}\phi'^2(\sigma \cdot z_i)z_i^2(2\pi)^{-\frac{1}{2}}\exp(-\frac{\|z_i\|^2}{2})dz_i - (\int_{-\infty}^{\infty}\phi'(\sigma \cdot z_i)z_i^2(2\pi)^{-\frac{1}{2}}\exp(-\frac{\|z_i\|^2}{2})dz_i)^2 \tag{28}$$

Since that (27) and (28) are positive due to Jensen's inequality, we can derive that $\rho(\boldsymbol{u}, \sigma)$ converges to a positive value function of $\sigma$ as $u_i$ goes to 0, i.e.

$$\lim_{u \to 0}\rho(\boldsymbol{u}, \sigma) := \mathcal{C}_m(\sigma) \tag{29}$$

(3) When all $u_i \neq 0$ $(i \in [d])$,

$$\lim_{\sigma \to 0} \left( \beta_2(i, \frac{\boldsymbol{u}}{\sigma}, \sigma) - \frac{1}{\frac{u_i^2}{\sigma^2} + 1} \alpha_2^2(i, \frac{\boldsymbol{u}}{\sigma}, \sigma) \right)$$

$$= \lim_{\sigma \to 0} \int_{-\infty}^{\infty} \phi'^2(\sigma \cdot z_i) z_i^2 (2\pi)^{-\frac{1}{2}} \exp(-\frac{\|z_i - \frac{u_i}{\sigma}\|^2}{2}) dz_i$$

$$- \frac{1}{\frac{u_i^2}{\sigma^2} + 1} \left( \int_{-\infty}^{\infty} \phi'(\sigma \cdot z_i) z_i^2 (2\pi)^{-\frac{1}{2}} \exp(-\frac{\|z_i - \frac{u_i}{\sigma}\|^2}{2}) dz_i \right)^2$$

$$= \lim_{\sigma \to 0} \int_{-\infty}^{\infty} \phi'^2(u_i \cdot x_i) \frac{u_i^2}{\sigma^2} x_i^2 (2\pi \frac{\sigma^2}{u_i^2})^{-\frac{1}{2}} \exp(-\frac{\|x_i - 1\|^2}{2\frac{\sigma^2}{u_i^2}}) dx_i$$

$$- \frac{1}{\frac{u_i^2}{\sigma^2} + 1} \left( \int_{-\infty}^{\infty} \phi'(u_i \cdot x_i) \frac{u_i^2}{\sigma^2} x_i^2 (2\pi \frac{\sigma^2}{u_i^2})^{-\frac{1}{2}} \exp(-\frac{\|x_i - 1\|^2}{2\frac{\sigma^2}{u_i^2}}) dx_i \right)^2 \qquad z_i = \frac{u_i}{\sigma} x_i \qquad (30)$$

$$= \lim_{\sigma \to 0} \phi'^2(u_i) \frac{u_i^2}{\sigma^2} - \frac{1}{\frac{u_i^2}{\sigma^2} + 1} (\phi'(u_i) \frac{u_i^2}{\sigma^2})^2$$

$$= \lim_{\sigma \to 0} \phi'^2(u_i) \frac{u_i^2}{\sigma^2} \left( 1 - \frac{\frac{u_i^2}{\sigma^2}}{1 + \frac{u_i^2}{\sigma^2}} \right)$$

$$= \lim_{\sigma \to 0} \phi'^2(u_i) \frac{1}{1 + \frac{\sigma^2}{u_i^2}}$$

$$= \phi'^2(u_i)$$

The first step of (30) comes from Definition 3. The second step and the last three steps are derived from some basic mathematical computation and the limit laws. The third step of (30) is by the fact that the Gaussian distribution goes to a Dirac delta function when $\sigma$ goes to 0. Then the integral will take the value when $x_i = 1$. Similarly, we can obtain the following

$$\lim_{\sigma \to 0} \left( \beta_0(i, \frac{\boldsymbol{u}}{\sigma}, \sigma) - \alpha_0^2(i, \frac{\boldsymbol{u}}{\sigma}, \sigma) \right)$$

$$= \lim_{\sigma \to 0} \int_{-\infty}^{\infty} \phi'^2(\sigma \cdot z_i)(2\pi)^{-\frac{1}{2}} \exp(-\frac{\|z_i - \frac{u_i}{\sigma}\|^2}{2}) dz_i \qquad (31)$$

$$- \left( \int_{-\infty}^{\infty} \phi'(\sigma \cdot z_i)(2\pi)^{-\frac{1}{2}} \exp(-\frac{\|z_i - \frac{u_i}{\sigma}\|^2}{2}) dz_i \right)^2$$

$$= \phi'^2(u_i) - \phi'^2(u_i) = 0$$

$$
\begin{aligned}
&\lim_{\sigma \to 0} \Big( \frac{\partial}{\partial \sigma} \big( \beta_0(i, \frac{\boldsymbol{u}}{\sigma}, \sigma) - \alpha_0^2(i, \frac{\boldsymbol{u}}{\sigma}, \sigma) \big) \Big) \\
&= \lim_{\sigma \to 0} \Big( \frac{\partial}{\partial \sigma} \Big( \int_{-\infty}^{\infty} \phi'^2(x_i)(2\pi\sigma^2)^{-\frac{1}{2}} \exp(-\frac{\|x_i - u_i\|^2}{2\sigma^2}) dx_i \\
&\qquad - \big( \int_{-\infty}^{\infty} \phi'(x_i)(2\pi\sigma^2)^{-\frac{1}{2}} \exp(-\frac{\|x_i - u_i\|^2}{2\sigma^2}) dx_i \big)^2 \Big) \Big) \qquad x_i = \sigma \cdot z_i \\
&= \lim_{\sigma \to 0} \Big( \int_{-\infty}^{\infty} \phi'^2(x_i)(2\pi\sigma^2)^{-\frac{1}{2}} \exp(-\frac{\|x_i - u_i\|^2}{2\sigma^2})(-\sigma^{-1} + \|x_i - u_i\|^2 \sigma^{-2}) dx_i \\
&\qquad - 2 \big( \int_{-\infty}^{\infty} \phi'(x_i)(2\pi\sigma^2)^{-\frac{1}{2}} \exp(-\frac{\|x_i - u_i\|^2}{2\sigma^2}) dx_i \big) \\
&\qquad \cdot \int_{-\infty}^{\infty} \phi'(x_i)(2\pi\sigma^2)^{-\frac{1}{2}} \exp(-\frac{\|x_i - u_i\|^2}{2\sigma^2})(-\sigma^{-1} + \|x_i - u_i\|^2 \sigma^{-2}) dx_i \Big) \\
&= \lim_{\sigma \to 0} \Big( \frac{\phi'^2(u_i)}{-\sigma} - 2\phi'(u_i)\frac{\phi'(u_i)}{-\sigma} \Big) \\
&= \lim_{\sigma \to 0} \frac{\phi'^2(u_i)}{\sigma} = +\infty
\end{aligned}
\tag{32}
$$

Therefore, by L'Hopital's rule and (31), (32), we have

$$
\begin{aligned}
&\lim_{\sigma \to 0} (\frac{u_j^2}{\sigma^2} + 1)(\beta_0(i, \frac{\boldsymbol{u}}{\sigma}, \sigma) - \alpha_0(i, \frac{\boldsymbol{u}}{\sigma}, \sigma)) \\
&= \lim_{\sigma \to 0} \frac{u_i^2}{2\sigma} \frac{\partial}{\partial \sigma}(\beta_0(i, \frac{\boldsymbol{u}}{\sigma}, \sigma) - \alpha_0(i, \frac{\boldsymbol{u}}{\sigma}, \sigma)) \\
&= +\infty
\end{aligned}
\tag{33}
$$

Combining (33) and (30), we can derive that $\rho(\frac{\boldsymbol{u}}{\sigma}, \sigma)$ converges to a positive value function of $\boldsymbol{u}$ as $\sigma$ goes to 0, i.e.

$$
\lim_{\sigma \to 0} \rho(\frac{\boldsymbol{u}}{\sigma}, \sigma) := \mathcal{C}_s(\boldsymbol{u}).
\tag{34}
$$

When $u_i = 0$ for some $i \in [d]$, $\lim_{\sigma \to 0}(\frac{u_i^2}{\sigma^2} + 1)(\beta_0(j, \frac{\boldsymbol{u}}{\sigma}, \sigma) - \alpha^2(j, \frac{\boldsymbol{u}}{\sigma}, \sigma)) = 0$ by (31). Then from the Definition 3, we have

$$
\lim_{\sigma \to 0} \rho(\frac{\boldsymbol{u}}{\sigma}, \sigma) = 0
\tag{35}
$$

(4) We can define $\mathcal{L}_m(\frac{(\boldsymbol{\Lambda W}^*)^\top \boldsymbol{\Lambda u}}{\sigma \delta_K(\boldsymbol{W}^*)}, \sigma \delta_K(\boldsymbol{W}^*))$ as

$$
\mathcal{L}_m(\frac{(\boldsymbol{\Lambda W}^*)^\top \boldsymbol{\Lambda u}}{\sigma \delta_K(\boldsymbol{W}^*)}, \sigma \delta_K(\boldsymbol{W}^*)) = \min_{v_i \in [0, u_i]} \Big\{ \rho(\frac{(\boldsymbol{\Lambda W}^*)^\top \boldsymbol{\Lambda v}}{\sigma_l \delta_K(\boldsymbol{W}^*)}, \sigma \delta_K(\boldsymbol{W}^*)) : v_j = u_j \text{ for all } j \neq i \Big\}
\tag{36}
$$

Then by this definition, we have

$$
0 < \mathcal{L}_m(\frac{(\boldsymbol{\Lambda W}^*)^\top \boldsymbol{\Lambda u}}{\sigma \delta_K(\boldsymbol{W}^*)}, \sigma \delta_K(\boldsymbol{W}^*)) \leq \rho(\frac{(\boldsymbol{\Lambda W}^*)^\top \boldsymbol{\Lambda u}}{\sigma_l \delta_K(\boldsymbol{W}^*)}, \sigma \delta_K(\boldsymbol{W}^*))
\tag{37}
$$

Meanwhile, for any $0 \leq u_i' \leq u_i^*$, since that $[0, u_i'] \subset [0, u_i^*]$, we can obtain

$$
\mathcal{L}_m(\frac{(\boldsymbol{\Lambda W}^*)^\top \boldsymbol{\Lambda u}}{\sigma \delta_K(\boldsymbol{W}^*)}, \sigma \delta_K(\boldsymbol{W}^*))|_{u_i = u_i'} \geq \mathcal{L}_m(\frac{(\boldsymbol{\Lambda W}^*)^\top \boldsymbol{\Lambda u}}{\sigma \delta_K(\boldsymbol{W}^*)}, \sigma \delta_K(\boldsymbol{W}^*))|_{u_i = u_i^*}
\tag{38}
$$

Hence, $\mathcal{L}_m(\frac{(\boldsymbol{\Lambda W}^*)^\top \boldsymbol{\Lambda u}}{\sigma \delta_K(\boldsymbol{W}^*)}, \sigma \delta_K(\boldsymbol{W}^*))$ is a strictly positive real function which is monotonically decreasing.

(5) Therefore, we only need to show the condition (a).
When $(\boldsymbol{W}^{*\top} \boldsymbol{u})_i \neq 0$ for all $i \in [K]$,

$$
\lim_{\sigma \to 0} \rho(\frac{\boldsymbol{W}^{*\top} \boldsymbol{u}}{\sigma \delta_K(\boldsymbol{W}^*)}, \sigma \delta_K(\boldsymbol{W}^*)) = \mathcal{C}_s(\boldsymbol{u}) > 0.
\tag{39}
$$

Therefore, there exists $\zeta_s > 0$, such that when $0 < \sigma < \zeta_s$,

$$\rho(\frac{\boldsymbol{W}^{*\top}\boldsymbol{u}}{\sigma\delta_K(\boldsymbol{W}^*)}, \sigma\delta_K(\boldsymbol{W}^*)) > \frac{\mathcal{C}_s(\boldsymbol{W}^{*\top}\boldsymbol{u})}{2}. \tag{40}$$

Then we can define

$$\mathcal{L}_s(\frac{\boldsymbol{W}^{*\top}\boldsymbol{u}}{\sigma\delta_K(\boldsymbol{W}^*)}, \sigma\delta_K(\boldsymbol{W}^*)) := \frac{\mathcal{C}_s(\boldsymbol{W}^{*\top}\boldsymbol{u})}{2\zeta_s}\sigma^2 \tag{41}$$

such that $\sigma^{-1}\mathcal{L}_s(\frac{\boldsymbol{W}^{*\top}\boldsymbol{u}}{\sigma\delta_K(\boldsymbol{W}^*)}, \sigma\delta_K(\boldsymbol{W}^*))$ is an increasing function of $\sigma$ below $\rho(\frac{\boldsymbol{W}^{*\top}\boldsymbol{u}}{\sigma\delta_K(\boldsymbol{W}^*)}, \sigma\delta_K(\boldsymbol{W}^*))$.

When $(\boldsymbol{W}^{*\top}\boldsymbol{u})_i = 0$ for some $i \in [K]$, then

$$\lim_{\sigma\to0} \rho(\frac{\boldsymbol{W}^{*\top}\boldsymbol{u}}{\sigma\delta_K(\boldsymbol{W}^*)}, \sigma\delta_K(\boldsymbol{W}^*)) = 0. \tag{42}$$

We can define

$$\mathcal{L}_s\Big(\frac{\boldsymbol{W}^{*\top}\boldsymbol{u}}{\sigma\delta_K(\boldsymbol{W}^*)}, \sigma\delta_K(\boldsymbol{W}^*)\Big) = \sigma \cdot \min_{v_i\in[u_i,\zeta_{s'}]} \Big\{\rho(\frac{\boldsymbol{W}^{*\top}\boldsymbol{v}}{\sigma\delta_K(\boldsymbol{W}^*)}, \sigma\delta_K(\boldsymbol{W}^*)) : v_j \neq u_j \text{ for all } j \neq i\Big\} \tag{43}$$

Then,

$$\sigma^{-1}\mathcal{L}_s\Big(\frac{\boldsymbol{W}^{*\top}\boldsymbol{u}}{\sigma\delta_K(\boldsymbol{W}^*)}, \sigma\delta_K(\boldsymbol{W}^*)\Big) = \min_{v_i\in[u_i,\zeta_{s'}]} \Big\{\rho(\frac{\boldsymbol{W}^{*\top}\boldsymbol{v}}{\sigma\delta_K(\boldsymbol{W}^*)}, \sigma\delta_K(\boldsymbol{W}^*)) : v_j = u_j \text{ for all } j \neq i\Big\} \tag{44}$$

For any $0 \leq u'_i \leq u^*_i < \zeta_{s'}$, since that $[u^*_i, \zeta_{s'}] \subset [u'_i, \zeta_{s'}]$, we can obtain

$$\sigma^{-1}\mathcal{L}_s(\frac{\boldsymbol{W}^{*\top}\boldsymbol{u}}{\sigma\delta_K(\boldsymbol{W}^*)}, \sigma\delta_K(\boldsymbol{W}^*))|_{u_i=u'_i} \leq \sigma^{-1}\mathcal{L}_s(\frac{\boldsymbol{W}^{*\top}\boldsymbol{u}}{\sigma\delta_K(\boldsymbol{W}^*)}, \sigma\delta_K(\boldsymbol{W}^*))|_{u_i=u^*_i} \tag{45}$$

Therefore, we can derive that $\sigma^{-1}\mathcal{L}_s(\frac{\boldsymbol{W}^{*\top}\boldsymbol{u}}{\sigma\delta_K(\boldsymbol{W}^*)}, \sigma\delta_K(\boldsymbol{W}^*))$ is monotonically increasing. Following the steps in (4), we can have that $\sigma^{-1}\mathcal{L}_s(\frac{\boldsymbol{W}^{*\top}\boldsymbol{u}}{\sigma\delta_K(\boldsymbol{W}^*)}, \sigma\delta_K(\boldsymbol{W}^*))$ is a strictly positive real function which is upper bounded by $\rho(\frac{\boldsymbol{W}^{*\top}\boldsymbol{u}}{\sigma\delta_K(\boldsymbol{W}^*)}, \sigma\delta_K(\boldsymbol{W}^*))$.

In conclusion, condition (a) is proved.

For condition (b), since that $\zeta_s > 0$, $\rho(\frac{\boldsymbol{W}^{*\top}\boldsymbol{u}}{\sigma_l\delta_K(\boldsymbol{W}^*)}, \sigma\delta_K(\boldsymbol{W}^*))$ is continuous and positive, we can obtain

$$\rho(\frac{\boldsymbol{W}^{*\top}\boldsymbol{v}}{\sigma\delta_K(\boldsymbol{W}^*)}, \sigma\delta_K(\boldsymbol{W}^*))\Big|_{\sigma=\zeta_s} > 0 \tag{46}$$

Then condition (b) can be easily proved as in (4).

We then characterize the order of the $\rho$ function in different cases as follows.

**Property 4.** *To specify the order with regard to the distribution parameters, $\rho(\boldsymbol{u}, \sigma)$ in Definition 3 satisfies the following properties,*

    1. *(**Small variance**) $\lim_{\sigma\to0^+} \rho(\boldsymbol{u}, \sigma) = \Theta(\sigma^4)$.*

    2. *(**Large variance**) For any $\epsilon > 0$, $\lim_{\sigma\to\infty} \rho(\boldsymbol{u}, \sigma) \geq \Theta(\frac{1}{\sigma^{3+\epsilon}})$.*

    3. *(**Large mean**) For any $\epsilon > 0$, $\lim_{\mu\to\infty} \rho(\boldsymbol{u}, \sigma) \geq \Theta(e^{-\frac{\|\boldsymbol{u}\|^2}{2}})\frac{1}{\|\boldsymbol{u}\|^{3+\epsilon}}$.*

**Proof:**

(1)

$$\beta_0(i, \boldsymbol{u}, \sigma) - \alpha_0(i, \boldsymbol{u}, \sigma)^2$$

$$=\mathbb{E}_{z\sim\mathcal{N}(\mu,1)}[\phi'^2(\sigma \cdot z)] - (\mathbb{E}_{z\sim\mathcal{N}(\mu,1)}[\phi'(\sigma \cdot z)])^2$$

$$=\int_{-\infty}^{\infty} \phi'^2(\sigma \cdot z)\frac{1}{\sqrt{2\pi}}e^{-\frac{(z-\mu)^2}{2}}dz - (\int_{-\infty}^{\infty} \phi'(\sigma \cdot z)\frac{1}{\sqrt{2\pi}}e^{-\frac{(z-\mu)^2}{2}}dz)^2$$

$$=\int_{-\infty}^{\infty} (\frac{1}{4} - \frac{t^2}{16} + \frac{t^4}{96}\cdots)^2\frac{1}{\sqrt{2\pi}\sigma}e^{-\frac{(t-\mu\sigma)^2}{2\sigma^2}}dt$$

$$\qquad - (\int_{-\infty}^{\infty} (\frac{1}{4} - \frac{t^2}{16} + \frac{t^4}{96} + \cdots)\frac{1}{\sqrt{2\pi}\sigma}e^{-\frac{(t-\mu\sigma)^2}{2\sigma^2}}dt)^2 \qquad (47)$$

$$=(\frac{1}{16} - \frac{1}{32}(\mu^2\sigma^2 + \sigma^2) + \frac{7}{768}(3\sigma^4 + 6\mu^2\sigma^4 + \mu^4\sigma^4) + \cdots)$$

$$\qquad - (\frac{1}{4} - \frac{\mu^2\sigma^2 + \sigma^2}{16} + \frac{3\sigma^4 + 6\mu^2\sigma^4 + \mu^4\sigma^4}{192} + \cdots)^2$$

$$=\frac{1}{128}\sigma^4 + \frac{\mu^2\sigma^4}{64} + o(\sigma^4), \quad \text{as } \sigma \to 0^+.$$

The first step of (47) is by Definition 3. The second step and the last steps come from some basic mathematical computation. The third step is from Taylor expansion. Hence,

$$\lim_{\sigma\to 0^+} (\beta_0(i, \boldsymbol{u}, \sigma) - \alpha_0(i, \boldsymbol{u}, \sigma)^2) = \frac{1}{128}\sigma^4 + \frac{\mu^2\sigma^4}{64} + o(\sigma^4) \qquad (48)$$

Similarly, we can obtain

$$\beta_2(i, \boldsymbol{u}, \sigma) - \frac{\alpha_2(i, \boldsymbol{u}, \sigma)^2}{\mu^2 + 1}$$

$$=\mathbb{E}_{z\sim\mathcal{N}(0,1)}[\phi'^2(\sigma \cdot z)z^2] - \frac{(\mathbb{E}_{z\sim\mathcal{N}(0,1)}[\phi'(\sigma \cdot z)z^2])^2}{\mu^2 + 1}$$

$$=\int_{-\infty}^{\infty} \phi'^2(\sigma \cdot z)z^2\frac{1}{\sqrt{2\pi}}e^{-\frac{(z-\mu)^2}{2}}dz - \frac{1}{\mu^2 + 1}(\int_{-\infty}^{\infty} \phi'(\sigma \cdot z)z^2\frac{1}{\sqrt{2\pi}}e^{-\frac{(z-\mu)^2}{2}}dz)^2$$

$$=\int_{-\infty}^{\infty} (\frac{t}{4\sigma} - \frac{t^3}{16\sigma} + \frac{t^5}{96\sigma}\cdots)^2\frac{1}{\sqrt{2\pi}\sigma}e^{-\frac{(t-\mu\sigma)^2}{2\sigma^2}}dt$$

$$\qquad - \frac{1}{\mu^2 + 1}(\int_{-\infty}^{\infty} (\frac{t^2}{4\sigma^2} - \frac{t^4}{16\sigma^2} + \frac{t^6}{96\sigma^2} + \cdots)\frac{1}{\sqrt{2\pi}\sigma}e^{-\frac{(t-\mu\sigma)^2}{2\sigma^2}}dt)^2 \qquad (49)$$

$$=(\frac{1 + \mu^2}{16} - \frac{3\sigma^2 + 6\mu^2\sigma^2 + \mu^4\sigma^2}{32} + \cdots)$$

$$\qquad - \frac{1}{\mu^2 + 1}(\frac{1 + \mu^2}{4} - \frac{15\sigma^2 + 45\mu^2\sigma^2 + 15\mu^4\sigma^2 + \mu^6\sigma^2}{32} + \cdots)^2$$

$$=\frac{9}{64}\sigma^2 + \frac{33}{64}\mu^2\sigma^2 + \frac{13}{64}\mu^4\sigma^2 + \frac{1}{64}\mu^6\sigma^2 + o(\sigma^2), \quad \text{as } \sigma \to 0^+$$

Hence,

$$\lim_{\sigma\to 0^+} (\beta_2(i, \boldsymbol{u}, \sigma) - \frac{\alpha_2(i, \boldsymbol{u}, \sigma)^2}{\mu^2 + 1}) = \frac{9}{64}\sigma^2 + o(\sigma^2) \qquad (50)$$

Therefore,

$$\lim_{\sigma\to 0^+} \rho(\boldsymbol{u}, \sigma) = \min_{j\in[d], u_j\neq\mu} \{(u_j^2 + 1)\}\frac{1}{128}\sigma^4 \qquad (51)$$

(2) Note that by some basic mathematical derivation,

$$
\begin{aligned}
\int_{-\infty}^{\infty} \phi'^2(\sigma \cdot z) \frac{1}{\sqrt{2\pi}} e^{-\frac{(z-\mu)^2}{2}} dz &= \int_{-\infty}^{\infty} \frac{1}{(e^{\sigma \cdot z} + e^{-\sigma \cdot z} + 2)^2} \frac{1}{\sqrt{2\pi}} e^{-\frac{(z-\mu)^2}{2}} dz \\
&\geq 2 \int_{0}^{\infty} \frac{1}{16 e^{2\sigma \cdot z}} \frac{1}{\sqrt{2\pi}} e^{-\frac{(z+|\mu|)^2}{2}} dz \\
&= \frac{1}{8} e^{2|\mu|\sigma + 2\sigma^2} \int_{0}^{\infty} \frac{1}{\sqrt{2\pi}} e^{-\frac{(z+2\sigma)^2}{2}} dz \\
&= \frac{1}{8\sqrt{2\pi}} e^{2|\mu|\sigma + 2\sigma^2} \int_{|\mu|+2\sigma}^{\infty} e^{-\frac{t^2}{2}} dt
\end{aligned}
\tag{52}
$$

We then provide the following Claim with its proof to give a lower bound for (52).

**Claim**: $\int_{|\mu|+2\sigma}^{\infty} e^{-\frac{t^2}{2}} dt > e^{-2|\mu|\sigma - 2\sigma^2 - k_1 \log \sigma}$ for $k_1 > 1$.

**Proof:** Let

$$
f(\sigma) = \int_{|\mu|+2\sigma}^{\infty} e^{-\frac{t^2}{2}} dt - e^{-2|\mu|\sigma - 2\sigma^2 - k_1 \log \sigma}.
\tag{53}
$$

Then,

$$
f'(\sigma) = e^{-2\sigma^2} \left( (2|\mu| + 4\sigma + \frac{k_1}{\sigma}) \sigma^{-k_1} - 2e^{-\frac{1}{2}\mu^2} \right).
\tag{54}
$$

It can be easily verified that for a given $|\mu| \geq 0$, $f'(\sigma) < 0$ when $\sigma$ is large enough if $k_1 > 1$. Combining that $\lim_{\sigma \to \infty} f(\sigma) = 0$, we have $f(\sigma) > 0$ when $\sigma$ is large enough by showing the contradiction in the following:

Suppose there is a strictly increasing function $f(x) > 0$ with $\lim_{x \to \infty} f(x) = 0$ when $x$ is large enough. Then there exists $x_0 > 0$ such that for any $\epsilon > 0$, $f(x) < \epsilon$ for $x > x_0$. Pick $\epsilon = f(x_0) > 0$, then for $x_1 > x_0$, $f(x_1) > f(x_0) = \epsilon$. Contradiction!

Similarly, we also have

$$
\begin{aligned}
\int_{-\infty}^{\infty} \phi'(\sigma \cdot z) \frac{1}{\sqrt{2\pi}} e^{-\frac{z^2}{2}} dz &= \int_{-\infty}^{\infty} \frac{1}{e^{\sigma \cdot z} + e^{-\sigma \cdot z} + 2} \frac{1}{\sqrt{2\pi}} e^{-\frac{(z-\mu)^2}{2}} dz \\
&\leq 2 \int_{0}^{\infty} \frac{1}{e^{\sigma \cdot z}} \frac{1}{\sqrt{2\pi}} e^{-\frac{(z-\mu)^2}{2}} dz \\
&= e^{|\mu|\sigma + \frac{1}{2}\sigma^2} \int_{0}^{\infty} \frac{2}{\sqrt{2\pi}} e^{-\frac{(z+|\mu|+\sigma)^2}{2}} dz \\
&= \frac{2}{\sqrt{2\pi}} e^{|\mu|\sigma + \frac{1}{2}\sigma^2} \int_{|\mu|+\sigma}^{\infty} e^{-\frac{t^2}{2}} dt,
\end{aligned}
\tag{55}
$$

and the **Claim**: $\int_{|\mu|+\sigma}^{\infty} e^{-\frac{t^2}{2}} dt < e^{-|\mu|\sigma - \frac{1}{2}\sigma^2 - k_2 \log \sigma}$ for $k_2 \leq 1$ to give an upper bound for (55).

Therefore, combining (52, 55) and two claims, we have that for any $\epsilon > 0$,

$$
\beta_0(i, \boldsymbol{u}, \sigma) - \alpha_0(i, \boldsymbol{u}, \sigma)^2 \geq \frac{1}{8\sqrt{2\pi}} \frac{1}{\sigma^{k_1}} - \frac{1}{2\pi} \frac{1}{\sigma^{2k_2}} \gtrsim \frac{1}{\sigma^{1+\epsilon}}
\tag{56}
$$

(The above inequality holds for any $2k_2 > k_1$ where $k_1 > 1$ and $k_2 \leq 1$.)

Similarly,

$$
\begin{aligned}
\int_{-\infty}^{\infty} \phi'^2(\sigma \cdot z) z^2 \frac{1}{\sqrt{2\pi}} e^{-\frac{z^2}{2}} dz &= \int_{-\infty}^{\infty} \frac{z^2}{(e^{\sigma \cdot z} + e^{-\sigma \cdot z} + 2)^2} \frac{1}{\sqrt{2\pi}} e^{-\frac{(z-\mu)^2}{2}} dz \\
&\geq 2 \int_{0}^{\infty} \frac{z^2}{16 e^{2\sigma \cdot z}} \frac{1}{\sqrt{2\pi}} e^{-\frac{(z+|\mu|)^2}{2}} dz \\
&= \frac{1}{8\sqrt{2\pi}} e^{|\mu|\sigma + 2\sigma^2} \int_{2|\mu|+2\sigma}^{\infty} (t - 2\sigma)^2 e^{-\frac{t^2}{2}} dt
\end{aligned}
\tag{57}
$$

**Claim**: $\int_{|\mu|+2\sigma}^{\infty} (t - 2\sigma)^2 e^{-\frac{t^2}{2}} dt \geq e^{-2|\mu|\sigma - 2\sigma^2 - k_1 \log \sigma}$ if $k_1 > 3$.

**Proof**: Let

$$
f(\sigma) = \int_{|\mu|+2\sigma}^{\infty} (t - 2\sigma)^2 e^{-\frac{t^2}{2}} dt - e^{-2|\mu|\sigma - 2\sigma^2 - k_1 \log \sigma}.
\tag{58}
$$

$$f'(\sigma) = 8\sigma \int_{|\mu|+2\sigma}^{\infty} e^{-\frac{t^2}{2}} dt + e^{-2|\mu|\sigma - 2\sigma^2}(4\sigma^{1-k_1} + k_1\sigma^{-1-k_1} + 2|\mu|\sigma^{-k_1} - 4e^{-\frac{1}{2}\mu^2}). \quad (59)$$

We need $f'(\sigma) < 0$ when $\sigma$ is large enough. Since that $f'(\sigma) \to 0$, $f''(\sigma) \to 0$ when $\sigma$ is large, we need $f''(\sigma) > 0$ and $f'''(\sigma) < 0$ recursively. Hence,

$$\begin{aligned}
f'''(\sigma) =& e^{-2|\mu|\sigma - 2\sigma^2}(64\sigma^{3-k_1} + 96\mu\sigma^{2-k_1} + 16(3k_1 - 3 + \mu^2)\sigma^{1-k_1} + 8\mu(-\mu^2 - 3 + 6k_1)\sigma^{-k_1} \\
& + 4k_1(3k_1 + \mu^2)\sigma^{-1-k_1} + 2k_1(1+k_1)(\mu+2)\sigma^{-2-k_1} \\
& + k_1(1+k_1)(2+k_1)\sigma^{-3-k_1} - 16e^{-\frac{1}{2}\mu^2}) < 0
\end{aligned}$$

$$\tag{60}$$

requires $k_1 > 3$.
Similarly, we have

$$\int_{-\infty}^{\infty} \phi'(\sigma \cdot z)z^2 \frac{1}{\sqrt{2\pi}} e^{-\frac{z^2}{2}} dz \leq 2 \int_0^{\infty} \frac{1}{e^{\sigma \cdot z}} \frac{1}{\sqrt{2\pi}} z^2 e^{-\frac{z^2}{2}} dz = \frac{2}{\sqrt{2\pi}} e^{\frac{1}{2}\sigma^2} \int_{\sigma}^{\infty} (t-\sigma)^2 e^{-\frac{t^2}{2}} dt \quad (61)$$

and the **Claim**: $\int_{\sigma}^{\infty} (t-\sigma)^2 e^{-\frac{t^2}{2}} dt < e^{-\frac{\sigma^2}{2} - k_2 \log \sigma}$. Hence,

$$\beta_2(i, \boldsymbol{u}, \sigma) - \frac{\alpha_2(i, \boldsymbol{u}, \sigma)^2}{\mu^2 + 1} \geq \frac{1}{8\sqrt{2\pi}} \frac{1}{\sigma^{k_1}} - \frac{2}{\pi(\mu^2+1)} \frac{1}{\sigma^{2k_2}} \gtrsim \frac{1}{\sigma^{3.1}} \quad (62)$$

(The above inequality holds for any $2k_2 > k_1$ where $k_1 > 3$ and $k_2 < 3$.)
Therefore, by combining (56) and (62), for any $\epsilon > 0$

$$\lim_{\sigma \to \infty} \rho(\boldsymbol{u}, \sigma) \geq \Theta(\frac{1}{\sigma^{3+\epsilon}}). \quad (63)$$

(3) Let $\sigma$ be fixed. For any $\epsilon > 0$, following the steps in (2), we can obtain

$$\begin{aligned}
\int_{-\infty}^{\infty} \phi'^2(\sigma \cdot z) \frac{1}{\sqrt{2\pi}} e^{-\frac{(z-\mu)^2}{2}} dz &= \int_{-\infty}^{\infty} \frac{1}{(e^{\sigma \cdot z} + e^{-\sigma \cdot z} + 2)^2} \frac{1}{\sqrt{2\pi}} e^{-\frac{(z-\mu)^2}{2}} dz \\
&\geq 2 \int_0^{\infty} \frac{1}{16e^{2\sigma \cdot z}} \frac{1}{\sqrt{2\pi}} e^{-\frac{(z+|\mu|)^2}{2}} dz \\
&= \frac{1}{8\sqrt{2\pi}} e^{2|\mu|\sigma + 2\sigma^2} \int_{|\mu|+2\sigma}^{\infty} e^{-\frac{t^2}{2}} dt \\
&\geq \frac{1}{8\sqrt{2\pi}} e^{-\frac{\mu^2}{2}} \frac{1}{\mu^{1+\epsilon}}
\end{aligned} \quad (64)$$

$$\begin{aligned}
\int_{-\infty}^{\infty} \phi'(\sigma \cdot z) \frac{1}{\sqrt{2\pi}} e^{-\frac{(z-\mu)^2}{2}} dz &= \int_{-\infty}^{\infty} \frac{1}{e^{\sigma \cdot z} + e^{-\sigma \cdot z} + 2} \frac{1}{\sqrt{2\pi}} e^{-\frac{(z-\mu)^2}{2}} dz \\
&\leq 2 \int_0^{\infty} \frac{1}{e^{\sigma \cdot z}} \frac{1}{\sqrt{2\pi}} e^{-\frac{(z-\mu)^2}{2}} dz \\
&= \frac{2}{\sqrt{2\pi}} e^{-\frac{\mu^2}{2}} \frac{1}{\mu^{1-\epsilon}}
\end{aligned} \quad (65)$$

Similarly,

$$\int_{-\infty}^{\infty} \phi'^2(\sigma \cdot z) z^2 \frac{1}{\sqrt{2\pi}} e^{-\frac{(z-\mu)^2}{2}} dz \geq \frac{1}{8\sqrt{2\pi}} e^{-\frac{\mu^2}{2}} \frac{1}{\mu^{3+\epsilon}} \quad (66)$$

$$\int_{-\infty}^{\infty} \phi'(\sigma \cdot z) z^2 \frac{1}{\sqrt{2\pi}} e^{-\frac{(z-\mu)^2}{2}} dz \leq \frac{2}{\sqrt{2\pi}} e^{-\frac{\mu^2}{2}} \frac{1}{\mu^{3-\epsilon}} \quad (67)$$

We can conclude that $\lim_{\mu \to \infty} \rho(\boldsymbol{u}, \sigma) \geq \Theta(e^{-\frac{\|\boldsymbol{u}\|^2}{2}}) \frac{1}{\|\boldsymbol{u}\|^{3+\epsilon}}$.

**Property 5.** *If a function $f(\boldsymbol{x})$ is an even function, then*

$$\mathbb{E}_{\boldsymbol{x} \sim \mathcal{N}(\boldsymbol{\mu}, \boldsymbol{\Sigma})}[f(\boldsymbol{x})] = \mathbb{E}_{\boldsymbol{x} \sim \frac{1}{2}\mathcal{N}(\boldsymbol{\mu}, \boldsymbol{\Sigma}) + \frac{1}{2}\mathcal{N}(-\boldsymbol{\mu}, \boldsymbol{\Sigma})}[f(\boldsymbol{x})] \quad (68)$$

**Proof:**

Denote

$$g(\boldsymbol{x}) = f(\boldsymbol{x})(2\pi|\boldsymbol{\Sigma}|^2)^{-\frac{d}{2}}\exp(-\frac{1}{2}(\boldsymbol{x}-\boldsymbol{\mu})\boldsymbol{\Sigma}^{-1}(\boldsymbol{x}-\boldsymbol{\mu})) \tag{69}$$

By some basic mathematical computation,

$$
\begin{aligned}
\mathbb{E}_{\boldsymbol{x}\sim\mathcal{N}(\boldsymbol{\mu},\boldsymbol{\Sigma})}[f(\boldsymbol{x})] &= \int_{\boldsymbol{x}\in\mathbb{R}^d} g(\boldsymbol{x})d\boldsymbol{x} = \int_{-\infty}^{\infty}\cdots\int_{-\infty}^{\infty} g(x_1,\cdots,x_d)dx_1\cdots dx_d \\
&= \int_{-\infty}^{\infty}\cdots\int_{-\infty}^{\infty}\int_{\infty}^{-\infty} g(x_1,x_2,\cdots,x_d)d(-x_1)dx_2\cdots dx_d \\
&= \int_{-\infty}^{\infty}\cdots\int_{-\infty}^{\infty} g(-x_1,x_2\cdots,x_d)dx_1 dx_2\cdots dx_d \\
&= \int_{\boldsymbol{x}\in\mathbb{R}^d} g(-\boldsymbol{x})d\boldsymbol{x} \\
&= \int_{\boldsymbol{x}\in\mathbb{R}^d} f(\boldsymbol{x})(2\pi|\boldsymbol{\Sigma}|^2)^{-\frac{d}{2}}\exp(-\frac{1}{2}(\boldsymbol{x}+\boldsymbol{\mu})\boldsymbol{\Sigma}^{-1}(\boldsymbol{x}+\boldsymbol{\mu})) \\
&= \mathbb{E}_{\boldsymbol{x}\sim\mathcal{N}(-\boldsymbol{\mu},\boldsymbol{\Sigma})}[f(\boldsymbol{x})]
\end{aligned}
\tag{70}
$$

Therefore, we have

$$\mathbb{E}_{\boldsymbol{x}\sim\mathcal{N}(\boldsymbol{\mu},\boldsymbol{\Sigma})}[f(\boldsymbol{x})] = \mathbb{E}_{\boldsymbol{x}\sim\frac{1}{2}\mathcal{N}(\boldsymbol{\mu},\boldsymbol{\Sigma})+\frac{1}{2}\mathcal{N}(-\boldsymbol{\mu},\boldsymbol{\Sigma})}[f(\boldsymbol{x})] \tag{71}$$

**Property 6.** *Under Gaussian Mixture Model* $\boldsymbol{x} \sim \sum_{l=1}^{L}\lambda_l\mathcal{N}(\boldsymbol{\mu}_l,\boldsymbol{\Sigma}_l)$ *where* $\boldsymbol{\Sigma}_l = diag(\sigma_{l1}^2,\cdots,\sigma_{ld}^2)$, *we have the following upper bound.*

$$\mathbb{E}_{\boldsymbol{x}\sim\sum_{l=1}^{L}\lambda_l\mathcal{N}(\boldsymbol{\mu}_l,\boldsymbol{\Sigma}_l)}[(\boldsymbol{u}^\top\boldsymbol{x})^{2t}] \le (2t-1)!!||\boldsymbol{u}||^{2t}\sum_{l=1}^{L}\lambda_l(||\boldsymbol{\mu}_l||+||\boldsymbol{\Sigma}_l^{\frac{1}{2}}||)^{2t} \tag{72}$$

**Proof:**

Note that

$$\mathbb{E}_{\boldsymbol{x}\sim\sum_{l=1}^{L}\lambda_l\mathcal{N}(\boldsymbol{\mu}_l,\boldsymbol{\Sigma}_l)}[(\boldsymbol{u}^\top\boldsymbol{x})^{2t}] = \sum_{l=1}^{L}\lambda_l\mathbb{E}_{\boldsymbol{x}\sim\mathcal{N}(\boldsymbol{\mu}_l,\boldsymbol{\Sigma}_l)}[(\boldsymbol{u}^\top\boldsymbol{x})^{2t}] = \sum_{l=1}^{L}\lambda_l\mathbb{E}_{y\sim\mathcal{N}(\boldsymbol{u}^\top\boldsymbol{\mu}_l,\boldsymbol{u}^\top\boldsymbol{\Sigma}_l\boldsymbol{u})}[y^{2t}], \tag{73}$$

where the last step is by that $\boldsymbol{u}^\top\boldsymbol{x} \sim \mathcal{N}(\boldsymbol{u}^\top\boldsymbol{\mu},\boldsymbol{u}^\top\boldsymbol{\Sigma}_l\boldsymbol{u})$ for $\boldsymbol{x} \sim \mathcal{N}(\boldsymbol{\mu}_l,\boldsymbol{\Sigma}_l)$. By some basic mathematical computation, we know

$$
\begin{aligned}
&\mathbb{E}_{y\sim\mathcal{N}(\boldsymbol{u}^\top\boldsymbol{\mu}_l,\boldsymbol{u}^\top\boldsymbol{\Sigma}_l\boldsymbol{u})}[y^{2t}] \\
&= \int_{-\infty}^{\infty}(y-\boldsymbol{u}^\top\boldsymbol{\mu}_l+\boldsymbol{u}^\top\boldsymbol{\mu}_l)^{2t}\frac{1}{\sqrt{2\pi\boldsymbol{u}^\top\boldsymbol{\Sigma}_l\boldsymbol{u}}}e^{-\frac{(y-\boldsymbol{u}^\top\boldsymbol{\mu}_l)^2}{2\boldsymbol{u}^\top\boldsymbol{\Sigma}_l\boldsymbol{u}}}dy \\
&= \int_{-\infty}^{\infty}\sum_{p=0}^{2t}\binom{2t}{p}(\boldsymbol{u}^\top\boldsymbol{\mu}_l)^{2t-p}(y-\boldsymbol{u}^\top\boldsymbol{\mu}_l)^p\frac{1}{\sqrt{2\pi\boldsymbol{u}^\top\boldsymbol{\Sigma}_l\boldsymbol{u}}}e^{-\frac{(y-\boldsymbol{u}^\top\boldsymbol{\mu}_l)^2}{2\boldsymbol{u}^\top\boldsymbol{\Sigma}_l\boldsymbol{u}}}dy \\
&= \sum_{p=0}^{2t}\binom{2t}{p}(\boldsymbol{u}^\top\boldsymbol{\mu}_l)^{2t-p}\cdot\begin{cases} 0, & p\text{ is odd} \\ (p-1)!!(\boldsymbol{u}^\top\boldsymbol{\Sigma}_l\boldsymbol{u})^{\frac{p}{2}}, & p\text{ is even} \end{cases} \\
&\le \sum_{p=0}^{2t}\binom{2t}{p}|\boldsymbol{u}^\top\boldsymbol{\mu}_l|^{2t-p}(p-1)!!|\boldsymbol{u}^\top\boldsymbol{\Sigma}_l\boldsymbol{u}|^{\frac{p}{2}} \\
&\le (2t-1)!!(|\boldsymbol{u}^\top\boldsymbol{\mu}_l|+|\boldsymbol{u}^\top\boldsymbol{\Sigma}_l\boldsymbol{u}|^{\frac{1}{2}})^{2t} \\
&\le (2t-1)!!||\boldsymbol{u}||^{2t}(||\boldsymbol{\mu}_l||+||\boldsymbol{\Sigma}||^{\frac{1}{2}})^{2t},
\end{aligned}
\tag{74}
$$

where the second step is by the Binomial theorem. Hence,

$$\mathbb{E}_{\boldsymbol{x}\sim\sum_{l=1}^{L}\lambda_l\mathcal{N}(\boldsymbol{\mu}_l,\boldsymbol{\Sigma}_l)}[(\boldsymbol{u}^\top\boldsymbol{x})^{2t}] \le (2t-1)!!||\boldsymbol{u}||^{2t}\sum_{l=1}^{L}\lambda_l(||\boldsymbol{\mu}_l||+||\boldsymbol{\Sigma}_l^{\frac{1}{2}}||)^{2t} \tag{75}$$

**Property 7.** *With the Gaussian Mixture Model, we have*

$$\mathbb{E}_{\boldsymbol{x} \sim \sum_{l=1}^{L} \lambda_l \mathcal{N}(\boldsymbol{\mu}_l, \boldsymbol{\Sigma}_l)}[||\boldsymbol{x}||^{2t}] \leq d^t (2t-1)!! \sum_{l=1}^{L} \lambda_l (||\boldsymbol{\mu}_l|| + ||\boldsymbol{\Sigma}_l^{\frac{1}{2}}||)^{2t} \tag{76}$$

**Proof:**

$$\mathbb{E}_{\boldsymbol{x} \sim \sum_{l=1}^{L} \lambda_l \mathcal{N}(\boldsymbol{\mu}_l, \boldsymbol{\Sigma}_l)}[||\boldsymbol{x}||_2^{2t}]$$

$$= \mathbb{E}_{\boldsymbol{x} \sim \sum_{l=1}^{L} \lambda_l \mathcal{N}(\boldsymbol{\mu}_l, \boldsymbol{\Sigma}_l)}[(\sum_{i=1}^{d} x_i^2)^t]$$

$$= \mathbb{E}_{\boldsymbol{x} \sim \sum_{l=1}^{L} \lambda_l \mathcal{N}(\boldsymbol{\mu}_l, \boldsymbol{\Sigma}_l)}[d^t (\sum_{i=1}^{d} \frac{x_i^2}{d})^t]$$

$$\leq \mathbb{E}_{\boldsymbol{x} \sim \sum_{l=1}^{L} \lambda_l \mathcal{N}(\boldsymbol{\mu}_l, \boldsymbol{\Sigma}_l)}[d^t \sum_{i=1}^{d} \frac{x_i^{2t}}{d}]$$

$$= d^{t-1} \sum_{i=1}^{d} \sum_{j=1}^{L} \int_{-\infty}^{\infty} (x_i - \mu_{ji} + \mu_{ji})^{2t} \lambda_j \frac{1}{\sqrt{2\pi}\sigma_{ji}} \exp(-\frac{(x_i - \mu_{ji})^2}{2\sigma_{ji}^2}) dx_i \tag{77}$$

$$= d^{t-1} \sum_{i=1}^{d} \sum_{j=1}^{L} \sum_{k=1}^{2t} \binom{2t}{k} \lambda_j |\mu_{ji}|^{2t-k} \cdot \begin{cases} 0 & , \quad k \text{ is odd} \\ (k-1)!!\sigma_{ji}^k, & k \text{ is even} \end{cases}$$

$$\leq d^{t-1} \sum_{i=1}^{d} \sum_{j=1}^{L} \sum_{k=1}^{2t} \binom{2t}{k} \lambda_j |\mu_{ji}|^{2t-k} \sigma_j^k \cdot (2t-1)!!$$

$$= d^{t-1} \sum_{i=1}^{d} \sum_{j=1}^{L} \lambda_j (|\mu_{ji}| + \sigma_{ji})^{2t} (2t-1)!!$$

$$\leq d^t (2t-1)!! \sum_{l=1}^{L} \lambda_l (||\boldsymbol{\mu}|| + ||\boldsymbol{\Sigma}_l^{\frac{1}{2}}||)^{2t}$$

In the 3rd step, we apply Jensen inequality because $f(x) = x^t$ is convex when $x \geq 0$ and $t \geq 1$. In the 4th step we apply the Binomial theorem and the result of k-order central moment of Gaussian variable.

**Property 8.** *Under the Gaussian Mixture Model $\boldsymbol{x} \sim \sum_{l=1}^{L} \lambda_l \mathcal{N}(\boldsymbol{\mu}_l, \boldsymbol{\Sigma}_l)$ where $\boldsymbol{\Sigma}_l = \boldsymbol{\Lambda}_l^\top \boldsymbol{D}_l \boldsymbol{\Lambda}_l$, we have the following upper bound.*

$$\mathbb{E}_{\boldsymbol{x} \sim \sum_{l=1}^{L} \lambda_l \mathcal{N}(\boldsymbol{\mu}_l, \boldsymbol{\Sigma}_l)}[(\boldsymbol{u}^\top \boldsymbol{x})^{2t}] \leq (2t-1)!! ||\boldsymbol{u}||^{2t} \sum_{l=1}^{L} \lambda_l (||\boldsymbol{\mu}_l|| + ||\boldsymbol{\Sigma}_l^{\frac{1}{2}}||)^{2t} \tag{78}$$

**Proof:**
If $\boldsymbol{x} \sim \mathcal{N}(\boldsymbol{\mu}_l, \boldsymbol{\Sigma}_l)$, then $\boldsymbol{u}^\top \boldsymbol{x} \sim \mathcal{N}(\boldsymbol{u}^\top \boldsymbol{\mu}_l, \boldsymbol{u}^\top \boldsymbol{\Sigma}_l \boldsymbol{u}) = \mathcal{N}((\boldsymbol{\Lambda}_l \boldsymbol{u})^\top \boldsymbol{\Lambda}_l \boldsymbol{\mu}_l, (\boldsymbol{\Lambda}_l \boldsymbol{u})^\top \boldsymbol{D}_l (\boldsymbol{\Lambda}_l \boldsymbol{u}))$. By Property 6, we have

$$\mathbb{E}_{\boldsymbol{x} \sim \mathcal{N}(\boldsymbol{\mu}_l, \boldsymbol{\Sigma}_l)}[(\boldsymbol{u}^\top \boldsymbol{x})^{2t}] \leq (2t-1)!! ||\boldsymbol{u}||^{2t} (||\boldsymbol{\mu}_l|| + ||\boldsymbol{\Sigma}_l^{\frac{1}{2}}||)^{2t} \tag{79}$$

Then we can derive the final result.

**Property 9.** *The population risk function $\bar{f}(\boldsymbol{W})$ is defined as*

$$\bar{f}(\boldsymbol{W}) = \mathbb{E}_{\boldsymbol{x} \sim \sum_{l=1}^{L} \lambda_l \mathcal{N}(\boldsymbol{\mu}_l, \boldsymbol{\Sigma}_l)}[f_n(\boldsymbol{W})]$$

$$= \mathbb{E}_{\boldsymbol{x} \sim \sum_{l=1}^{L} \lambda_l \mathcal{N}(\boldsymbol{\mu}_l, \boldsymbol{\Sigma}_l)}\left[\frac{1}{n} \sum_{i=1}^{n} \ell(\boldsymbol{W}; \boldsymbol{x}_i, y_i)\right] \tag{80}$$

$$= \mathbb{E}_{\boldsymbol{x} \sim \sum_{l=1}^{L} \lambda_l \mathcal{N}(\boldsymbol{\mu}_l, \boldsymbol{\Sigma}_l)}[\ell(\boldsymbol{W}; \boldsymbol{x}_i, y_i)]$$

*For any permutation matrix $\boldsymbol{P}$, where $\{\pi(j)\}_{j=1}^{K}$ is the indices permuted by $\boldsymbol{P}$, we have*

$$H(\boldsymbol{WP}, \boldsymbol{x}) = \frac{1}{K} \sum_{\pi^*(j)} \phi(\boldsymbol{w}_{\pi(j)}^\top \boldsymbol{x})$$

$$= \frac{1}{K} \sum_{j=1}^{K} \phi(\boldsymbol{w}_j^\top \boldsymbol{x}) \tag{81}$$

$$= H(\boldsymbol{W}, \boldsymbol{x})$$

*Therefore,*

$$\bar{f}(\boldsymbol{W}) = \bar{f}(\boldsymbol{WP}) \tag{82}$$

*Based on (1) and (3), we can derive its gradient and Hessian as follows.*

$$\frac{\partial \ell(\boldsymbol{W}; \boldsymbol{x}, y)}{\partial \boldsymbol{w}_j} = -\frac{1}{K} \frac{y - H(\boldsymbol{W})}{H(\boldsymbol{W})(1 - H(\boldsymbol{W}))} \phi'(\boldsymbol{w}_j^\top \boldsymbol{x}) \boldsymbol{x} = \zeta(\boldsymbol{W}) \cdot \boldsymbol{x} \tag{83}$$

$$\frac{\partial^2 \ell(\boldsymbol{W}; \boldsymbol{x}, y)}{\partial \boldsymbol{w}_j \partial \boldsymbol{w}_l} = \xi_{j,l} \cdot \boldsymbol{x}\boldsymbol{x}^\top \tag{84}$$

$$\xi_{j,l}(\boldsymbol{W}) = \begin{cases} \frac{1}{K^2} \phi'(\boldsymbol{w}_j^\top \boldsymbol{x}) \phi'(\boldsymbol{w}_l^\top \boldsymbol{x}) \frac{H(\boldsymbol{W})^2 + y - 2y \cdot H(\boldsymbol{W})}{H^2(\boldsymbol{W})(1 - H(\boldsymbol{W}))^2}, & j \neq l \\ \frac{1}{K^2} \phi'(\boldsymbol{w}_j^\top \boldsymbol{x}) \phi'(\boldsymbol{w}_l^\top \boldsymbol{x}) \frac{H(\boldsymbol{W})^2 + y - 2y \cdot H(\boldsymbol{W})}{H^2(\boldsymbol{W})(1 - H(\boldsymbol{W}))^2} - \frac{1}{K} \phi''(\boldsymbol{w}_j^\top \boldsymbol{x}) \frac{y - H(\boldsymbol{W})}{H(\boldsymbol{W})(1 - H(\boldsymbol{W}))}, & j = l \end{cases} \tag{85}$$

**Property 10.** *With $D_m(\Psi$ defined in definition 5, we have*

$$(i) \ D_m(\Psi) D_{2m}(\Psi) \leq D_{3m}(\Psi) \tag{86}$$

$$(ii) \ \big(D_m(\Psi)\big)^2 \leq D_{2m}(\Psi) \tag{87}$$

**Proof:**
To prove (86), we can first compare the terms $\sum_{i=1}^{L} \lambda_i a_i \sum_{i=1}^{L} \lambda_i a_i^2$ and $\sum_{i=1}^{L} \lambda_i a_i^3$, where $a_i \geq 1$, $i \in [L]$ and $\sum_{i=1}^{L} \lambda_i = 1$.

$$\begin{aligned}
\sum_{i=1}^{L} \lambda_i a_i^3 - \sum_{i=1}^{L} \lambda_i a_i \sum_{i=1}^{L} \lambda_i a_i^2 &= \sum_{i=1}^{L} \lambda_i a_i \cdot \big(a_i^2 - \sum_{j=1}^{L} \lambda_j a_j^2\big) \\
&= \sum_{i=1}^{L} \lambda_i a_i \cdot \big((1 - \lambda_i)a_i^2 - \sum_{1 \leq j \leq L, j \neq i} \lambda_j a_j^2\big) \\
&= \sum_{i=1}^{L} \lambda_i a_i \cdot \big(\sum_{1 \leq j \leq L, j \neq i} \lambda_j a_i^2 - \sum_{1 \leq j \leq L, j \neq i} \lambda_j a_j^2\big) \\
&= \sum_{i=1}^{L} \lambda_i a_i \cdot \big(\sum_{1 \leq j \leq L, j \neq i} \lambda_j(a_i^2 - a_j^2)\big) \\
&= \sum_{1 \leq i,j \leq L, i \neq j} \big(\lambda_i \lambda_j a_i(a_i^2 - a_j^2) + \lambda_i \lambda_j a_j(a_j^2 - a_i^2)\big) \\
&= \sum_{1 \leq i,j \leq L, i \neq j} \lambda_i \lambda_j(a_i - a_j)^2(a_i + a_j) \geq 0
\end{aligned} \tag{88}$$

The second to last step is because we can find the pairwise terms $\lambda_i a_i \cdot \lambda_j(a_i^2 - a_j^2)$ and $\lambda_j a_j \cdot \lambda_i(a_j^2 - a_i^2)$ in the summation that can be putted together. From (88), we can obtain

$$\sum_{i=1}^{L} \lambda_i a_i \sum_{i=1}^{L} \lambda_i a_i^2 \leq \sum_{i=1}^{L} \lambda_i a_i^3 \tag{89}$$

Combining (89) and the definition of $D_m(\Psi)$ in (5), we can derive (86).

Similarly, to prove (87), we can first compare the terms $(\sum_{i=1}^{L} \lambda_i a_i)^2$ and $\sum_{i=1}^{L} \lambda_i a_i^2$, where $a_i \geq 1$, $i \in [L]$ and $\sum_{i=1}^{L} \lambda_i = 1$.

$$
\begin{aligned}
\sum_{i=1}^{L} \lambda_i a_i^2 - (\sum_{i=1}^{L} \lambda_i a_i)^2 &= \sum_{i=1}^{L} \lambda_i a_i \cdot \left(a_i - \sum_{j=1}^{L} \lambda_j a_j\right) \\
&= \sum_{i=1}^{L} \lambda_i a_i \cdot \left((1 - \lambda_i)a_i - \sum_{1 \leq j \leq L, j \neq i} \lambda_j a_j\right) \\
&= \sum_{i=1}^{L} \lambda_i a_i \cdot \left(\sum_{1 \leq j \leq L, j \neq i} \lambda_j a_i - \sum_{1 \leq j \leq L, j \neq i} \lambda_j a_j\right) \\
&= \sum_{i=1}^{L} \lambda_i a_i \cdot \left(\sum_{1 \leq j \leq L, j \neq i} \lambda_j (a_i - a_j)\right) \\
&= \sum_{1 \leq i,j \leq L, i \neq j} \left(\lambda_i \lambda_j a_i(a_i - a_j) + \lambda_i \lambda_j a_j(a_j - a_i)\right) \\
&= \sum_{1 \leq i,j \leq L, i \neq j} \lambda_i \lambda_j (a_i - a_j)^2 \geq 0
\end{aligned}
\tag{90}
$$

The derivation of (90) is close to (88). By (90) we have

$$
(\sum_{i=1}^{L} \lambda_i a_i)^2 \leq \sum_{i=1}^{L} \lambda_i a_i^2
\tag{91}
$$

Combining (91) and the definition of $D_m(\Psi)$ in (5), we can derive (87).

# D   PROOF OF THEOREM 1 AND COROLLARY 1

Theorem 1 is built upon **three lemmas**.

**Lemma 1** shows that with $O(dK^5 \log^2 d)$ samples, the empirical risk function is strongly convex in the neighborhood of $\boldsymbol{W}^*$.

**Lemma 2** shows that if initialized in the convex region, the gradient descent algorithm converges linearly to a critical point $\widehat{\boldsymbol{W}}_n$, which is close to $\boldsymbol{W}^*$.

**Lemma 3** shows that the Tensor Initialization Method in Subroutine 1 initializes $\boldsymbol{W}_0 \in \mathbb{R}^{d \times K}$ in the local convex region. Theorem 1 follows naturally by combining these three lemmas.

This proving approach is built upon those in Fu et al. (2020). One of our major technical contribution is extending Lemmas 1 and 2 to the Gaussian mixture model, while the results in Fu et al. (2020) only apply to Standard Gaussian models. The second major contribution is a new tensor initialization method for Gaussian mixture model such that the initial point is in the convex region (see Lemma 3). Both contributions require the development of new tools, and our analyses are much more involved than those for the standard Gaussian due to the complexity introduced by the Gaussian mixture model.

To present these lemmas, the Euclidean ball $\mathbb{B}(\boldsymbol{W}^* \boldsymbol{P}^*, r)$ is used to denote the neighborhood of $\boldsymbol{W}^* \boldsymbol{P}^*$, where $r$ is the radius of the ball.

$$
\mathbb{B}(\boldsymbol{W}^* \boldsymbol{P}^*, r) = \{\boldsymbol{W} \in \mathbb{R}^{d \times K} : ||\boldsymbol{W} - \boldsymbol{W}^* \boldsymbol{P}^*||_F \leq r\}
\tag{92}
$$

The radius of the convex region is

$$
r := \Theta\left(\frac{C_3 \epsilon_0 \cdot \sum_{l=1}^{L} \lambda_l \frac{\|\boldsymbol{\Sigma}_l^{-1}\|^{-1}}{\eta \tau^K \kappa^2} \rho\left(\frac{\boldsymbol{W}^{*\top} \boldsymbol{\mu}_l}{\delta_K(\boldsymbol{W}^*)\|\boldsymbol{\Sigma}_l^{-1}\|^{-\frac{1}{2}}}, \delta_K(\boldsymbol{W}^*)\|\boldsymbol{\Sigma}_l^{-1}\|^{-\frac{1}{2}}\right)}{K^{\frac{7}{2}}\left(\sum_{l=1}^{L} \lambda_l(\|\boldsymbol{\mu}_l\| + \|\boldsymbol{\Sigma}_l^{\frac{1}{2}}\|)^4 \sum_{l=1}^{L} \lambda_l(\|\boldsymbol{\mu}_l\| + \|\boldsymbol{\Sigma}_l^{\frac{1}{2}}\|)^8\right)^{\frac{1}{4}}}\right)
\tag{93}
$$

with some constant $C_3 > 0$.

**Lemma 1.** *(Strongly local convexity) Consider the classification model with FCN (1) and the sigmoid activation function. There exists a constant $C$ such that as long as the sample size*

$$
\begin{aligned}
n \geq & C_1 \epsilon_0^{-2} \cdot \Big( \sum_{l=1}^{L} \lambda_l (\|\boldsymbol{\mu}_l\| + \|\boldsymbol{\Sigma}_l^{\frac{1}{2}}\|)^2 \Big)^2 \\
& \cdot \Big( \sum_{l=1}^{L} \lambda_l \frac{\|\boldsymbol{\Sigma}_l^{-1}\|^{-1}}{\eta \tau^K \kappa^2} \rho\Big( \frac{\boldsymbol{W}^{*\top} \boldsymbol{\mu}_l}{\delta_K(\boldsymbol{W}^*)\|\boldsymbol{\Sigma}_l^{-1}\|^{-\frac{1}{2}}}, \delta_K(\boldsymbol{W}^*)\|\boldsymbol{\Sigma}_l^{-1}\|^{-\frac{1}{2}} \Big) \Big)^{-2} dK^5 \log^2 d
\end{aligned}
\tag{94}
$$

*for some constant $C_1 > 0$, $\epsilon_0 \in (0, \frac{1}{4})$, and any fixed permutation matrix $\boldsymbol{P} \in \mathbb{R}^{K \times K}$ we have for all $\boldsymbol{W} \in \mathbb{B}(\boldsymbol{W}^*\boldsymbol{P}, r)$,*

$$
\begin{aligned}
\Omega\Big( \frac{1 - 2\epsilon_0}{K^2} \sum_{l=1}^{L} \lambda_l \frac{\|\boldsymbol{\Sigma}_l^{-1}\|^{-1}}{\eta \tau^K \kappa^2} \rho\Big( \frac{\boldsymbol{W}^{*\top} \boldsymbol{\mu}_l}{\delta_K(\boldsymbol{W}^*)\|\boldsymbol{\Sigma}_l^{-1}\|^{-\frac{1}{2}}}, \delta_K(\boldsymbol{W}^*)\|\boldsymbol{\Sigma}_l^{-1}\|^{-\frac{1}{2}} \Big) \Big) \cdot \boldsymbol{I}_{dK} \\
\preceq \nabla^2 f_n(\boldsymbol{W}) \preceq C_2 \sum_{l=1}^{L} \lambda_l (\|\tilde{\boldsymbol{\mu}}_l\|_\infty + \|\boldsymbol{\Sigma}_l^{\frac{1}{2}}\|)^2 \cdot \boldsymbol{I}_{dK}
\end{aligned}
\tag{95}
$$

*with probability at least $1 - d^{-10}$ for some constant $C_2 > 0$.*

**Lemma 2.** *(Linear convergence of gradient descent) Assume the conditions in Lemma 1 hold. Given any fixed permutation matrix $\boldsymbol{P} \in \mathbb{R}^{K \times K}$, if the local convexity of $\mathbb{B}(\boldsymbol{W}^*\boldsymbol{P}, r)$ holds, there exists a critical point in $\mathbb{B}(\boldsymbol{W}^*\boldsymbol{P}, r)$ for some constant $C_3 > 0$, and $\epsilon_0 \in (0, \frac{1}{2})$, such that*

$$
\|\widehat{\boldsymbol{W}}_n - \boldsymbol{W}^*\boldsymbol{P}\|_F \leq O\Big( \frac{K^{\frac{5}{2}} \sqrt{\sum_{l=1}^{L} \lambda_l (\|\boldsymbol{\mu}_l\| + \|\boldsymbol{\Sigma}_l^{\frac{1}{2}}\|)^2}(1 + \xi)}{\sum_{l=1}^{L} \lambda_l \frac{\|\boldsymbol{\Sigma}_l^{-1}\|^{-1}}{\eta \tau^K \kappa^2} \rho\Big( \frac{\boldsymbol{W}^{*\top} \boldsymbol{\mu}_l}{\delta_K(\boldsymbol{W}^*)\|\boldsymbol{\Sigma}_l^{-1}\|^{-\frac{1}{2}}}, \delta_K(\boldsymbol{W}^*)\|\boldsymbol{\Sigma}_l^{-1}\|^{-\frac{1}{2}} \Big)} \sqrt{\frac{d \log n}{n}} \Big)
\tag{96}
$$

*If the initial point $\boldsymbol{W}_0 \in \mathbb{B}(\boldsymbol{W}^*\boldsymbol{P}, r)$, the gradient descent linearly converges to $\widehat{\boldsymbol{W}}_n$, i.e.,*

$$
\|\boldsymbol{W}_t - \widehat{\boldsymbol{W}}_n\|_F \leq \Big( 1 - \Omega\Big( \frac{\sum_{l=1}^{L} \lambda_l \frac{\|\boldsymbol{\Sigma}_l^{-1}\|^{-1}}{\eta \tau^K \kappa^2} \rho\Big( \frac{\boldsymbol{W}^{*\top} \boldsymbol{\mu}_l}{\delta_K(\boldsymbol{W}^*)\|\boldsymbol{\Sigma}_l^{-1}\|^{-\frac{1}{2}}}, \delta_K(\boldsymbol{W}^*)\|\boldsymbol{\Sigma}_l^{-1}\|^{-\frac{1}{2}} \Big)}{K^2 \sum_{l=1}^{L} \lambda_l (\|\boldsymbol{\mu}_l\| + \|\boldsymbol{\Sigma}_l^{\frac{1}{2}}\|)^2} \Big) \Big)^t \|\boldsymbol{W}_0 - \widehat{\boldsymbol{W}}_n\|_F
\tag{97}
$$

*with probability at least $1 - d^{-10}$.*

**Lemma 3.** *(Tensor initialization) For classification model, with $D_6(\Psi)$ defined in Definition 5, we have that if the sample size*

$$
n \geq \kappa^8 K^4 \tau^{12} D_6(\Psi) \cdot d \log^2 d,
\tag{98}
$$

*then the output $\boldsymbol{W}_0 \in \mathbb{R}^{d \times K}$ satisfies*

$$
\|\boldsymbol{W}_0 - \boldsymbol{W}^*\boldsymbol{P}^*\| \lesssim \kappa^6 K^3 \cdot \tau^6 \sqrt{D_6(\Psi)} \sqrt{\frac{d \log n}{n}} \|\boldsymbol{W}^*\|
\tag{99}
$$

*with probability at least $1 - n^{-\Omega(\delta_1^4)}$ for a specific permutation matrix $\boldsymbol{P}^* \in \mathbb{R}^{K \times K}$.*

**Proof of Theorem 1**
From Lemma 2 and Lemma 3, we know that if $n$ is sufficiently large such that the initialization $\boldsymbol{W}_0$ by the tensor method is in the region $\mathbb{B}(\boldsymbol{W}^*\boldsymbol{P}, r)$, then the gradient descent method converges to a critical point $\widehat{\boldsymbol{W}}_n$ that is sufficiently close to $\boldsymbol{W}^*$. To achieve that, one sufficient condition is

$$
\begin{aligned}
\|\boldsymbol{W}_0 - \boldsymbol{W}^*\boldsymbol{P}^*\|_F \leq & \sqrt{K}\|\boldsymbol{W}_0 - \boldsymbol{W}^*\boldsymbol{P}^*\| \leq \kappa^6 K^{\frac{7}{2}} \cdot \tau^6 \sqrt{D_6(\Psi)} \sqrt{\frac{d \log n}{n}} \|\boldsymbol{W}^*\boldsymbol{P}\| \\
& \leq \frac{C_3 \epsilon_0 \Gamma(\Psi) \sigma_{\max}^2}{K^{\frac{7}{2}} \Big( \sum_{l=1}^{L} \lambda_l (\|\boldsymbol{\mu}_l\| + \|\boldsymbol{\Sigma}_l^{\frac{1}{2}}\|)^4 \sum_{l=1}^{L} \lambda_l (\|\boldsymbol{\mu}_l\| + \|\boldsymbol{\Sigma}_l^{\frac{1}{2}}\|)^8 \Big)^{\frac{1}{4}}}
\end{aligned}
\tag{100}
$$

where the first inequality follows from $||\boldsymbol{W}||_F \leq \sqrt{K}||\boldsymbol{W}||$ for $\boldsymbol{W} \in \mathbb{R}^{d \times K}$, the second inequality comes from Lemma 3, and the third inequality comes from the requirement to be in the region $\mathbb{B}(\boldsymbol{W}^*\boldsymbol{P}, r)$. That is equivalent to the following condition

$$
\begin{aligned}
n \geq & C_0 \epsilon_0^{-2} \cdot \tau^{12} \kappa^{12} K^{14} \Big( \sum_{l=1}^{L} \lambda_l (\|\boldsymbol{\mu}_l\| + \|\boldsymbol{\Sigma}_l^{\frac{1}{2}}\|)^4 \sum_{l=1}^{L} \lambda_l (\|\boldsymbol{\mu}_l\| + \|\boldsymbol{\Sigma}_l^{\frac{1}{2}}\|)^8 \Big)^{\frac{1}{2}} \\
& \cdot (\delta_1(\boldsymbol{W}^*))^2 D_6(\Psi) \Gamma(\Psi)^{-2} \sigma_{\max}^{-4} \cdot d \log^2 d
\end{aligned}
\tag{101}
$$

where $C_0 = \max\{C_4, C_3^{-2}\}$. By Definition 5, we can obtain

$$
\Big( \sum_{l=1}^{L} \lambda_l (\|\boldsymbol{\mu}_l\| + \|\boldsymbol{\Sigma}_l^{\frac{1}{2}}\|)^4 \sum_{l=1}^{L} \lambda_l (\|\boldsymbol{\mu}_l\| + \|\boldsymbol{\Sigma}_l^{\frac{1}{2}}\|)^8 \Big)^{\frac{1}{2}} \leq \sqrt{D_4(\Psi) D_8(\Psi)} \sigma_{\max}^6
\tag{102}
$$

From Property 10, we have that

$$
\begin{aligned}
& \sqrt{D_4(\Psi) D_8(\Psi)} D_6(\Psi) \\
\leq & \sqrt{D_{12}(\Psi)} \sqrt{D_{12}(\Psi)} = D_{12}(\Psi)
\end{aligned}
\tag{103}
$$

Plugging (102), (103) into (101), we have

$$
n \geq C_0 \epsilon_0^{-2} \cdot \kappa^{12} K^{14} (\sigma_{\max} \delta_1(\boldsymbol{W}^*))^2 \tau^{12} \Gamma(\Psi)^{-2} D_{12}(\Psi) \cdot d \log^2 d
\tag{104}
$$

Considering the requirements on the sample complexity in (94), (98), and (104), (104) shows a sufficient number of samples. Taking the union bound of all the failure probabilities in Lemma 1, and 3, (104) holds with probability $1 - d^{-10}$.

By Property 3.4, $\rho(\frac{\boldsymbol{W}^{*\top}\boldsymbol{\mu}_l}{\delta_K(\boldsymbol{W}^*)\|\boldsymbol{\Sigma}_l^{-1}\|^{-\frac{1}{2}}}, \delta_K(\boldsymbol{W}^*)\|\boldsymbol{\Sigma}_l^{-1}\|^{-\frac{1}{2}})$ can be lower bounded by positive and monotonically decreasing functions $\mathcal{L}_m(\frac{(\boldsymbol{\Lambda}_l \boldsymbol{W}^*)^{\top}\tilde{\boldsymbol{\mu}}_l}{\delta_K(\boldsymbol{W}^*)\|\boldsymbol{\Sigma}_l^{-1}\|^{-\frac{1}{2}}}, \delta_K(\boldsymbol{W}^*)\|\boldsymbol{\Sigma}_l^{-1}\|^{-\frac{1}{2}})$ when everything else except $|\tilde{\boldsymbol{\mu}}_{l(i)}|$ is fixed, or $\mathcal{L}_s(\frac{\boldsymbol{W}^{*\top}\boldsymbol{\mu}_l}{\delta_K(\boldsymbol{W}^*)\|\boldsymbol{\Sigma}_l^{-1}\|^{-\frac{1}{2}}}, \delta_K(\boldsymbol{W}^*)\|\boldsymbol{\Sigma}_l^{-1}\|^{-\frac{1}{2}})$ when everything else except $\|\boldsymbol{\Sigma}_l^{\frac{1}{2}}\|$ is fixed. Then, by replacing the lower bound of $\rho(\frac{\boldsymbol{W}^{*\top}\boldsymbol{\mu}_l}{\delta_K(\boldsymbol{W}^*)\|\boldsymbol{\Sigma}_l^{-1}\|^{-\frac{1}{2}}}, \delta_K(\boldsymbol{W}^*)\|\boldsymbol{\Sigma}_l^{-1}\|^{-\frac{1}{2}})$ with these two functions in $\Gamma(\Psi)$, we can have an upper bound of $(\sigma_{\max}\delta_1(\boldsymbol{W}^*))^2 \tau^{12} \Gamma(\Psi)^{-2} D_{12}(\Psi)$, denoted as $\mathcal{B}(\Psi)$.

To be more specific, when everything else except $|\tilde{\boldsymbol{\mu}}_{l(i)}|$ is fixed, $\mathcal{L}_m(\frac{(\boldsymbol{\Lambda}_l \boldsymbol{W}^*)^{\top}\tilde{\boldsymbol{\mu}}_l}{\delta_K(\boldsymbol{W}^*)\|\boldsymbol{\Sigma}_l^{-1}\|^{-\frac{1}{2}}}, \delta_K(\boldsymbol{W}^*)\|\boldsymbol{\Sigma}_l^{-1}\|^{-\frac{1}{2}})$ is plugged in $\mathcal{B}(\Psi)$. Then since that $D_{12}(\Psi)$ and $\mathcal{L}_m(\frac{(\boldsymbol{\Lambda}_l \boldsymbol{W}^*)^{\top}\tilde{\boldsymbol{\mu}}_l}{\delta_K(\boldsymbol{W}^*)\|\boldsymbol{\Sigma}_l^{-1}\|^{-\frac{1}{2}}}, \delta_K(\boldsymbol{W}^*)\|\boldsymbol{\Sigma}_l^{-1}\|^{-\frac{1}{2}})$ are both increasing function of $|\tilde{\boldsymbol{\mu}}_{l(i)}|$, $\mathcal{B}(\Psi)$ is an increasing function of $|\tilde{\boldsymbol{\mu}}_{l(i)}|$.

When everything else except $\|\boldsymbol{\Sigma}_l^{\frac{1}{2}}\|$ is fixed, if $\|\boldsymbol{\Sigma}_l^{\frac{1}{2}}\| = \sigma_{\max} > \zeta_s$, then $\sigma_{\max}^2 \tau^{12} D_{12}(\Psi)$ is an increasing function of $\|\boldsymbol{\Sigma}_l^{\frac{1}{2}}\|$. Since that $\mathcal{L}_s(\frac{\boldsymbol{W}^{*\top}\boldsymbol{\mu}_l}{\delta_K(\boldsymbol{W}^*)\|\boldsymbol{\Sigma}_l^{-1}\|^{-\frac{1}{2}}}, \delta_K(\boldsymbol{W}^*)\|\boldsymbol{\Sigma}_l^{-1}\|^{-\frac{1}{2}})$ is a decreasing function, $\mathcal{L}_s(\frac{\boldsymbol{W}^{*\top}\boldsymbol{\mu}_l}{\delta_K(\boldsymbol{W}^*)\|\boldsymbol{\Sigma}_l^{-1}\|^{-\frac{1}{2}}}, \delta_K(\boldsymbol{W}^*)\|\boldsymbol{\Sigma}_l^{-1}\|^{-\frac{1}{2}})^{-2}$ is an increasing function of $\|\boldsymbol{\Sigma}_l^{\frac{1}{2}}\|$. Hence, $\mathcal{B}(\Psi)$ is an increasing function of $\|\boldsymbol{\Sigma}_l^{\frac{1}{2}}\|$. Moreover, when all $\|\boldsymbol{\Sigma}_l^{\frac{1}{2}}\| < \zeta_{s'}$ and go to 0, two decreasing functions of $\|\boldsymbol{\Sigma}_l^{\frac{1}{2}}\|$, $\sigma_{\max}^2 \mathcal{L}_s(\frac{\boldsymbol{W}^{*\top}\boldsymbol{\mu}_l}{\delta_K(\boldsymbol{W}^*)\|\boldsymbol{\Sigma}_l^{-1}\|^{-\frac{1}{2}}}, \delta_K(\boldsymbol{W}^*)\|\boldsymbol{\Sigma}_l^{-1}\|^{-\frac{1}{2}})^{-2}$ and $D_{12}(\Psi)$ will be the dominant term of $\mathcal{B}(\Psi)$. Therefore, $\mathcal{B}(\Psi)$ increases to infinity as all $\|\boldsymbol{\Sigma}_l^{\frac{1}{2}}\|$'s go to 0. In sum,

we can define a universe $\mathcal{B}(\Psi)$ as:

$$
\begin{aligned}
&\mathcal{B}(\Psi) \\
&=
\begin{cases}
(\sigma_{\max}\delta_1(\boldsymbol{W}^*))^2\tau^{12}\Big(\sum_{l=1}^{L}\frac{\lambda_l\|\boldsymbol{\Sigma}_l^{-1}\|^{-1}}{\eta\sigma_{\max}^2}\mathcal{L}_m\big(\frac{(\boldsymbol{\Lambda}_l\boldsymbol{W}^*)^\top\tilde{\boldsymbol{\mu}}_l}{\delta_K(\boldsymbol{W}^*)\|\boldsymbol{\Sigma}_l^{-1}\|^{-\frac{1}{2}}},\delta_K(\boldsymbol{W}^*)\|\boldsymbol{\Sigma}_l^{-1}\|^{-\frac{1}{2}}\big)\Big)^{-2} \\
\quad\cdot D_{12}(\Psi),\text{if }\boldsymbol{S}\text{ is fixed} \\[4pt]
(\sigma_{\max}\delta_1(\boldsymbol{W}^*))^2\tau^{12}\Big(\sum_{l=1}^{L}\frac{\lambda_l\|\boldsymbol{\Sigma}_l^{-1}\|^{-1}}{\eta\sigma_{\max}^2}\mathcal{L}_s\big(\frac{(\boldsymbol{\Lambda}_l\boldsymbol{W}^*)^\top\tilde{\boldsymbol{\mu}}_l}{\delta_K(\boldsymbol{W}^*)\|\boldsymbol{\Sigma}_l^{-1}\|^{-\frac{1}{2}}},\delta_K(\boldsymbol{W}^*)\|\boldsymbol{\Sigma}_l^{-1}\|^{-\frac{1}{2}}\big)\Big)^{-2} \\
\quad\cdot D_{12}(\Psi),\text{if }\mathbf{M}\text{ is fixed} \\[4pt]
(\sigma_{\max}\delta_1(\boldsymbol{W}^*))^2\tau^{12}\Big(\sum_{l=1}^{L}\frac{\lambda_l\|\boldsymbol{\Sigma}_l^{-1}\|^{-1}}{\eta\sigma_{\max}^2}\rho\big(\frac{(\boldsymbol{\Lambda}_l\boldsymbol{W}^*)^\top\tilde{\boldsymbol{\mu}}_l}{\delta_K(\boldsymbol{W}^*)\|\boldsymbol{\Sigma}_l^{-1}\|^{-\frac{1}{2}}},\delta_K(\boldsymbol{W}^*)\|\boldsymbol{\Sigma}_l^{-1}\|^{-\frac{1}{2}}\big)\Big)^{-2} \\
\quad\cdot D_{12}(\Psi),\text{otherwise}
\end{cases}
\end{aligned}
\tag{105}
$$

where $\mathcal{L}_m$, $\mathcal{L}_s$ and $D_{12}$ are defined in (38), (43) and Definition 5, respectively. Hence, we have

$$
n \geq poly(\epsilon_0^{-1},\kappa,\eta,\tau K)\mathcal{B}(\Psi)\cdot d\log^2 d
\tag{106}
$$

Similarly, by replacing $\rho\big(\frac{\boldsymbol{W}^{*\top}\boldsymbol{\mu}_l}{\delta_K(\boldsymbol{W}^*)\|\boldsymbol{\Sigma}_l^{-1}\|^{-\frac{1}{2}}},\delta_K(\boldsymbol{W}^*)\|\boldsymbol{\Sigma}_l^{-1}\|^{-\frac{1}{2}}\big)$ with $\mathcal{L}_m\big(\frac{(\boldsymbol{\Lambda}_l\boldsymbol{W}^*)^\top\tilde{\boldsymbol{\mu}}_l}{\delta_K(\boldsymbol{W}^*)\|\boldsymbol{\Sigma}_l^{-1}\|^{-\frac{1}{2}}},\delta_K(\boldsymbol{W}^*)\|\boldsymbol{\Sigma}_l^{-1}\|^{-\frac{1}{2}}\big)$ when everything else except $|\tilde{\boldsymbol{\mu}}_{l(i)}|$ is fixed, or $\mathcal{L}_s\big(\frac{\boldsymbol{W}^{*\top}\boldsymbol{\mu}_l}{\delta_K(\boldsymbol{W}^*)\|\boldsymbol{\Sigma}_l^{-1}\|^{-\frac{1}{2}}},\delta_K(\boldsymbol{W}^*)\|\boldsymbol{\Sigma}_l^{-1}\|^{-\frac{1}{2}}\big)$ (or $\|\boldsymbol{\Sigma}_l^{-1}\|\mathcal{L}_s\big(\frac{\boldsymbol{W}^{*\top}\boldsymbol{\mu}_l}{\delta_K(\boldsymbol{W}^*)\|\boldsymbol{\Sigma}_l^{-1}\|^{-\frac{1}{2}}},\delta_K(\boldsymbol{W}^*)\|\boldsymbol{\Sigma}_l^{-1}\|^{-\frac{1}{2}}\big)$ for $\|\boldsymbol{\Sigma}_l^{-1}\|^{-1} \geq 1$) when everything else except $\|\boldsymbol{\Sigma}_l^{\frac{1}{2}}\|$ is fixed, (97) can also be transferred to another feasible upper bound. We denote the modified version of the convergence rate as $v = 1 - K^{-2}q(\Psi)$. Since that $q(\Psi)$ is a ratio between the smallest and the largest singular value of $\nabla^2\bar{f}(\boldsymbol{W}^*)$, we have $q(\Psi) \in (0,1)$. Hence, we can obtain $1 - K^{-2}q(\Psi) \in (0,1)$ by $K \geq 1$. When everything else except $|\tilde{\boldsymbol{\mu}}_{l(i)}|$ is fixed, since that $\mathcal{L}_m\big(\frac{(\boldsymbol{\Lambda}_l\boldsymbol{W}^*)^\top\tilde{\boldsymbol{\mu}}_l}{\delta_K(\boldsymbol{W}^*)\|\boldsymbol{\Sigma}_l^{-1}\|^{-\frac{1}{2}}},\delta_K(\boldsymbol{W}^*)\|\boldsymbol{\Sigma}_l^{-1}\|^{-\frac{1}{2}}\big)$ is monotonically decreasing and $\sum_{l=1}^{L}\lambda(\|\boldsymbol{\mu}_l\| + \|\boldsymbol{\Sigma}_l^{\frac{1}{2}}\|)^2$ is increasing as $|\tilde{\boldsymbol{\mu}}_{l(i)}|$ increases, $v$ is an increasing function of $|\tilde{\boldsymbol{\mu}}_{l(i)}|$ to 1. Similarly, when everything else except $\|\boldsymbol{\Sigma}_l^{\frac{1}{2}}\|$ is fixed where $\|\boldsymbol{\Sigma}_l^{\frac{1}{2}}\| \geq \max\{1,\zeta_s\}$, $\frac{1}{\sum_{l=1}^{L}\lambda_l(\|\boldsymbol{\mu}_l\|+\|\boldsymbol{\Sigma}_l^{\frac{1}{2}}\|)^2}$ decreases to 0 as $\|\boldsymbol{\Sigma}_l\|$ increases. We replace $\rho\big(\frac{\boldsymbol{W}^{*\top}\boldsymbol{\mu}_l}{\delta_K(\boldsymbol{W}^*)\|\boldsymbol{\Sigma}_l^{-1}\|^{-\frac{1}{2}}},\delta_K(\boldsymbol{W}^*)\|\boldsymbol{\Sigma}_l^{-1}\|^{-\frac{1}{2}}\big)$ by $\|\boldsymbol{\Sigma}_l^{-1}\|\mathcal{L}_s\big(\frac{\boldsymbol{W}^{*\top}\boldsymbol{\mu}_l}{\delta_K(\boldsymbol{W}^*)\|\boldsymbol{\Sigma}_l^{-1}\|^{-\frac{1}{2}}},\delta_K(\boldsymbol{W}^*)\|\boldsymbol{\Sigma}_l^{-1}\|^{-\frac{1}{2}}\big)$ and then

$$
\begin{aligned}
&\|\boldsymbol{\Sigma}_l^{-1}\|^{-1}\cdot\|\boldsymbol{\Sigma}_l^{-1}\|\mathcal{L}_s\big(\frac{\boldsymbol{W}^{*\top}\boldsymbol{\mu}_l}{\delta_K(\boldsymbol{W}^*)\|\boldsymbol{\Sigma}_l^{-1}\|^{-\frac{1}{2}}},\delta_K(\boldsymbol{W}^*)\|\boldsymbol{\Sigma}_l^{-1}\|^{-\frac{1}{2}}\big) \\
&=\mathcal{L}_s\big(\frac{\boldsymbol{W}^{*\top}\boldsymbol{\mu}_l}{\delta_K(\boldsymbol{W}^*)\|\boldsymbol{\Sigma}_l^{-1}\|^{-\frac{1}{2}}},\delta_K(\boldsymbol{W}^*)\|\boldsymbol{\Sigma}_l^{-1}\|^{-\frac{1}{2}}\big)
\end{aligned}
\tag{107}
$$

is an decreasing function less than $\rho\big(\frac{\boldsymbol{W}^{*\top}\boldsymbol{\mu}_l}{\delta_K(\boldsymbol{W}^*)\|\boldsymbol{\Sigma}_l^{-1}\|^{-\frac{1}{2}}},\delta_K(\boldsymbol{W}^*)\|\boldsymbol{\Sigma}_l^{-1}\|^{-\frac{1}{2}}\big)$. Therefore, $v$ is an increasing function of $\|\boldsymbol{\Sigma}_l^{\frac{1}{2}}\|$ to 1 when $\|\boldsymbol{\Sigma}_l^{\frac{1}{2}}\| \geq \max\{1,\zeta_s\}$. When everything else except all $\|\boldsymbol{\Sigma}_l^{\frac{1}{2}}\| \leq \zeta_{s'}$'s go to 0, all $\mathcal{L}_s\big(\frac{\boldsymbol{W}^{*\top}\boldsymbol{\mu}_l}{\delta_K(\boldsymbol{W}^*)\|\boldsymbol{\Sigma}_l^{-1}\|^{-\frac{1}{2}}},\delta_K(\boldsymbol{W}^*)\|\boldsymbol{\Sigma}_l^{-1}\|^{-\frac{1}{2}}\big)$'s will decrease and all $\frac{\|\boldsymbol{\Sigma}_l^{-1}\|^{-1}}{\sum_{l=1}^{L}\lambda_l(\|\boldsymbol{\mu}_l\|_\infty+\|\boldsymbol{\Sigma}_l^{\frac{1}{2}}\|)^2}$'s will decrease to 0. Therefore, $v$ increases to 1.

$q(\Psi)$ can then be defined as

$$q(\Psi)$$
$$= \begin{cases} \Omega\Big(\dfrac{\sum_{l=1}^{L}\lambda_l \frac{\|\mathbf{\Sigma}_l^{-1}\|^{-1}}{\eta\tau K\kappa^2}\mathcal{L}_m\big(\frac{(\mathbf{\Lambda}_l\mathbf{W}^*)^\top\tilde{\boldsymbol{\mu}}_l}{\delta_K(\mathbf{W}^*)\|\mathbf{\Sigma}_l^{-1}\|^{-\frac{1}{2}}},\delta_K(\mathbf{W}^*)\|\mathbf{\Sigma}_l^{-1}\|^{-\frac{1}{2}}\big)}{\sum_{l=1}^{L}\lambda_l(\|\boldsymbol{\mu}_l\|+\|\mathbf{\Sigma}_l^{\frac{1}{2}}\|)^2}\Big)\Big), \\ \quad \text{if } \mathbf{S} \text{ is fixed} \\[2ex] \Omega\Big(\dfrac{\sum_{l=1}^{L}\lambda_l \frac{\|\mathbf{\Sigma}_l^{-1}\|^{-1}}{\eta\tau K\kappa^2}\mathcal{L}_s\big(\frac{\mathbf{W}^{*\top}\boldsymbol{\mu}_l}{\delta_K(\mathbf{W}^*)\|\mathbf{\Sigma}_l^{-1}\|^{-\frac{1}{2}}},\delta_K(\mathbf{W}^*)\|\mathbf{\Sigma}_l^{-1}\|^{-\frac{1}{2}}\big)}{\sum_{l=1}^{L}\lambda_l(\|\boldsymbol{\mu}_l\|+\|\mathbf{\Sigma}_l^{\frac{1}{2}}\|)^2}\Big), \\ \quad \text{if } \mathbf{M} \text{ is fixed and all } \|\mathbf{\Sigma}_l^{\frac{1}{2}}\| \le \zeta_{s'} \\[2ex] \Omega\Big(\dfrac{\lambda_l\frac{1}{\eta\tau K\kappa^2}\mathcal{L}_s\big(\frac{\mathbf{W}^{*\top}\boldsymbol{\mu}_i}{\delta_K(\mathbf{W}^*)\|\mathbf{\Sigma}_i^{-1}\|^{-\frac{1}{2}}},\delta_K(\mathbf{W}^*)\|\mathbf{\Sigma}_i^{-1}\|^{-\frac{1}{2}}\big)+\sum_{l\neq i}r(\lambda_l,\boldsymbol{\mu}_l,\mathbf{\Sigma}_l,\mathbf{W}^*)}{\sum_{l=1}^{L}\lambda_l(\|\boldsymbol{\mu}_l\|+\|\mathbf{\Sigma}_l^{\frac{1}{2}}\|)^2}\Big), \\ \quad \text{if } \mathbf{M} \text{ is fixed and one } \|\mathbf{\Sigma}_i^{\frac{1}{2}}\| \ge \max\{1,\zeta_s\} \\[2ex] \Omega\Big(\dfrac{\sum_{l=1}^{L}\lambda_l\frac{\|\mathbf{\Sigma}_l^{-1}\|^{-1}}{\eta\tau K\kappa^2}\rho\big(\frac{\mathbf{W}^{*\top}\boldsymbol{\mu}_l}{\delta_K(\mathbf{W}^*)\|\mathbf{\Sigma}_l^{-1}\|^{-\frac{1}{2}}},\delta_K(\mathbf{W}^*)\|\mathbf{\Sigma}_l^{-1}\|^{-\frac{1}{2}}\big)}{\sum_{l=1}^{L}\lambda_l(\|\boldsymbol{\mu}_l\|+\|\mathbf{\Sigma}_l^{\frac{1}{2}}\|)^2}\Big), \\ \quad \text{otherwise} \end{cases} \tag{108}$$

where $r(\lambda_l,\boldsymbol{\mu}_l,\mathbf{\Sigma}_l,\mathbf{W}^*) = \lambda_l\frac{\|\mathbf{\Sigma}_l^{-1}\|^{-1}}{\eta\tau K\kappa^2}\rho\big(\frac{\mathbf{W}^{*\top}\boldsymbol{\mu}_l}{\delta_K(\mathbf{W}^*)\|\mathbf{\Sigma}_l^{-1}\|^{-\frac{1}{2}}},\delta_K(\mathbf{W}^*)\|\mathbf{\Sigma}_l^{-1}\|^{-\frac{1}{2}}\big)$. Note that here the $\rho(\cdot)$ function is defined in Definition 3. $\mathcal{L}_m(\cdot)$ and $\mathcal{L}_s(\cdot)$ are defined in (38) and (43), respectively. The bound of $\|\widehat{\mathbf{W}}_n - \mathbf{W}^*\mathbf{P}\|_F$ is directly from (96). We can derive that

$$\mathcal{E}_w(\Psi) = O\Big(\frac{\sqrt{\sum_{j=1}^{L}\lambda_l(\|\boldsymbol{\mu}_j\|+\|\mathbf{\Sigma}_j^{\frac{1}{2}}\|)^2}}{\sum_{j=1}^{L}\lambda_l\|\mathbf{\Sigma}_j^{-1}\|^{-1}\rho\big(\frac{\mathbf{W}^{*\top}\boldsymbol{\mu}_j}{\delta_K(\mathbf{W}^*)\|\mathbf{\Sigma}_j^{-1}\|^{-\frac{1}{2}}},\delta_K(\mathbf{W}^*)\|\mathbf{\Sigma}_j^{-1}\|^{-\frac{1}{2}}\big)}\Big) \tag{109}$$

$$\mathcal{E}(\Psi) = O\Big(\frac{\sum_{j=1}^{L}\lambda_l(\|\boldsymbol{\mu}_j\|+\|\mathbf{\Sigma}_j^{\frac{1}{2}}\|)^2}{\sum_{j=1}^{L}\lambda_l\|\mathbf{\Sigma}_j^{-1}\|^{-1}\rho\big(\frac{\mathbf{W}^{*\top}\boldsymbol{\mu}_j}{\delta_K(\mathbf{W}^*)\|\mathbf{\Sigma}_j^{-1}\|^{-\frac{1}{2}}},\delta_K(\mathbf{W}^*)\|\mathbf{\Sigma}_j^{-1}\|^{-\frac{1}{2}}\big)}\Big) \tag{110}$$

$$\mathcal{E}_l(\Psi) = O\Big(\frac{\sqrt{\sum_{j=1}^{L}\lambda_l(\|\boldsymbol{\mu}_j\|+\|\mathbf{\Sigma}_j^{\frac{1}{2}}\|)^2}(\|\boldsymbol{\mu}_l\|+\|\mathbf{\Sigma}_l\|^{\frac{1}{2}})}{\sum_{j=1}^{L}\lambda_l\|\mathbf{\Sigma}_j^{-1}\|^{-1}\rho\big(\frac{\mathbf{W}^{*\top}\boldsymbol{\mu}_j}{\delta_K(\mathbf{W}^*)\|\mathbf{\Sigma}_j^{-1}\|^{-\frac{1}{2}}},\delta_K(\mathbf{W}^*)\|\mathbf{\Sigma}_j^{-1}\|^{-\frac{1}{2}}\big)}\Big) \tag{111}$$

The discussion of the monotonicity of $\mathcal{E}_w(\Psi)$, $\mathcal{E}(\Psi)$ and $\mathcal{E}_l(\Psi)$ can follow the analysis of $q(\Psi)$. We finish our proof of Theorem 1 here. The parameters $\mathcal{B}(\Psi)$, $q(\Psi)$, $\mathcal{E}_w(\Psi)$, $\mathcal{E}(\Psi)$, and $\mathcal{E}_l(\Psi)$ can be found in 105, 108, 109, 110, and 111, respectively.

**Proof of Corollary 1**:
The monotonicity analysis has been included in the proof of Theorem 1. In this part, we specify our proof for the results in Table 1. For simplicity, we denote $\rho_l = \rho\big(\frac{\mathbf{W}^{*\top}\boldsymbol{\mu}_l}{\delta_K(\mathbf{W}^*)\|\mathbf{\Sigma}_l^{-1}\|^{-\frac{1}{2}}},\delta_K(\mathbf{W}^*)\|\mathbf{\Sigma}_l^{-1}\|^{-\frac{1}{2}}\big)$.
When everything else except $\|\mathbf{\Sigma}_l\|^{\frac{1}{2}}$ is fixed, if $\|\mathbf{\Sigma}_l\| = o(1)$, by some basic mathematical computation, then we have

$$n_{sc} = C_0\epsilon_0^{-2}\cdot\eta^2\tau^{12}\kappa^{16}K^{14}\Big(\sum_{l=1}^{L}\lambda_l(\|\tilde{\boldsymbol{\mu}}_l\|+\|\mathbf{\Sigma}_l^{\frac{1}{2}}\|)^4\sum_{l=1}^{L}\lambda_l(\|\tilde{\boldsymbol{\mu}}_l\|+\|\mathbf{\Sigma}_l^{\frac{1}{2}}\|)^8\Big)^{\frac{1}{2}}(\delta_1(\mathbf{W}^*))^2 D_6(\Psi)$$
$$\cdot\Big(\frac{1}{\sum_{l=1}^{L}\lambda_l\|\mathbf{\Sigma}_l^{-1}\|^{-1}\rho_l}\Big)^2\cdot d\log^2 d$$
$$\lesssim \text{poly}(\epsilon_0^{-1},\eta,\tau,\kappa,K,\delta_1(\mathbf{W}^*))\cdot d\log^2 d\cdot O\Big(\lambda_L\frac{1}{\|\mathbf{\Sigma}_L^{\frac{1}{2}}\|^6}\Big)$$
$$\tag{112}$$

$$v(\Psi) = 1 - \frac{\sum_{l=1}^{L} \lambda_l \frac{\|\mathbf{\Sigma}_l^{-1}\|^{-1}}{\eta \kappa^2} \rho_l}{K^2 (\sum_{l=1}^{L} \lambda_l (\|\boldsymbol{\mu}_l\| + \|\mathbf{\Sigma}_l^{\frac{1}{2}}\|)^2)} \tag{113}$$

$$\leq 1 - \frac{\lambda_l}{K^2 \eta \kappa^2 \tau^K} \Theta(\|\mathbf{\Sigma}_l\|^3)$$

$$\|\widehat{\mathbf{W}}_n - \mathbf{W}^* \mathbf{P}^*\| \leq O\Big(\frac{K^{\frac{5}{2}} \sqrt{\sum_{l=1}^{L} \lambda_l (\|\boldsymbol{\mu}_l\| + \|\mathbf{\Sigma}_l^{\frac{1}{2}}\|)^2}(1 + \xi)}{\sum_{l=1}^{L} \lambda_l \frac{\|\mathbf{\Sigma}_l^{-1}\|^{-1}}{\eta \tau^K \kappa^2} \rho(\frac{\mathbf{W}^{* \top} \boldsymbol{\mu}_l}{\delta_K(\mathbf{W}^*) \|\mathbf{\Sigma}_l^{-1}\|^{-\frac{1}{2}}}, \delta_K(\mathbf{W}^*) \|\mathbf{\Sigma}_l^{-1}\|^{-\frac{1}{2}})} \sqrt{\frac{d \log n}{n}}\Big)$$

$$\lesssim \text{poly}(\eta, \kappa, \tau, \delta_K(\mathbf{W}^*)) \sqrt{\frac{d \log n}{n}} K^2 (1 + \xi) \cdot O(1 - \|\mathbf{\Sigma}_l\|^3) \tag{114}$$

$$\bar{f}_l(\mathbf{W}_t) = \bar{f}_l(\mathbf{W}_t) - \bar{f}_l(\mathbf{W}^*)$$

$$\leq \mathbb{E}\Big[\sum_{k=1}^{K} \frac{\partial(\bar{f}_l(\mathbf{W}_t))}{\partial \tilde{\boldsymbol{w}}_k})^\top (\boldsymbol{w}_{t(k)} - \boldsymbol{w}_k^*)\Big]$$

$$\leq \|\mathbf{W}_t - \mathbf{W}^* \mathbf{P}^*\|(\|\boldsymbol{\mu}_l\| + \|\mathbf{\Sigma}_l\|^{\frac{1}{2}})$$

$$\lesssim O\Big(\frac{\sum_{j=1}^{L} \sqrt{\lambda_l}(\|\boldsymbol{\mu}_j\| + \|\mathbf{\Sigma}_j\|^{\frac{1}{2}})}{\sum_{j=1}^{L} \lambda_j \|\mathbf{\Sigma}_j^{-1}\|^{-1} \rho_j}(\|\boldsymbol{\mu}_j\| + \|\mathbf{\Sigma}_j\|^{\frac{1}{2}}) \cdot \sqrt{\frac{d \log n}{n}} \eta \kappa^2 K^2 (1 + \xi)\Big) \tag{115}$$

$$\lesssim \text{poly}(\eta, \kappa, \tau, \delta_K(\mathbf{W}^*)) \sqrt{\frac{d \log n}{n}} K^2 (1 + \xi) \cdot O(\frac{1}{1 + \|\mathbf{\Sigma}_l\|^3})$$

$$\lesssim \text{poly}(\eta, \kappa, \tau, \delta_K(\mathbf{W}^*)) \sqrt{\frac{d \log n}{n}} K^2 (1 + \xi) \cdot O(1) - \Theta(\|\mathbf{\Sigma}_l\|^3),$$

The first inequality of (115) is by the Mean Value Theorem. The second inequality of (115) is from Property 8, and the third inequality is derived from (96, 97). The last inequality is obtained by the condition that $\|\mathbf{\Sigma}_l\| = o(1)$. We can similarly have

$$\bar{f}(\mathbf{W}_t) \leq \mathbb{E}\Big[\sum_{k=1}^{K} \frac{\partial(\bar{f}(\mathbf{W}_t))}{\partial \tilde{\boldsymbol{w}}_k})^\top (\boldsymbol{w}_{t(k)} - \boldsymbol{w}_k^*)\Big]$$

$$\lesssim \text{poly}(\eta, \kappa, \tau, \delta_K(\mathbf{W}^*)) \sqrt{\frac{d \log n}{n}} K^2 (1 + \xi) \cdot O(\frac{1}{1 + \|\mathbf{\Sigma}_l\|^3}) \tag{116}$$

$$\lesssim \text{poly}(\eta, \kappa, \tau, \delta_K(\mathbf{W}^*)) \sqrt{\frac{d \log n}{n}} K^2 (1 + \xi) \cdot O(1) - \Theta(\|\mathbf{\Sigma}_l\|^3)$$

If $\|\mathbf{\Sigma}_l\|^{\frac{1}{2}} = \Omega(1)$, we have

$$n_{sc} \lesssim \text{poly}(\epsilon_0^{-1}, \eta, \tau, \kappa, K, \delta_1(\mathbf{W}^*)) \cdot d \log^2 d \cdot O(\|\mathbf{\Sigma}_l\|^3) \tag{117}$$

$$v(\Psi) \leq 1 - \frac{1}{K^2 \tau^K \eta \kappa^2} \Theta(\frac{1}{1 + \|\mathbf{\Sigma}_l\|}) \tag{118}$$

$$\|\widehat{\mathbf{W}}_n - \mathbf{W}^* \mathbf{P}^*\|_F \lesssim \text{poly}(\eta, \tau, \kappa, \delta_K(\mathbf{W}^*)) \sqrt{\frac{d \log n}{n}} K^{\frac{5}{2}} (1 + \xi) \cdot \sqrt{\|\mathbf{\Sigma}_l\|} \tag{119}$$

$$\bar{f}_l(\mathbf{W}_t) \lesssim \text{poly}(\eta, \tau, \kappa, \delta_K(\mathbf{W}^*)) \sqrt{\frac{d \log n}{n}} K^2 (1 + \xi) \cdot \|\mathbf{\Sigma}_l\| \tag{120}$$

$$\bar{f}(\mathbf{W}_t) \lesssim \text{poly}(\eta, \tau, \kappa, \delta_K(\mathbf{W}^*)) \sqrt{\frac{d \log n}{n}} K^2 (1 + \xi) \cdot \|\mathbf{\Sigma}_l\| \tag{121}$$

When everything is fixed except $\|\boldsymbol{\mu}_l\|$, by combining (94) and (101), we have

$$n_{sc} \lesssim \text{poly}(\epsilon_0^{-1}, \eta, \tau, \kappa, K, \delta_1(\boldsymbol{W}^*)) \cdot d\log^2 d \cdot \begin{cases} O(\|\boldsymbol{\mu}_l\|^4), & \text{if } \|\boldsymbol{\mu}_l\| \leq 1 \\ O(\|\boldsymbol{\mu}_l\|^{12}), & \text{if } \|\boldsymbol{\mu}_l\| \geq 1 \end{cases} \tag{122}$$

$$v(\Psi) \leq 1 - \frac{1}{K^2\tau^K\eta\kappa^2}\Theta(\frac{1}{1+\|\boldsymbol{\mu}_l\|^2}) \tag{123}$$

$$\|\widehat{\boldsymbol{W}}_n - \boldsymbol{W}^*\boldsymbol{P}^*\|_F \lesssim \text{poly}(\eta, \tau, \kappa, \delta_K(\boldsymbol{W}^*))\sqrt{\frac{d\log n}{n}}K^{\frac{5}{2}}(1+\xi) \cdot (1+\|\boldsymbol{\mu}_l\|) \tag{124}$$

$$\bar{f}_l(\boldsymbol{W}_t) \lesssim \text{poly}(\eta, \tau, \kappa, \delta_K(\boldsymbol{W}^*))\sqrt{\frac{d\log n}{n}}K^2(1+\xi) \cdot (1+\|\boldsymbol{\mu}_l\|^2) \tag{125}$$

$$\bar{f}(\boldsymbol{W}_t) \lesssim \text{poly}(\eta, \tau, \kappa, \delta_K(\boldsymbol{W}^*))\sqrt{\frac{d\log n}{n}}K^2(1+\xi) \cdot (1+\|\boldsymbol{\mu}_l\|^2) \tag{126}$$

When everything else is fixed except $\lambda_1, \lambda_2, \cdots, \lambda_L$, where $\|\boldsymbol{\Sigma}_j\| = \Omega(1)$, $j \in [L]$ and $\|\boldsymbol{\mu}_j\| = \|\boldsymbol{\mu}_i\|$, $i, j \in [L]$, if $\|\boldsymbol{\Sigma}_l\| \leq \|\boldsymbol{\Sigma}_j\|$, $j \in [L]$, we have

$$\begin{aligned} n_{sc} &\lesssim \text{poly}(\epsilon_0^{-1}, \eta, \kappa, K, \delta_1(\boldsymbol{W}^*)) \cdot d\log^2 d \cdot \frac{(a_1\lambda_l^2 + a_2\lambda_l^{\frac{3}{2}} + a_3\lambda_l + a_4\lambda_l^{\frac{1}{2}} + a_5)}{(\sum_{j=1}^L \lambda_j\rho_j)^2} \\ &\leq \text{poly}(\epsilon_0^{-1}, \eta, \kappa, K, \delta_1(\boldsymbol{W}^*)) \cdot d\log^2 d \cdot \frac{a_5}{(\sum_{j=1}^L \lambda_j\rho_j)^2} \\ &\lesssim \text{poly}(\epsilon_0^{-1}, \eta, \kappa, K, \delta_1(\boldsymbol{W}^*)) \cdot d\log^2 d \cdot O((1+\lambda_l)^{-2}) \end{aligned} \tag{127}$$

where $a_1 = (\|\boldsymbol{\mu}_l\| + \|\boldsymbol{\Sigma}_l\|^{\frac{1}{2}})^{12}/\|\boldsymbol{\Sigma}_l\|^3$, $a_2 = (\|\boldsymbol{\mu}_l\| + \|\boldsymbol{\Sigma}_l^{\frac{1}{2}}\|)^8(\sum_{j\neq l}\lambda_j(\|\boldsymbol{\mu}_j\| + \|\boldsymbol{\Sigma}_j\|^{\frac{1}{2}})^8)^{\frac{1}{2}}/\|\boldsymbol{\Sigma}_l\|^3$, $a_3 = (\|\boldsymbol{\mu}_l\|/\|\boldsymbol{\Sigma}_l\|^{\frac{1}{2}} + 1)^6(\sum_{j\neq l}\lambda_j(\|\boldsymbol{\mu}_j\| + \|\boldsymbol{\Sigma}_j\|^{\frac{1}{2}})^4\sum_{j\neq l}\lambda_j(\|\boldsymbol{\mu}_j\| + \|\boldsymbol{\Sigma}_j\|^{\frac{1}{2}})^8)^{\frac{1}{2}} + (\|\boldsymbol{\mu}_l\| + \|\boldsymbol{\Sigma}_l\|^{\frac{1}{2}})^6\sum_{j\neq l}\lambda_j(\|\boldsymbol{\mu}_j\|/\|\boldsymbol{\Sigma}_j\|^{\frac{1}{2}} + 1)^6$, $a_4 = \sum_{j\neq l}\lambda_j(\|\boldsymbol{\mu}_j\|/\|\boldsymbol{\Sigma}_j\|^{\frac{1}{2}} + 1)^6(\|\boldsymbol{\mu}_l\| + \|\boldsymbol{\Sigma}_l\|^{\frac{1}{2}})^2(\sum_{j\neq l}\lambda_j(\|\boldsymbol{\mu}_j\| + \|\boldsymbol{\Sigma}_j\|^{\frac{1}{2}})^8)^{\frac{1}{2}}$, $a_5 = (\sum_{j\neq l}\lambda_j(\|\boldsymbol{\mu}_j\| + \|\boldsymbol{\Sigma}_j\|^{\frac{1}{2}})^4\sum_{j\neq l}\lambda_j(\|\boldsymbol{\mu}_j\| + \|\boldsymbol{\Sigma}_j\|^{\frac{1}{2}})^8)^{\frac{1}{2}} \cdot \sum_{j\neq l}\lambda_j(\|\boldsymbol{\mu}_j\|/\|\boldsymbol{\Sigma}_j\|^{\frac{1}{2}} + 1)^6$. The second step of (127) is by $a_i = O(a_5)$, $i = 1, 2, 3, 4$.

$$v \leq \frac{1}{K^2\eta\tau^K\kappa^2}\Theta(\frac{1}{1+\lambda_l}) \tag{128}$$

$$\|\widehat{\boldsymbol{W}}_n - \boldsymbol{W}^*\boldsymbol{P}\|_F \leq \text{poly}(\eta, \kappa, , \tau, \delta_1(\boldsymbol{W}^*)) \cdot \sqrt{\frac{d\log n}{n}}K^{\frac{5}{2}}(1+\xi) \cdot O(\frac{1}{1+\sqrt{\lambda_l}}) \tag{129}$$

$$\bar{f}_l(\boldsymbol{W}_t) \leq \text{poly}(\eta, \kappa, , \tau, \delta_1(\boldsymbol{W}^*)) \cdot \sqrt{\frac{d\log n}{n}}K^2(1+\xi) \cdot O(\frac{1}{1+\sqrt{\lambda_l}}) \tag{130}$$

$$\bar{f}(\boldsymbol{W}_t) \leq \text{poly}(\eta, \kappa, , \tau, \delta_1(\boldsymbol{W}^*)) \cdot \sqrt{\frac{d\log n}{n}}K^2(1+\xi) \cdot O(\frac{1}{1+\lambda_l}) \tag{131}$$

If $\|\boldsymbol{\Sigma}_l\| \geq \|\boldsymbol{\Sigma}_j\|$, $j \in [L]$, we can similarly derive that

$$\begin{aligned} n_{sc} &\lesssim \text{poly}(\epsilon_0^{-1}, \eta, \kappa, K, \delta_1(\boldsymbol{W}^*)) \cdot d\log^2 d \cdot \frac{(a_1\lambda_l^2 + a_2\lambda_l^{\frac{3}{2}} + a_3\lambda_l + a_4\lambda_l^{\frac{1}{2}} + a_5)}{(\sum_{j=1}^L \lambda_j\rho_j)^2} \\ &\lesssim \text{poly}(\epsilon_0^{-1}, \eta, \kappa, K, \delta_1(\boldsymbol{W}^*)) \cdot d\log^2 d \cdot (O(1) - \Theta((1+\lambda_l)^{-2})) \end{aligned} \tag{132}$$

$$v \leq 1 - \frac{1}{K^2\eta\tau^K\kappa^2}\Theta(\frac{1}{1+\lambda_l}) \tag{133}$$

$$\|\widehat{\boldsymbol{W}}_n - \boldsymbol{W}^*\boldsymbol{P}\|_F \leq \text{poly}(\eta, \kappa, , \tau, \delta_1(\boldsymbol{W}^*)) \cdot \sqrt{\frac{d\log n}{n}}K^{\frac{5}{2}}(1+\xi) \cdot O(1+\sqrt{\lambda_l}) \tag{134}$$

$$\bar{f}_l(\boldsymbol{W}_t) \leq \text{poly}(\eta, \kappa, , \tau, \delta_1(\boldsymbol{W}^*)) \cdot \sqrt{\frac{d\log n}{n}}K^2(1+\xi) \cdot O(1+\sqrt{\lambda_l}) \tag{135}$$

$$\bar{f}(\boldsymbol{W}_t) \leq \text{poly}(\eta, \kappa, , \tau, \delta_1(\boldsymbol{W}^*)) \cdot \sqrt{\frac{d\log n}{n}}K^2(1+\xi) \cdot (O(1) - \frac{\Theta(1)}{1+\lambda_l}) \tag{136}$$

# E  PROOF OF LEMMA 1

We first state some important lemmas used in proof in Section E.1 and describe the proof in Section E.2. The proofs of these lemmas are provided in Section E.3 to E.7 in sequence. The proof idea mainly follows from Fu et al. (2020). Lemma 6 shows the Hessian $\nabla^2 \bar{f}(\boldsymbol{W})$ of the population risk function is smooth. Lemma 7 illustrates that $\nabla^2 \bar{f}(\boldsymbol{W})$ is strongly convex in the neighborhood around $\boldsymbol{\mu}^*$. Lemma 8 shows the Hessian of the empirical risk function $\nabla^2 f_n(\boldsymbol{W}^*)$ is close to its population risk $\nabla^2 \bar{f}(\boldsymbol{W}^*)$ in the local convex region. Summing up these three lemmas, we can derive the proof of Lemma 1. Lemma 4 is used in the proof of Lemma 7. Lemma 5 is used in the proof of Lemma 8.

The analysis of the Hessian matrix of the population loss in Fu et al. (2020) and Zhong et al. (2017b) can not be extended to the Gaussian mixture model. To solve this problem, we develop new tools using some good properties of symmetric distribution and even function. Our approach can also be applied to other activations like tanh or erf. Moreover, if we directly apply the existing matrix concentration inequalities in these works in bounding the error between the empirical loss and the population loss, the resulting sample complexity bound is loose and cannot reflect the influence of each component of the Gaussian mixture distribution. We develop a new version of Bernstein's inequality (see (208)) so that the final bound is $O(d \log^2 d)$.

Mei et al. (2016) showed that the landscape of the empirical risk is close to that of the population risk when the number of samples is sufficiently large for the special case that $K = 1$. Focusing on Gaussian mixture models, our result explicitly shows how the parameters of the input distribution, including the proportion, mean and, variance of each component will affect the error bound between the empirical loss and the population loss in Lemma 8.

## E.1  USEFUL LEMMAS IN THE PROOF OF LEMMA 1

**Lemma 4.**

$$\mathbb{E}_{\boldsymbol{x} \sim \frac{1}{2}\mathcal{N}(\boldsymbol{\mu}, \boldsymbol{I}_d) + \frac{1}{2}\mathcal{N}(-\boldsymbol{\mu}, \boldsymbol{I}_d)} \left[ \left( \sum_{i=1}^{k} \boldsymbol{r}_i^\top \boldsymbol{x} \cdot \phi'(\sigma \cdot x_i) \right)^2 \right] \geq \rho(\boldsymbol{\mu}, \sigma) \|\boldsymbol{R}\|_F^2 , \tag{137}$$

*where $\rho(\boldsymbol{\mu}, \sigma)$ is defined in Definition 3 and $\boldsymbol{R} = (\boldsymbol{r}_1, \cdots, \boldsymbol{r}_k) \in \mathbb{R}^{d \times k}$ is an arbitrary matrix.*

**Lemma 5.** *With the FCN model (1) and the Gaussian Mixture Model, for any permutation matrix $\boldsymbol{P}$, for some constant $C_{12} > 0$, we have we have*

$$\mathbb{E}_{\boldsymbol{x} \sim \sum_{l=1}^{L} \lambda_l \mathcal{N}(\boldsymbol{\mu}_l, \boldsymbol{\Sigma}_l)} \left[ \sup_{\boldsymbol{W} \neq \boldsymbol{W}' \in \mathbb{B}(\boldsymbol{W}^* \boldsymbol{P}, r)} \frac{\|\nabla^2 \ell(\boldsymbol{W}, \boldsymbol{x}) - \nabla^2 \ell(\boldsymbol{W}', \boldsymbol{x})\|}{\|\boldsymbol{W} - \boldsymbol{W}'\|_F} \right]$$
$$\leq C_{12} \cdot d^{\frac{3}{2}} K^{\frac{5}{2}} \sqrt{\sum_{l=1}^{L} \lambda_l (\|\boldsymbol{\mu}_l\|_\infty + \|\boldsymbol{\Sigma}_l\|)^2 \sum_{l=1}^{L} \lambda_l (\|\boldsymbol{\mu}_l\|_\infty + \|\boldsymbol{\Sigma}_l\|)^4} \tag{138}$$

**Lemma 6.** *(Hessian smoothness of population loss) In the FCN model (1), for some constant $C_5 > 0$, for any permutation matrix $\boldsymbol{P}$, we have*

$$\|\nabla^2 \bar{f}(\boldsymbol{W}) - \nabla^2 \bar{f}(\boldsymbol{W}^* \boldsymbol{P})\|$$
$$\leq C_5 \cdot K^{\frac{3}{2}} \cdot \left( \sum_{l=1}^{L} \lambda_l (\|\boldsymbol{\mu}_l\| + \|\boldsymbol{\Sigma}_l^{\frac{1}{2}}\|)^4 \sum_{l=1}^{L} \lambda_l (\|\boldsymbol{\mu}_l\| + \|\boldsymbol{\Sigma}_l^{\frac{1}{2}}\|)^8 \right)^{\frac{1}{4}} \cdot \|\boldsymbol{W} - \boldsymbol{W}^* \boldsymbol{P}\|_F \tag{139}$$

**Lemma 7.** *(Local strong convexity of population loss) In the FCN model* (1)*, for any permutation matrix $\boldsymbol{P}$, if $\|\boldsymbol{W} - \boldsymbol{W}^* \boldsymbol{P}\|_F \leq r$ for an $\epsilon_0 \in (0, \frac{1}{4})$, then for some constant $C_4 > 0$,*

$$\frac{4(1 - \epsilon_0)}{K^2} \sum_{l=1}^{L} \lambda_l \frac{\|\boldsymbol{\Sigma}_l^{-1}\|^{-1}}{\eta \tau^K \kappa^2} \rho\left( \frac{\boldsymbol{W}^{*\top} \boldsymbol{\mu}_l}{\delta_K(\boldsymbol{W}^*) \|\boldsymbol{\Sigma}_l^{-1}\|^{-\frac{1}{2}}}, \delta_K(\boldsymbol{W}^*) \|\boldsymbol{\Sigma}_l^{-1}\|^{-\frac{1}{2}} \right) \cdot \boldsymbol{I}_{dK}$$

$$\preceq \nabla^2 \bar{f}(\boldsymbol{W}) \preceq C_4 \cdot \sum_{l=1}^{L} \lambda_l (\|\boldsymbol{\mu}_l\| + \boldsymbol{\Sigma}_l^{\frac{1}{2}})^2 \cdot \boldsymbol{I}_{dK} \tag{140}$$

**Lemma 8.** *In the FCN model* (1), *for any permutation matrix $\boldsymbol{P}$, as long as $n \geq C' \cdot dK \log dK$ for some constant $C' > 0$, we have*

$$\sup_{\boldsymbol{W} \in \mathbb{B}(\boldsymbol{W}^*\boldsymbol{P}, r)} ||\nabla^2 f_n(\boldsymbol{W}) - \nabla^2 \bar{f}(\boldsymbol{W})|| \leq C_6 \cdot \sum_{l=1}^{L} \lambda_l (||\boldsymbol{\mu}_l|| + ||\boldsymbol{\Sigma}_l^{\frac{1}{2}}||)^2 \sqrt{\frac{dK \log n}{n}}) \quad (141)$$

*with probability at least $1 - d^{-10}$ for some constant $C_6 > 0$.*

### E.2 PROOF OF LEMMA 1

From Lemma 7 and 8, with probability at least $1 - d^{-10}$,

$$\nabla^2 f_n(\boldsymbol{W}) \succeq \nabla^2 \bar{f}(\boldsymbol{W}) - ||\nabla^2 \bar{f}(\boldsymbol{W}) - \nabla^2 f_n(\boldsymbol{W})|| \cdot \boldsymbol{I}$$

$$\succeq \Omega\Big(\frac{(1 - \epsilon_0)}{K^2} \sum_{l=1}^{L} \lambda_l \frac{||\boldsymbol{\Sigma}_l^{-1}||^{-1}}{\eta \tau^K \kappa^2} \rho\big(\frac{\boldsymbol{W}^{*\top}\boldsymbol{\mu}_l}{\delta_K(\boldsymbol{W}^*)||\boldsymbol{\Sigma}_l^{-1}||^{-\frac{1}{2}}}, \delta_K(\boldsymbol{W}^*)||\boldsymbol{\Sigma}_l^{-1}||^{-\frac{1}{2}}\big)\Big) \cdot \boldsymbol{I}$$

$$- O\Big(C_6 \cdot \sum_{l=1}^{L} \lambda_l (||\boldsymbol{\mu}_l|| + ||\boldsymbol{\Sigma}_l^{\frac{1}{2}}||)^2 \sqrt{\frac{dK \log n}{n}}\Big) \cdot \boldsymbol{I} \tag{142}$$

As long as the sample complexity is set to satisfy

$$C_6 \cdot \sum_{l=1}^{L} \lambda_l (||\boldsymbol{\mu}_l|| + ||\boldsymbol{\Sigma}_l^{\frac{1}{2}}||)^2 \sqrt{\frac{dK \log n}{n}}$$

$$\leq \frac{\epsilon_0}{K^2} \sum_{l=1}^{L} \lambda_l \frac{||\boldsymbol{\Sigma}_l^{-1}||^{-1}}{\eta \tau^K \kappa^2} \rho\big(\frac{\boldsymbol{W}^{*\top}\boldsymbol{\mu}_l}{\delta_K(\boldsymbol{W}^*)||\boldsymbol{\Sigma}_l^{-1}||^{-\frac{1}{2}}}, \delta_K(\boldsymbol{W}^*)||\boldsymbol{\Sigma}_l^{-1}||^{-\frac{1}{2}}\big) \cdot \boldsymbol{I} \tag{143}$$

i.e.,

$$n \geq C_1 \epsilon_0^{-2} \cdot \Big(\sum_{l=1}^{L} \lambda_l (||\boldsymbol{\mu}_l|| + ||\boldsymbol{\Sigma}_l^{\frac{1}{2}}||)^2\Big)^2$$

$$\cdot \Big(\sum_{l=1}^{L} \lambda_l \frac{||\boldsymbol{\Sigma}_l^{-1}||^{-1}}{\eta \tau^K \kappa^2} \rho\big(\frac{\boldsymbol{W}^{*\top}\boldsymbol{\mu}_l}{\delta_K(\boldsymbol{W}^*)||\boldsymbol{\Sigma}_l^{-1}||^{-\frac{1}{2}}}, \delta_K(\boldsymbol{W}^*)||\boldsymbol{\Sigma}_l^{-1}||^{-\frac{1}{2}}\big) \cdot \boldsymbol{I}\Big)^{-2} dK^5 \log^2 d \tag{144}$$

for some constant $C_1 > 0$, then we have the lower bound of the Hessian with probability at least $1 - d^{-10}$.

$$\nabla^2 f_n(\boldsymbol{W}) \succeq \Omega\Big(\frac{1 - 2\epsilon_0}{K^2} \sum_{l=1}^{L} \lambda_l \frac{||\boldsymbol{\Sigma}_l^{-1}||^{-1}}{\eta \tau^K \kappa^2} \rho\big(\frac{\boldsymbol{W}^{*\top}\boldsymbol{\mu}_l}{\delta_K(\boldsymbol{W}^*)||\boldsymbol{\Sigma}_l^{-1}||^{-\frac{1}{2}}}, \delta_K(\boldsymbol{W}^*)||\boldsymbol{\Sigma}_l^{-1}||^{-\frac{1}{2}}\big)\Big) \cdot \boldsymbol{I} \tag{145}$$

By (140) and (141), we can also derive the upper bound as follows,

$$||\nabla^2 f_n(\boldsymbol{W})|| \leq ||\nabla^2 \bar{f}(\boldsymbol{W})|| + ||\nabla^2 f_n(\boldsymbol{W}) - \nabla^2 \bar{f}(\boldsymbol{W})||$$

$$\leq C_4 \cdot \sum_{l=1}^{L} \lambda_l (||\boldsymbol{\mu}_l|| + ||\boldsymbol{\Sigma}_l^{\frac{1}{2}}||)^2 + C_6 \cdot \sum_{1=1}^{L} \lambda_l (||\boldsymbol{\mu}_l|| + ||\boldsymbol{\Sigma}_l^{\frac{1}{2}}||)^2 \sqrt{\frac{dK \log n}{n}}$$

$$\leq C_2 \cdot \sum_{l=1}^{L} \lambda_l (||\boldsymbol{\mu}_l|| + ||\boldsymbol{\Sigma}_l^{\frac{1}{2}}||)^2 \tag{146}$$

for some constant $C_2 > 0$. Combining (145) and (146), we have

$$\Omega\Big(\frac{1 - 2\epsilon_0}{K^2} \sum_{l=1}^{L} \lambda_l \frac{||\boldsymbol{\Sigma}_l^{-1}||^{-1}}{\eta \tau^K \kappa^2} \rho\big(\frac{\boldsymbol{W}^{*\top}\boldsymbol{\mu}_l}{\delta_K(\boldsymbol{W}^*)||\boldsymbol{\Sigma}_l^{-1}||^{-\frac{1}{2}}}, \delta_K(\boldsymbol{W}^*)||\boldsymbol{\Sigma}_l^{-1}||^{-\frac{1}{2}}\big)\Big) \cdot \boldsymbol{I}$$

$$\preceq \nabla^2 f_n(\boldsymbol{W}) \preceq C_2 \sum_{l=1}^{L} \lambda_l (||\tilde{\boldsymbol{\mu}}_l||_\infty + ||\boldsymbol{\Sigma}_l^{\frac{1}{2}}||)^2 \cdot \boldsymbol{I} \tag{147}$$

with probability at least $1 - d^{-10}$.

### E.3 PROOF OF LEMMA 4

Following the proof idea in Lemma D.4 of Zhong et al. (2017b), we have

$$\mathbb{E}_{\boldsymbol{x} \sim \frac{1}{2}\mathcal{N}(\boldsymbol{\mu}, \boldsymbol{I}_d) + \frac{1}{2}\mathcal{N}(-\boldsymbol{\mu}, \boldsymbol{I}_d)} \left[ \left( \sum_{i=1}^{k} \boldsymbol{r}_i^\top \boldsymbol{x} \cdot \phi'(\sigma \cdot x_i) \right)^2 \right] = A_0 + B_0 \tag{148}$$

$$A_0 = \mathbb{E}_{\boldsymbol{x} \sim \frac{1}{2}\mathcal{N}(\boldsymbol{\mu}, \boldsymbol{I}_d) + \frac{1}{2}\mathcal{N}(-\boldsymbol{\mu}, \boldsymbol{I}_d)} \left( \sum_{i=1}^{k} \boldsymbol{r}_i^\top \boldsymbol{x} \cdot \phi'^2(\sigma \cdot x_i) \cdot \boldsymbol{x} \boldsymbol{x}^\top \boldsymbol{r}_i \right) \tag{149}$$

$$B_0 = \mathbb{E}_{\boldsymbol{x} \sim \frac{1}{2}\mathcal{N}(\boldsymbol{\mu}, \boldsymbol{I}_d) + \frac{1}{2}\mathcal{N}(-\boldsymbol{\mu}, \boldsymbol{I}_d)} \left( \sum_{i \neq l} \boldsymbol{r}_i^\top \phi'(\sigma \cdot x_i) \phi'(\sigma \cdot x_l) \cdot \boldsymbol{x} \boldsymbol{x}^\top \boldsymbol{r}_l \right) \tag{150}$$

In $A_0$, we know that $\mathbb{E}_{\boldsymbol{x} \sim \frac{1}{2}\mathcal{N}(\boldsymbol{\mu}, \boldsymbol{I}_d) + \frac{1}{2}\mathcal{N}(-\boldsymbol{\mu}, \boldsymbol{I}_d)} x_j = 0$. Therefore, by some basic mathematical computation,

$$
\begin{aligned}
A_0 &= \sum_{i=1}^{k} \mathbb{E}_{\boldsymbol{x} \sim \frac{1}{2}\mathcal{N}(\boldsymbol{\mu}, \boldsymbol{I}_d) + \frac{1}{2}\mathcal{N}(-\boldsymbol{\mu}, \boldsymbol{I}_d)} \Big[ \boldsymbol{r}_i^\top \Big( \phi'^2(\sigma \cdot x_i) \Big( x_i^2 \boldsymbol{e}_i \boldsymbol{e}_i^\top + \sum_{j \neq i} x_i x_j (\boldsymbol{e}_i \boldsymbol{e}_j^\top \\
&\quad + \boldsymbol{e}_j \boldsymbol{e}_i^\top) + \sum_{j \neq i} \sum_{l \neq i} x_j x_l \boldsymbol{e}_j \boldsymbol{e}_l^\top \Big) \Big) \boldsymbol{r}_i \Big] \\
&= \sum_{i=1}^{k} \mathbb{E}_{\boldsymbol{x} \sim \frac{1}{2}\mathcal{N}(\boldsymbol{\mu}, \boldsymbol{I}_d) + \frac{1}{2}\mathcal{N}(-\boldsymbol{\mu}, \boldsymbol{I}_d)} \Big[ \boldsymbol{r}_i^\top \Big( \phi'^2(\sigma \cdot x_i) \Big( x_i^2 \boldsymbol{e}_i \boldsymbol{e}_i^\top + \sum_{j \neq i} x_j^2 \boldsymbol{e}_j \boldsymbol{e}_j^\top \Big) \Big) \boldsymbol{r}_i \Big] \\
&= \sum_{i=1}^{k} \Big[ \mathbb{E}_{\boldsymbol{x} \sim \frac{1}{2}\mathcal{N}(\boldsymbol{\mu}, \boldsymbol{I}_d) + \frac{1}{2}\mathcal{N}(-\boldsymbol{\mu}, \boldsymbol{I}_d)} [\phi'^2(\sigma \cdot x_i) x_i^2] \boldsymbol{r}_i^\top \boldsymbol{e}_i \boldsymbol{e}_i^\top \boldsymbol{r}_i \\
&\quad + \sum_{j \neq i} \mathbb{E}_{\boldsymbol{x} \sim \frac{1}{2}\mathcal{N}(\boldsymbol{\mu}, \boldsymbol{I}_d) + \frac{1}{2}\mathcal{N}(-\boldsymbol{\mu}, \boldsymbol{I}_d)} [x_j^2] \mathbb{E}_{\boldsymbol{x} \sim \frac{1}{2}\mathcal{N}(\boldsymbol{\mu}, \boldsymbol{I}) + \frac{1}{2}\mathcal{N}(-\boldsymbol{\mu}, \boldsymbol{I})} [\phi'^2(\sigma \cdot x_i)] \boldsymbol{r}_i^\top \boldsymbol{e}_j \boldsymbol{e}_j^\top \boldsymbol{r}_i \Big] \\
&= \sum_{i=1}^{k} r_{ii}^2 \beta_2(i, \boldsymbol{\mu}, \sigma) + \sum_{i=1}^{k} \sum_{j \neq i} r_{ij}^2 \beta_0(i, \boldsymbol{\mu}, \sigma)(1 + \mu_j^2)
\end{aligned}
\tag{151}
$$

In $B_0$, $\alpha_1(i, \boldsymbol{\mu}, \sigma) = \mathbb{E}_{\boldsymbol{x} \sim \frac{1}{2}\mathcal{N}(\boldsymbol{\mu}, \boldsymbol{I}_d) + \frac{1}{2}\mathcal{N}(-\boldsymbol{\mu}, \boldsymbol{I}_d)} (x_i \phi'(x_i)) = 0$. By the equation in Page 30 of Zhong et al. (2017b), we have

$$
\begin{aligned}
B_0 &= \sum_{i \neq l}^{k} \mathbb{E}_{\boldsymbol{x} \sim \frac{1}{2}\mathcal{N}(\boldsymbol{\mu}, \boldsymbol{I}_d) + \frac{1}{2}\mathcal{N}(-\boldsymbol{\mu}, \boldsymbol{I}_d)} \Big[ \boldsymbol{r}_i^\top \Big( \phi'(\sigma \cdot x_i) \phi'(\sigma \cdot x_l) \Big( x_i^2 \boldsymbol{e}_i \boldsymbol{e}_i^\top + x_l^2 \boldsymbol{e}_l \boldsymbol{e}_l^\top + x_i x_l (\boldsymbol{e}_i \boldsymbol{e}_l^\top + \\
&\quad \boldsymbol{e}_l \boldsymbol{e}_i^\top) + \sum_{j \neq i} x_j x_l \boldsymbol{e}_j \boldsymbol{e}_l^\top + \sum_{j \neq l} x_j x_i \boldsymbol{e}_j \boldsymbol{e}_i^\top + \sum_{j \neq i, l} \sum_{j' \neq i, l} x_j x_{j'} \boldsymbol{e}_j \boldsymbol{e}_{j'}^\top \Big) \Big) \boldsymbol{r}_l \Big] \\
&= \sum_{i \neq l} r_{ii} r_{li} \alpha_2(i, \boldsymbol{\mu}, \sigma) \alpha_0(l, \boldsymbol{\mu}, \sigma) + \sum_{i \neq l} r_{ij} r_{lj} \alpha_0(i, \boldsymbol{\mu}, \sigma) \alpha_0(l, \boldsymbol{\mu}, \sigma)(1 + \mu_j^2)
\end{aligned}
\tag{152}
$$

Therefore,

$$
\begin{aligned}
A_0 + B_0 &= \sum_{i=1}^{k} \left( r_{ii} \frac{\alpha_2(i, \boldsymbol{\mu}, \sigma)}{\sqrt{1+\mu_i^2}} + \sum_{l \neq i} r_{li} \alpha_0(l, \boldsymbol{\mu}, \sigma) \sqrt{1+\mu_i^2} \right)^2 - \sum_{i=1}^{k} r_{ii}^2 \frac{\alpha_2^2(i, \boldsymbol{\mu}, \sigma)}{1+\mu_i^2} \\
&\quad - \sum_{i=1}^{k} \sum_{l \neq i} r_{li}^2 \alpha_0(l, \boldsymbol{\mu}, \sigma)^2 (1+\mu_i^2) + \sum_{i=1}^{k} r_{ii}^2 \beta_2(i, \boldsymbol{\mu}, \sigma) + \sum_{i=1}^{k} \sum_{j \neq i} r_{ij}^2 \beta_0(i, \boldsymbol{\mu}, \sigma)(1+\mu_j^2) \\
&\geq \sum_{i=1}^{k} r_{ii}^2 \left( \beta_2(i, \boldsymbol{\mu}, \sigma) - \frac{\alpha_2^2(i, \boldsymbol{\mu}, \sigma)}{1+\mu_i^2} \right) + \sum_{i=1}^{k} \sum_{j \neq i} r_{ij}^2 \left( \beta_0(i, \boldsymbol{\mu}, \sigma) - \alpha_0^2(i, \boldsymbol{\mu}, \sigma) \right)(1+\mu_j^2) \\
&\geq \rho(\boldsymbol{\mu}, \sigma) ||\boldsymbol{R}||_F^2
\end{aligned}
\tag{153}
$$

### E.4 PROOF OF LEMMA 5

Following the equation (92) in Lemma 8 of Fu et al. (2020) and by (85)

$$
||\nabla^2 \ell(\boldsymbol{W}) - \nabla^2 \ell(\boldsymbol{W}')|| \leq \sum_{j=1}^{K} \sum_{l=1}^{K} |\xi_{j,l}(\boldsymbol{W}) - \xi_{j,l}(\boldsymbol{W}')| \cdot ||\boldsymbol{x}\boldsymbol{x}^\top||
\tag{154}
$$

By Lagrange's inequality, we have

$$
|\xi_{j,l}(\boldsymbol{W}) - \xi_{j,l}(\boldsymbol{W}')| \leq (\max_k |T_{j,k,l}|) \cdot ||\boldsymbol{x}|| \cdot \sqrt{K} ||\boldsymbol{W} - \boldsymbol{W}'||_F
\tag{155}
$$

From Lemma 6, we know

$$
\max_k |T_{j,k,l}| \leq C_7
\tag{156}
$$

By Property 7, we have

$$
\mathbb{E}_{\boldsymbol{x} \sim \sum_{l=1}^{L} \lambda_l \mathcal{N}(\boldsymbol{\mu}_l, \boldsymbol{\Sigma}_l)}[||\boldsymbol{x}||^{2t}||] \leq d^t (2t-1)!! \sum_{l=1}^{L} \lambda_l (||\boldsymbol{\mu}_l||_\infty + ||\boldsymbol{\Sigma}_l||)^{2t}
\tag{157}
$$

Therefore, for some constant $C_{12} > 0$

$$
\begin{aligned}
&\mathbb{E}_{\boldsymbol{x} \sim \sum_{l=1}^{L} \lambda_l \mathcal{N}(\boldsymbol{\mu}_l, \boldsymbol{\Sigma}_l)} \left[ \sup_{\boldsymbol{W} \neq \boldsymbol{W}'} \frac{||\nabla^2 \ell(\boldsymbol{W}) - \nabla^2 \ell(\boldsymbol{W}')||}{||\boldsymbol{W} - \boldsymbol{W}'||_F} \right] \leq K^{\frac{5}{2}} \mathbb{E}[||\boldsymbol{x}||_2^3] \\
&\leq K^{\frac{5}{2}} \sqrt{d \sum_{l=1}^{L} \lambda_l (||\boldsymbol{\mu}||_\infty + ||\boldsymbol{\Sigma}_l||)^2} \sqrt{3d^2 \sum_{l=1}^{L} \lambda_l (||\boldsymbol{\mu}_l||_\infty + ||\boldsymbol{\Sigma}_l||)^4} \\
&= C_{12} \cdot d^{\frac{3}{2}} K^{\frac{5}{2}} \sqrt{\sum_{l=1}^{L} \lambda_l (||\boldsymbol{\mu}_l||_\infty + ||\boldsymbol{\Sigma}_l||)^2 \sum_{l=1}^{L} \lambda_l (||\boldsymbol{\mu}_l||_\infty + ||\boldsymbol{\Sigma}_l||)^4}
\end{aligned}
\tag{158}
$$

### E.5 PROOF OF LEMMA 6

Let $\boldsymbol{a} = (\boldsymbol{a}_1^\top, \cdots, \boldsymbol{a}_K^\top)^\top \in \mathbb{R}^{dK}$. Let $\Delta_{j,l} \in \mathbb{R}^{d \times d}$ be the $(j,l)$-th block of $\nabla^2 \bar{f}(\boldsymbol{W}) - \nabla^2 \bar{f}(\boldsymbol{W}^* \boldsymbol{P}) \in \mathbb{R}^{dK \times dK}$. By definition,

$$
||\nabla^2 \bar{f}(\boldsymbol{W}) - \nabla^2 \bar{f}(\boldsymbol{W}^* \boldsymbol{P})|| = \max_{||\boldsymbol{a}||=1} \sum_{j=1}^{K} \sum_{l=1}^{K} \boldsymbol{a}_j^\top \Delta_{j,l} \boldsymbol{a}_l
\tag{159}
$$

Denote $\boldsymbol{P} = (\boldsymbol{p}_1, \cdots, \boldsymbol{p}_K) \in \mathbb{R}^{K \times K}$. By the mean value theorem and (85),

$$
\begin{aligned}
\Delta_{j,l} = \frac{\partial^2 \bar{f}(\boldsymbol{W})}{\partial \boldsymbol{w}_j \partial \boldsymbol{w}_l} - \frac{\partial^2 \bar{f}(\boldsymbol{W}^*\boldsymbol{P})}{\partial \boldsymbol{w}_j^* \partial \boldsymbol{w}_l^*} &= \mathbb{E}_{\boldsymbol{x} \sim \sum_{l=1}^L \lambda_l \mathcal{N}(\boldsymbol{\mu}_l, \sigma_l^2 \boldsymbol{I}_d)}[(\xi_{j,l}(\boldsymbol{W}) - \xi_{j,l}(\boldsymbol{W}^*\boldsymbol{P})) \cdot \boldsymbol{x}\boldsymbol{x}^\top] \\
&= \mathbb{E}_{\boldsymbol{x} \sim \sum_{l=1}^L \lambda_l \mathcal{N}(\boldsymbol{\mu}_l, \boldsymbol{\Sigma}_l)}[\sum_{k=1}^K \left\langle \frac{\partial \xi_{j,l}(\boldsymbol{W}')}{\partial \boldsymbol{w}_k'}, \boldsymbol{w}_k - \boldsymbol{W}^*\boldsymbol{p}_k \right\rangle \cdot \boldsymbol{x}\boldsymbol{x}^\top] \\
&= \mathbb{E}_{\boldsymbol{x} \sim \sum_{l=1}^L \lambda_l \mathcal{N}(\boldsymbol{\mu}_l, \boldsymbol{\Sigma}_l)}[\sum_{k=1}^K \langle T_{j,l,k} \cdot \boldsymbol{x}, \boldsymbol{w}_k - \boldsymbol{W}^*\boldsymbol{p}_k \rangle \cdot \boldsymbol{x}\boldsymbol{x}^\top]
\end{aligned}
$$

(160)

where $\boldsymbol{W}' = \gamma \boldsymbol{W} + (1-\gamma)\boldsymbol{W}^*\boldsymbol{P}$ for some $\gamma \in (0,1)$ and $T_{j,l,k}$ is defined such that $\frac{\partial \xi_{j,l}(\boldsymbol{W}')}{\partial \boldsymbol{w}_k'} = T_{j,l,k} \cdot x \in \mathbb{R}^d$. Then we provide an upper bound for $\xi_{j,l}$. Since that $y = 1$ or $0$, we first compute the case in which $y = 1$. From (85) we can obtain

$$
\xi_{j,l}(\boldsymbol{W}) = \begin{cases} \frac{1}{K^2}\phi'(\boldsymbol{w}_j^\top \boldsymbol{x})\phi'(\boldsymbol{w}_l^\top \boldsymbol{x}) \cdot \frac{1}{H^2(\boldsymbol{W})}, & j \neq l \\ \frac{1}{K^2}\phi'(\boldsymbol{w}_j^\top \boldsymbol{x})\phi'(\boldsymbol{w}_l^\top \boldsymbol{x}) \cdot \frac{1}{H^2(\boldsymbol{W})} - \frac{1}{K}\phi''(\boldsymbol{w}_j^\top \boldsymbol{x}) \cdot \frac{1}{H(\boldsymbol{W})}, & j = l \end{cases}
$$

(161)

We can bound $\xi_{j,l}(\boldsymbol{W})$ by bounding each component of (161). Note that we have

$$
\frac{1}{K^2}\phi'(\boldsymbol{w}_j^\top \boldsymbol{x})\phi'(\boldsymbol{w}_l^\top \boldsymbol{x}) \cdot \frac{1}{H^2(\boldsymbol{W})} \leq \frac{1}{K^2}\frac{\phi(\boldsymbol{w}_j^\top \boldsymbol{x})\phi(\boldsymbol{w}_l^\top \boldsymbol{x})(1 - \phi(\boldsymbol{w}_j^\top \boldsymbol{x}))(1 - \phi(\boldsymbol{w}_l^\top \boldsymbol{x}))}{\frac{1}{K^2}\phi(\boldsymbol{w}_j^\top \boldsymbol{x})\phi(\boldsymbol{w}_l^\top \boldsymbol{x})} \leq 1
$$

(162)

$$
\frac{1}{K}\phi''(\boldsymbol{w}_j^\top \boldsymbol{x}) \cdot \frac{1}{H(\boldsymbol{W})} \leq \frac{1}{K}\frac{\phi(\boldsymbol{w}_j^\top \boldsymbol{x})(1 - \phi(\boldsymbol{w}_j^\top \boldsymbol{x}))(1 - 2\phi(\boldsymbol{w}_j^\top \boldsymbol{x}))}{\frac{1}{K}\phi(\boldsymbol{w}_j^\top \boldsymbol{x})} \leq 1
$$

(163)

where (162) holds for any $j, l \in [K]$. The case $y = 0$ can be computed with the same upper bound by substituting $(1 - H(\boldsymbol{W})) = \frac{1}{K}\sum_{j=1}^K (1 - \phi(\boldsymbol{w}_j^\top \boldsymbol{x}))$ for $H(\boldsymbol{W})$ in (161), (162) and (163). Therefore, there exists a constant $C_9 > 0$, such that

$$
|\xi_{j,l}(\boldsymbol{W})| \leq C_9
$$

(164)

We then need to calculate $T_{j,l,k}$. Following the analysis of $\xi_{j,l}(\boldsymbol{W})$, we only consider the case of $y = 1$ here for simplicity.

$$
T_{j,l,k} = \frac{-2}{K^3 H^3(\boldsymbol{W}')}\phi'(\boldsymbol{w}_j'^\top \boldsymbol{x})\phi'(\boldsymbol{w}_l'^\top \boldsymbol{x})\phi'(\boldsymbol{w}_k'^\top \boldsymbol{x}), \quad \text{where } j, l, k \text{ are not equal to each other}
$$

(165)

$$
T_{j,j,k} = \begin{cases} \frac{-2}{K^3 H^3(\boldsymbol{W}')}\phi'(\boldsymbol{w}_j'^\top \boldsymbol{x})\phi'(\boldsymbol{w}_j'^\top \boldsymbol{x})\phi'(\boldsymbol{w}_k'^\top \boldsymbol{x}) + \frac{1}{K^2 H^2(\boldsymbol{W}')}\phi''(\boldsymbol{w}_j'^\top \boldsymbol{x})\phi'(\boldsymbol{w}_k'^\top \boldsymbol{x}), & j \neq k \\ \frac{-2}{K^3 H^3(\boldsymbol{W}')}(\phi'(\boldsymbol{w}_j'^\top \boldsymbol{x}))^3 + \frac{3}{K^2 H^2(\boldsymbol{W}')}\phi''(\boldsymbol{w}_j'^\top \boldsymbol{x})\phi'(\boldsymbol{w}_j'^\top \boldsymbol{x}) - \frac{\phi'''(\boldsymbol{w}_j'^\top \boldsymbol{x})}{KH(\boldsymbol{W}')}, & j = k \end{cases}
$$

(166)

$$\boldsymbol{a}_j^\top \Delta_{j,l} \boldsymbol{a}_l = \mathbb{E}_{\boldsymbol{x} \sim \sum_{l=1}^L \mathcal{N}(\boldsymbol{\mu}_l, \boldsymbol{\Sigma}_l)}[(\sum_{k=1}^K T_{j,l,k} \langle \boldsymbol{x}, \boldsymbol{w}_k - \boldsymbol{W}^* \boldsymbol{p}_k \rangle) \cdot (\boldsymbol{a}_j^\top \boldsymbol{x})(\boldsymbol{a}_l^\top \boldsymbol{x})]$$

$$\leq \sqrt{\mathbb{E}_{\boldsymbol{x} \sim \sum_{l=1}^L \mathcal{N}(\boldsymbol{\mu}_l, \boldsymbol{\Sigma}_l)}[\sum_{k=1}^K T_{j,k,l}^2] \cdot \mathbb{E}[\sum_{k=1}^K (\langle \boldsymbol{x}, \boldsymbol{w}_k - \boldsymbol{W}^* \boldsymbol{p}_k \rangle (\boldsymbol{a}_j^\top \boldsymbol{x})(\boldsymbol{a}_l^\top \boldsymbol{x}))^2]}$$

$$\leq \sqrt{\mathbb{E}_{\boldsymbol{x} \sim \sum_{l=1}^L \mathcal{N}(\boldsymbol{\mu}_l, \boldsymbol{\Sigma}_l)}[\sum_{k=1}^K T_{j,k,l}^2]} \sqrt{\sum_{k=1}^K \sqrt{\mathbb{E}((\boldsymbol{w}_k - \boldsymbol{W}^* \boldsymbol{p}_k)^\top \boldsymbol{x})^4} \cdot \sqrt{\mathbb{E}[(\boldsymbol{a}_j^\top \boldsymbol{x})^4 (\boldsymbol{a}_l^\top \boldsymbol{x})^4]}}$$

$$\leq C_8 \sqrt{\mathbb{E}_{\boldsymbol{x} \sim \sum_{l=1}^L \mathcal{N}(\boldsymbol{\mu}_l, \boldsymbol{\Sigma}_l)}[\sum_{k=1}^K T_{j,k,l}^2]} \sqrt{\sum_{k=1}^K ||\boldsymbol{w}_k - \boldsymbol{W}^* \boldsymbol{p}_k||_2^2 \cdot ||\boldsymbol{a}_j||_2^2 \cdot ||\boldsymbol{a}_l||_2^2}$$

$$\cdot \Big( \sum_{l=1}^L \lambda_l (||\boldsymbol{\mu}_l|| + ||\boldsymbol{\Sigma}_l^{\frac{1}{2}}||)^4 \sum_{l=1}^L \lambda_l (||\boldsymbol{\mu}_l|| + ||\boldsymbol{\Sigma}_l^{\frac{1}{2}}||)^8 \Big)^{\frac{1}{4}}$$

$$(167)$$

for some constant $C_8 > 0$. All the three inequalities of (167) are derived from Cauchy-Schwarz inequality. Note that we have

$$\Big| \frac{-2}{K^3 H^3(\boldsymbol{W})} (\phi'(\boldsymbol{w}_j^\top \boldsymbol{x}))^2 \phi'(\boldsymbol{w}_k^\top \boldsymbol{x}) \Big| \leq \frac{2\phi^2(\boldsymbol{w}_j^\top \boldsymbol{x})(1 - \phi(\boldsymbol{w}_j^\top \boldsymbol{x}))^2 \phi(\boldsymbol{w}_k^\top \boldsymbol{x})(1 - \phi(\boldsymbol{w}_k^\top \boldsymbol{x}))}{K^3 \frac{1}{K^3} \phi^2(\boldsymbol{w}_j^\top \boldsymbol{x}) \phi(\boldsymbol{w}_k^\top \boldsymbol{x})}$$

$$= 2(1 - \phi(\boldsymbol{w}_j^\top \boldsymbol{x}))^2 (1 - \phi(\boldsymbol{w}_k^\top \boldsymbol{x})) \leq 2$$

$$(168)$$

$$\Big| \frac{-2}{K^3 H^3(\boldsymbol{W})} \phi'(\boldsymbol{w}_j^\top \boldsymbol{x}) \phi'(\boldsymbol{w}_l^\top \boldsymbol{x}) \phi'(\boldsymbol{w}_k^\top \boldsymbol{x}) \Big| \leq 2 \tag{169}$$

$$\Big| \frac{3}{K^2 H^2(\boldsymbol{W})} \phi''(\boldsymbol{w}_j^\top \boldsymbol{x}) \phi'(\boldsymbol{w}_k^\top \boldsymbol{x}) \Big|$$

$$\leq \Big| \frac{3\phi(\boldsymbol{w}_j^\top \boldsymbol{x})(1 - \phi(\boldsymbol{w}_j^\top \boldsymbol{x}))(1 - 2\phi(\boldsymbol{w}_j^\top \boldsymbol{x}))\phi(\boldsymbol{w}_k^\top \boldsymbol{x})(1 - \phi(\boldsymbol{w}_k^\top \boldsymbol{x}))}{K^2 \frac{1}{K^2} \phi(\boldsymbol{w}_j^\top \boldsymbol{x}) \phi(\boldsymbol{w}_k^\top \boldsymbol{x})} \Big| \tag{170}$$

$$= \Big| 3(1 - \phi(\boldsymbol{w}_j^\top \boldsymbol{x}))(1 - 2\phi(\boldsymbol{w}_j^\top \boldsymbol{x}))(1 - \phi(\boldsymbol{w}_k^\top \boldsymbol{x})) \Big| \leq 3$$

$$\Big| \frac{\phi'''(\boldsymbol{w}_j^\top \boldsymbol{x})}{K H(\boldsymbol{W})} \Big| \leq \Big| \frac{\phi(\boldsymbol{w}_j^\top \boldsymbol{x})(1 - \phi(\boldsymbol{w}_j^\top \boldsymbol{x}))(1 - 6\phi(\boldsymbol{w}_j^\top \boldsymbol{x}) + 6\phi^2(\boldsymbol{w}_j^\top \boldsymbol{x}))}{K \frac{1}{K} \phi(\boldsymbol{w}_j^\top \boldsymbol{x})} \Big| \leq 1 \tag{171}$$

Therefore, by combining (165), (166) and (168) to (171), we have

$$|T_{j,l,k}| \leq C_7 \quad \Rightarrow \quad T_{j,l,k}^2 \leq C_7^2, \forall j, l, k \in [K], \tag{172}$$

for some constants $C_7 > 0$. By (159), (160), (167), (172) and the Cauchy-Schwarz's Inequality, we have

$$\|\nabla^2 \bar{f}(\boldsymbol{W}) - \nabla^2 \bar{f}(\boldsymbol{W}^* \boldsymbol{P})\|$$

$$\leq C_8 \sqrt{C_7^2 K} \|\boldsymbol{W} - \boldsymbol{W}^* \boldsymbol{P}\|_F \Big( \sum_{l=1}^L \lambda_l (||\boldsymbol{\mu}_l|| + ||\boldsymbol{\Sigma}_l^{\frac{1}{2}}||)^4 \sum_{l=1}^L \lambda_l (||\boldsymbol{\mu}_l|| + ||\boldsymbol{\Sigma}_l^{\frac{1}{2}}||)^8 \Big)^{\frac{1}{4}}$$

$$\cdot \max_{||\boldsymbol{a}||=1} \sum_{j=1}^K \sum_{l=1}^K ||\boldsymbol{a}_j||_2 ||\boldsymbol{a}_l||_2$$

$$\leq C_8 \sqrt{C_7^2 K} \cdot \|\boldsymbol{W} - \boldsymbol{W}^* \boldsymbol{P}\|_F \cdot \Big( \sum_{l=1}^L \lambda_l (||\boldsymbol{\mu}_l|| + ||\boldsymbol{\Sigma}_l^{\frac{1}{2}}||)^4 \sum_{l=1}^L \lambda_l (||\boldsymbol{\mu}_l|| + ||\boldsymbol{\Sigma}_l^{\frac{1}{2}}||)^8 \Big)^{\frac{1}{4}} \cdot \Big( \sum_{j=1}^K ||\boldsymbol{a}_j|| \Big)^2$$

$$\leq C_8 \sqrt{C_7^2 K^3} \cdot \|\boldsymbol{W} - \boldsymbol{W}^* \boldsymbol{P}\|_F \cdot \Big( \sum_{l=1}^L \lambda_l (||\boldsymbol{\mu}_l|| + ||\boldsymbol{\Sigma}_l^{\frac{1}{2}}||)^4 \sum_{l=1}^L \lambda_l (||\boldsymbol{\mu}_l|| + ||\boldsymbol{\Sigma}_l^{\frac{1}{2}}||)^8 \Big)^{\frac{1}{4}}$$

$$(173)$$

Hence, we have

$$
\begin{aligned}
&||\nabla^2 \bar{f}(\boldsymbol{W}) - \nabla^2 \bar{f}(\boldsymbol{W}^*\boldsymbol{P})|| \\
&\leq C_5 K^{\frac{3}{2}} \Big( \sum_{l=1}^{L} \lambda_l(\|\boldsymbol{\mu}_l\| + \|\boldsymbol{\Sigma}_l^{\frac{1}{2}}\|)^4 \sum_{l=1}^{L} \lambda_l(\|\boldsymbol{\mu}_l\| + \|\boldsymbol{\Sigma}_l^{\frac{1}{2}}\|)^8 \Big)^{\frac{1}{4}} ||\boldsymbol{W} - \boldsymbol{W}^*\boldsymbol{P}||_F
\end{aligned}
\tag{174}
$$

for some constant $C_5 > 0$.

### E.6 Proof of Lemma 7

From Fu et al. (2020), we know

$$
\begin{aligned}
\nabla^2 \bar{f}(\boldsymbol{W}^*\boldsymbol{P}) &\succeq \min_{||\boldsymbol{a}||=1} \frac{4}{K^2} \mathbb{E}_{\boldsymbol{x} \sim \sum_{l=1}^{L} \lambda_l \mathcal{N}(\boldsymbol{\mu}_l, \boldsymbol{\Sigma}_l)} \Big[ \Big( \sum_{j=1}^{K} \phi'(\boldsymbol{w}_{\pi^*(j)}^{*\top}\boldsymbol{x})(\boldsymbol{a}_{\pi^*(j)}^{\top}\boldsymbol{x}) \Big)^2 \Big] \cdot \boldsymbol{I}_{dK} \\
&= \min_{||\boldsymbol{a}||=1} \frac{4}{K^2} \mathbb{E}_{\boldsymbol{x} \sim \sum_{l=1}^{L} \lambda_l \mathcal{N}(\boldsymbol{\mu}_l, \boldsymbol{\Sigma}_l)} \Big[ \Big( \sum_{j=1}^{K} \phi'(\boldsymbol{w}_j^{*\top}\boldsymbol{x})(\boldsymbol{a}_j^{\top}\boldsymbol{x}) \Big)^2 \Big] \cdot \boldsymbol{I}_{dK}
\end{aligned}
\tag{175}
$$

with $\boldsymbol{a} = (\boldsymbol{a}_1^{\top}, \cdots, \boldsymbol{a}_K^{\top})^{\top} \in \mathbb{R}^{dK}$, where $\boldsymbol{P}$ is a specific permutation matrix and $\{\pi^*(j)\}_{j=1}^{K}$ is the indices permuted by $\boldsymbol{P}$. Similarly,

$$
\begin{aligned}
\nabla^2 \bar{f}(\boldsymbol{W}^*\boldsymbol{P}) &\preceq \Big( \max_{||\boldsymbol{a}||=1} \boldsymbol{a}^{\top} \nabla^2 \bar{f}(\boldsymbol{W}^*)\boldsymbol{a} \Big) \cdot \boldsymbol{I}_{dK} \preceq C_4 \cdot \max_{||\boldsymbol{a}||=1} \mathbb{E}_{\boldsymbol{x} \sim \sum_{l=1}^{L} \lambda_l \mathcal{N}(\boldsymbol{\mu}_l, \boldsymbol{\Sigma}_l)} \Big[ \sum_{j=1}^{K} (\boldsymbol{a}_{\pi^*(j)}^{\top}\boldsymbol{x})^2 \Big] \cdot \boldsymbol{I}_{dK} \\
&= C_4 \cdot \max_{||\boldsymbol{a}||=1} \mathbb{E}_{\boldsymbol{x} \sim \sum_{l=1}^{L} \lambda_l \mathcal{N}(\boldsymbol{\mu}_l, \boldsymbol{\Sigma}_l)} \Big[ \sum_{j=1}^{K} (\boldsymbol{a}_j^{\top}\boldsymbol{x})^2 \Big] \cdot \boldsymbol{I}_{dK}
\end{aligned}
\tag{176}
$$

for some constant $C_4 > 0$. By applying Property 8, we can derive the upper bound in (176) as

$$
C_4 \cdot \mathbb{E}_{\boldsymbol{x} \sim \sum_{l=1}^{L} \lambda_l \mathcal{N}(\boldsymbol{\mu}_l, \boldsymbol{\Sigma}_l)} \Big[ \sum_{j=1}^{K} (\boldsymbol{a}_j^{\top}\boldsymbol{x})^2 \Big] \cdot \boldsymbol{I}_{dK} \preceq C_4 \cdot \sum_{l=1}^{L} \lambda_l(\|\boldsymbol{\mu}_l\| + \|\boldsymbol{\Sigma}_l^{\frac{1}{2}}\|)^2 \cdot \boldsymbol{I}_{dK}
\tag{177}
$$

To find a lower bound for (175), we can first transfer the expectation of the Gaussian Mixture Model to the weight sum of the expectations over general Gaussian distributions.

$$
\begin{aligned}
&\min_{||\boldsymbol{a}||=1} \mathbb{E}_{\boldsymbol{x} \sim \sum_{l=1}^{L} \lambda_l \mathcal{N}(\boldsymbol{\mu}_l, \boldsymbol{\Sigma}_l)} \Big[ \Big( \sum_{j=1}^{K} \phi'(\boldsymbol{w}_j^{*\top}\boldsymbol{x})(\boldsymbol{a}_j^{\top}\boldsymbol{x}) \Big)^2 \Big] \\
&= \min_{||\boldsymbol{a}||=1} \sum_{l=1}^{L} \lambda_l \mathbb{E}_{\boldsymbol{x} \sim \mathcal{N}(\boldsymbol{\mu}_l, \boldsymbol{\Sigma}_l)} \Big[ \Big( \sum_{j=1}^{K} \phi'(\boldsymbol{w}_j^{*\top}\boldsymbol{x})(\boldsymbol{a}_j^{\top}\boldsymbol{x}) \Big)^2 \Big]
\end{aligned}
\tag{178}
$$

Denote $\boldsymbol{U} \in \mathbb{R}^{d \times k}$ as the orthogonal basis of $\boldsymbol{W}^*$. For any vector $\boldsymbol{a}_i \in \mathbb{R}^d$, there exists two vectors $\boldsymbol{b}_i \in \mathbb{R}^K$ and $\boldsymbol{c}_i \in \mathbb{R}^{d-K}$ such that

$$
\boldsymbol{a}_i = \boldsymbol{U}\boldsymbol{b}_i + \boldsymbol{U}_\perp \boldsymbol{c}_i
\tag{179}
$$

where $\boldsymbol{U}_\perp \in \mathbb{R}^{d\times(d-K)}$ denotes the complement of $\boldsymbol{U}$. We also have $\boldsymbol{U}_\perp^\top \boldsymbol{\mu}_l = 0$ by Property 1. Plugging (179) into RHS of (178), and then we have

$$\mathbb{E}_{\boldsymbol{x}\sim\mathcal{N}(\boldsymbol{\mu}_l,\boldsymbol{\Sigma}_l)}\Big[\Big(\sum_{i=1}^{K}\boldsymbol{a}_i^\top \boldsymbol{x}\cdot\phi'({\boldsymbol{w}_i^*}^\top\boldsymbol{x})\Big)^2\Big]$$

$$=\mathbb{E}_{\boldsymbol{x}\sim\mathcal{N}(\boldsymbol{\mu}_l,\boldsymbol{\Sigma}_l)}\Big[\Big(\sum_{i=1}^{K}(\boldsymbol{U}\boldsymbol{b}_i+\boldsymbol{U}_\perp\boldsymbol{c}_i)^\top \boldsymbol{x}\cdot\phi'({\boldsymbol{w}_i^*}^\top\boldsymbol{x})\Big)^2\Big]=A+B+C \quad (180)$$

$$A=\mathbb{E}_{\boldsymbol{x}\sim\mathcal{N}(\boldsymbol{\mu}_l,\boldsymbol{\Sigma}_l)}\Big[\Big(\sum_{i=1}^{K}\boldsymbol{b}_i^\top \boldsymbol{U}^\top \boldsymbol{x}\cdot\phi'({\boldsymbol{w}_i^*}^\top\boldsymbol{x})\Big)^2\Big] \quad (181)$$

$$C=\mathbb{E}_{\boldsymbol{x}\sim\mathcal{N}(\boldsymbol{\mu}_l,\boldsymbol{\Sigma}_l)}\Big[2\Big(\sum_{i=1}^{K}\boldsymbol{c}_i^\top \boldsymbol{U}_\perp^\top \boldsymbol{x}\cdot\phi'({\boldsymbol{w}_i^*}^\top\boldsymbol{x})\Big)\cdot\Big(\sum_{i=1}^{K}\boldsymbol{b}_i^\top \boldsymbol{U}^\top \boldsymbol{x}\cdot\phi'({\boldsymbol{w}_i^*}^\top\boldsymbol{x})\Big)\Big]$$

$$=\sum_{i=1}^{K}\sum_{j=1}^{K}\mathbb{E}_{\boldsymbol{x}\sim\mathcal{N}(\boldsymbol{\mu}_l,\boldsymbol{\Sigma}_l)}\Big[2\boldsymbol{c}_i^\top \boldsymbol{U}_\perp^\top \boldsymbol{x}\Big]\mathbb{E}_{\boldsymbol{x}\sim\mathcal{N}(\boldsymbol{\mu}_l,\boldsymbol{\Sigma}_l)}\Big[\boldsymbol{b}_i^\top \boldsymbol{U}^\top \boldsymbol{x}\cdot\phi'({\boldsymbol{w}_i^*}^\top\boldsymbol{x})\phi'({\boldsymbol{w}_j^*}^\top\boldsymbol{x})\Big] \quad (182)$$

$$=\sum_{i=1}^{K}\sum_{j=1}^{K}\Big[2\boldsymbol{c}_i^\top \boldsymbol{U}_\perp^\top \boldsymbol{\mu}_l\Big]\mathbb{E}_{\boldsymbol{x}\sim\mathcal{N}(\boldsymbol{\mu}_l,\boldsymbol{\Sigma}_l)}\Big[\boldsymbol{b}_i^\top \boldsymbol{U}^\top \boldsymbol{x}\cdot\phi'({\boldsymbol{w}_i^*}^\top\boldsymbol{x})\phi'({\boldsymbol{w}_j^*}^\top\boldsymbol{x})\Big]=0$$

where the last step is by $\boldsymbol{U}_\perp^\top \boldsymbol{\mu}_l = 0$ by Property 1.

$$B=\mathbb{E}_{\boldsymbol{x}\sim\mathcal{N}(\boldsymbol{\mu}_l,\boldsymbol{\Sigma}_l)}\Big[(\sum_{i=1}^{K}\boldsymbol{c}_i^\top \boldsymbol{U}_\perp^\top \boldsymbol{x}\cdot\phi'({\boldsymbol{w}_i^*}^\top\boldsymbol{x}))^2\Big]$$

$$=\mathbb{E}_{\boldsymbol{x}\sim\mathcal{N}(\boldsymbol{\mu}_l,\boldsymbol{\Sigma}_l)}[(\boldsymbol{t}^\top\boldsymbol{s})^2] \qquad \text{by defining } \boldsymbol{t}=\sum_{i=1}^{k}\phi'({\boldsymbol{w}_i^*}^\top\boldsymbol{x})\boldsymbol{c}_i\in\mathbb{R}^{d-K}\text{ and } \boldsymbol{s}=\boldsymbol{U}_\perp^\top\boldsymbol{x}$$

$$=\sum_{i=1}^{K}\mathbb{E}[t_i^2 s_i^2]+\sum_{i\neq j}\mathbb{E}[t_i t_j s_i s_j]$$

$$=\sum_{i=1}^{K}\mathbb{E}[t_i^2]\sum_{k=1}^{d}(\boldsymbol{U}_\perp)_{ik}^2\sigma_{lk}^2+\Big(\sum_{i=1}^{K}\mathbb{E}[t_i^2](\boldsymbol{U}_\perp^\top\boldsymbol{\mu}_l)_i^2+\sum_{i\neq j}\mathbb{E}[t_i t_j](\boldsymbol{U}_\perp^\top\boldsymbol{\mu}_l)_i\cdot(\boldsymbol{U}_\perp^\top\boldsymbol{\mu}_l)_j\Big)$$

$$=\mathbb{E}[\sum_{i=1}^{d-K}t_i^2\cdot\sum_{k=1}^{d}(\boldsymbol{U}_\perp)_{ik}^2\sigma_{lk}^2]+\mathbb{E}[(\boldsymbol{t}^\top\boldsymbol{U}_\perp^\top\boldsymbol{\mu}_l)^2]=\mathbb{E}[\sum_{i=1}^{d-K}t_i^2\cdot\sum_{k=1}^{d}(\boldsymbol{U}_\perp)_{ik}^2\sigma_{lk}^2]$$

$$(183)$$

The last step is by $\boldsymbol{U}_\perp^\top\boldsymbol{\mu}_l=0$. The 4th step is because that $s_i$ is independent of $t_i$, thus $\mathbb{E}[t_i t_j s_i s_j]=\mathbb{E}[t_i t_j]\mathbb{E}[s_i s_j]$

$$\mathbb{E}[s_i s_j]=\begin{cases}(\boldsymbol{U}_\perp^\top\boldsymbol{\mu}_l)_i\cdot(\boldsymbol{U}_\perp^\top\boldsymbol{\mu}_l)_j, & \text{if } i\neq j \\ (\boldsymbol{U}_\perp^\top\boldsymbol{\mu}_l)_i^2+\sum_{k=1}^{d}(\boldsymbol{U}_\perp)_{ik}^2\sigma_{lk}^2, & \text{if } i=j\end{cases} \quad (184)$$

Since $\Big(\sum_{i=1}^{k}\boldsymbol{r}_i^\top\boldsymbol{x}\cdot\phi'(\sigma\cdot x_i)\Big)^2$ is an even function for any $\boldsymbol{r}_i\in\mathbb{R}^d$, $i\in[k]$, so from Property 5 we have

$$\mathbb{E}_{\boldsymbol{x}\sim\mathcal{N}(\boldsymbol{\mu}_l,\boldsymbol{\Sigma}_l)}\Big[(\sum_{i=1}^{k}\boldsymbol{r}_i^\top\boldsymbol{x}\cdot\phi'(\sigma\cdot x_i))^2\Big]=\mathbb{E}_{\boldsymbol{x}\sim\frac{1}{2}\mathcal{N}(\boldsymbol{\mu}_l,\boldsymbol{\Sigma}_l)+\frac{1}{2}\mathcal{N}(-\boldsymbol{\mu}_l,\boldsymbol{\Sigma}_l)}\Big[(\sum_{i=1}^{k}\boldsymbol{r}_i^\top\boldsymbol{x}\cdot\phi'(\sigma\cdot x_i))^2\Big] \quad (185)$$

Combining Lemma 4 and Property 5, we next follow the derivation for the standard Gaussian distribution in Page 36 of Zhong et al. (2017b) and generalize the result to a Gaussian distribution with an arbitrary mean and variance as follows.

$$A = \mathbb{E}_{\boldsymbol{x} \sim \mathcal{N}(\boldsymbol{\mu}_l, \boldsymbol{\Sigma}_l)} \Big[ \Big( \sum_{i=1}^{K} \boldsymbol{b}_i^\top \boldsymbol{U}^\top \boldsymbol{x} \cdot \phi'(\boldsymbol{w}_i^{*\top} \boldsymbol{x}) \Big)^2 \Big]$$

$$\geq \int (2\pi)^{-\frac{K}{2}} |\boldsymbol{U}^\top \boldsymbol{\Sigma}_l \boldsymbol{U}|^{-\frac{1}{2}} \Big[ \Big( \sum_{i=1}^{K} \boldsymbol{b}_i^\top \boldsymbol{z} \cdot \phi'(\boldsymbol{v}_i^\top \boldsymbol{z}) \Big)^2 \Big] \exp \Big( -\frac{1}{2} \|\boldsymbol{\Sigma}_l^{-1}\| \|\boldsymbol{z} - \boldsymbol{U}^\top \boldsymbol{\mu}_l\|^2 \Big) d\boldsymbol{z}$$

$$= \int (2\pi)^{-\frac{K}{2}} |\boldsymbol{U}^\top \boldsymbol{\Sigma}_l \boldsymbol{U}|^{-\frac{1}{2}} \Big[ \Big( \sum_{i=1}^{K} \boldsymbol{b}_i^\top \boldsymbol{V}^{\dagger\top} \boldsymbol{s} \cdot \phi'(s_i) \Big)^2 \Big] \exp \Big( -\frac{1}{2} \|\boldsymbol{\Sigma}_l^{-1}\| \|\boldsymbol{V}^{\dagger\top} \boldsymbol{s} - \boldsymbol{U}^\top \boldsymbol{\mu}_l\|^2 \Big) \Big| \det(\boldsymbol{V}^\dagger) \Big| d\boldsymbol{s}$$

$$\geq \int (2\pi)^{-\frac{K}{2}} |\boldsymbol{U}^\top \boldsymbol{\Sigma}_l \boldsymbol{U}|^{-\frac{1}{2}} \Big[ \Big( \sum_{i=1}^{k} \boldsymbol{b}_i^\top \boldsymbol{V}^{\dagger\top} \boldsymbol{s} \cdot \phi'(s_i) \Big)^2 \Big] \exp \Big( -\frac{\|\boldsymbol{\Sigma}_l^{-1}\| \|\boldsymbol{s} - \boldsymbol{V}^\top \boldsymbol{U}^\top \boldsymbol{\mu}_l\|^2}{2\delta_K^2(\boldsymbol{W}^*)} \Big) \Big| \det(\boldsymbol{V}^\dagger) \Big| d\boldsymbol{s}$$

$$\geq \int (2\pi)^{-\frac{K}{2}} |\boldsymbol{U}^\top \boldsymbol{\Sigma}_l \boldsymbol{U}|^{-\frac{1}{2}} \Big[ \Big( \sum_{i=1}^{k} \boldsymbol{b}_i^\top \boldsymbol{V}^{\dagger\top} (\delta_K(\boldsymbol{W}^*) \|\boldsymbol{\Sigma}_l^{-1}\|^{-\frac{1}{2}}) \boldsymbol{g} \cdot \phi'(\delta_K(\boldsymbol{W}^*) \|\boldsymbol{\Sigma}_l^{-1}\|^{-\frac{1}{2}} \cdot g_i) \Big)^2 \Big]$$

$$\cdot \exp \Big( -\frac{\|\boldsymbol{g} - \frac{\sqrt{\|\boldsymbol{\Sigma}_l^{-1}\|} \boldsymbol{W}^{*\top} \boldsymbol{\mu}_l}{\delta_K(\boldsymbol{W}^*)}\|^2}{2} \Big) \Big| \det(\boldsymbol{V}^\dagger) \Big| \|\boldsymbol{\Sigma}_l^{-1}\|^{-\frac{K}{2}} \delta_K^K(\boldsymbol{W}^*) d\boldsymbol{g}$$

$$= \frac{\|\boldsymbol{\Sigma}_l^{-1}\|^{-1}}{\tau^K \eta} \mathbb{E}_{\boldsymbol{g}} \Big[ \Big( \sum_{i=1}^{K} (\boldsymbol{b}_i^\top \boldsymbol{V}^{\dagger\top} \delta_K(\boldsymbol{W}^*)) \boldsymbol{g} \cdot \phi'(\|\boldsymbol{\Sigma}_l^{-1}\|^{-\frac{1}{2}} \delta_K(\boldsymbol{W}^*) \cdot g_i) \Big)^2 \Big]$$

$$\geq \frac{\|\boldsymbol{\Sigma}_l^{-1}\|^{-1}}{\tau^K \kappa^2 \eta} \rho \Big( \frac{\boldsymbol{W}^{*\top} \boldsymbol{\mu}_l}{\|\boldsymbol{\Sigma}_l^{-1}\|^{-\frac{1}{2}} \delta_K(\boldsymbol{W}^*)}, \|\boldsymbol{\Sigma}_l^{-1}\|^{-\frac{1}{2}} \delta_K(\boldsymbol{W}^*) \Big) \|\boldsymbol{b}\|^2.$$

(186)

The second step is by letting $\boldsymbol{z} = \boldsymbol{U}^\top \boldsymbol{x} \sim \mathcal{N}(\boldsymbol{U}^\top \boldsymbol{\mu}_l, \boldsymbol{U}^\top \boldsymbol{\Sigma} \boldsymbol{U})$, $\boldsymbol{y}^\top \boldsymbol{U}^\top \boldsymbol{\Sigma}_l^{-1} \boldsymbol{U} \boldsymbol{y} \leq \|\boldsymbol{\Sigma}_l^{-1}\| \|\boldsymbol{y}\|^2$ for any $\boldsymbol{y} \in \mathbb{R}^K$. The third step is by letting $\boldsymbol{s} = \boldsymbol{V}^\top \boldsymbol{z}$. The last to second step follows from $\boldsymbol{g} = \frac{\boldsymbol{s}}{\|\boldsymbol{\Sigma}_l^{-1}\|^{-\frac{1}{2}} \delta_K(\boldsymbol{W}^*)}$, where $\boldsymbol{g} \sim \mathcal{N}(\frac{\boldsymbol{W}^{*\top} \boldsymbol{\mu}_l}{\|\boldsymbol{\Sigma}_l^{-1}\|^{-\frac{1}{2}} \delta_K(\boldsymbol{W}^*)}, \boldsymbol{I}_K)$ and the last inequality is by Lemma 4. Similarly, we extend the derivation in Page 37 of Zhong et al. (2017b) for the standard Gaussian distribution to a general Gaussian distribution as follows.

$$B = \sum_{k=1}^{d} (\boldsymbol{U}_\perp)_{ik}^2 \sigma_{lk}^2 \mathbb{E}_{\boldsymbol{x} \sim \mathcal{N}(\boldsymbol{\mu}_l, \boldsymbol{\Sigma}_l)} [\|\boldsymbol{t}\|^2] \geq \frac{\|\boldsymbol{\Sigma}_l^{-1}\|^{-1}}{\eta \tau^K \kappa^2} \rho \Big( \frac{\boldsymbol{W}^{*\top} \boldsymbol{\mu}_l}{\|\boldsymbol{\Sigma}_l^{-1}\|^{-\frac{1}{2}} |\delta_K(\boldsymbol{W}^*)}, \|\boldsymbol{\Sigma}_l^{-1}\|^{-\frac{1}{2}} \delta_K(\boldsymbol{W}^*) \Big) \|\boldsymbol{c}\|^2$$

(187)

Combining (180) - (183), (186) and (187), we have

$$\min_{\|\boldsymbol{a}\|=1} \mathbb{E}_{\boldsymbol{x} \sim \mathcal{N}(\boldsymbol{\mu}_l, \boldsymbol{\Sigma}_l)} \Big[ \Big( \sum_{i=1}^{k} \boldsymbol{a}_i^\top \boldsymbol{x} \cdot \phi'(\boldsymbol{w}_i^{*\top} \boldsymbol{x}) \Big)^2 \Big] \geq \frac{\|\boldsymbol{\Sigma}_l^{-1}\|^{-1}}{\eta \tau^K \kappa^2} \rho \Big( \frac{\boldsymbol{W}^{*\top} \boldsymbol{\mu}_l}{\delta_K(\boldsymbol{W}^*) \|\boldsymbol{\Sigma}_l^{-1}\|^{-\frac{1}{2}}}, \delta_K(\boldsymbol{W}^*) \|\boldsymbol{\Sigma}_l^{-1}\|^{-\frac{1}{2}} \Big).$$

(188)

For the Gaussian Mixture Model $\boldsymbol{x} \sim \sum_{l=1}^{L} \mathcal{N}(\boldsymbol{\mu}_l, \boldsymbol{\Sigma})$, we have

$$\min_{\|\boldsymbol{a}\|=1} \mathbb{E}_{\boldsymbol{x} \sim \sum_{l=1}^{L} \lambda_l \mathcal{N}(\boldsymbol{\mu}_l, \boldsymbol{\Sigma}_l)} \Big[ \Big( \sum_{i=1}^{k} \boldsymbol{a}_i^\top \boldsymbol{x} \cdot \phi'(\boldsymbol{w}_i^{*\top} \boldsymbol{x}) \Big)^2 \Big]$$

$$\geq \sum_{l=1}^{L} \lambda_l \frac{\|\boldsymbol{\Sigma}_l^{-1}\|^{-1}}{\eta \tau^K \kappa^2} \rho \Big( \frac{\boldsymbol{W}^{*\top} \boldsymbol{\mu}_l}{\delta_K(\boldsymbol{W}^*) \|\boldsymbol{\Sigma}_l^{-1}\|^{-\frac{1}{2}}}, \delta_K(\boldsymbol{W}^*) \|\boldsymbol{\Sigma}_l^{-1}\|^{-\frac{1}{2}} \Big)$$

(189)

Therefore,

$$\frac{4}{K^2} \sum_{l=1}^{L} \lambda_l \frac{\|\boldsymbol{\Sigma}_l^{-1}\|^{-1}}{\eta \tau^K \kappa^2} \rho \Big( \frac{\boldsymbol{W}^{*\top} \boldsymbol{\mu}_l}{\delta_K(\boldsymbol{W}^*) \|\boldsymbol{\Sigma}_l^{-1}\|^{-\frac{1}{2}}}, \delta_K(\boldsymbol{W}^*) \|\boldsymbol{\Sigma}_l^{-1}\|^{-\frac{1}{2}} \Big) \cdot \boldsymbol{I}_{dK}$$

$$\preceq \nabla^2 \bar{f}(\boldsymbol{W}^* \boldsymbol{P}) \preceq C_4 \cdot \sum_{l=1}^{L} \lambda_l (\|\boldsymbol{\mu}_l\| + \|\boldsymbol{\Sigma}_l^{\frac{1}{2}}\|)^2 \cdot \boldsymbol{I}_{dK}$$

(190)

From (139) in Lemma 6, since that we have the condition $\|\boldsymbol{W} - \boldsymbol{W}^*\boldsymbol{P}\|_F \leq r$ and (93), we can obtain

$$\|\nabla^2 \bar{f}(\boldsymbol{W}) - \nabla^2 \bar{f}(\boldsymbol{W}^*\boldsymbol{P})\|$$
$$\leq C_5 K^{\frac{3}{2}} \Big( \sum_{l=1}^{L} \lambda_l(\|\boldsymbol{\mu}_l\| + \|\boldsymbol{\Sigma}_l^{\frac{1}{2}}\|)^4 \sum_{l=1}^{L} \lambda_l(\|\boldsymbol{\mu}_l\| + \|\boldsymbol{\Sigma}_l^{\frac{1}{2}}\|)^8 \Big)^{\frac{1}{4}} \|\boldsymbol{W} - \boldsymbol{W}^*\boldsymbol{P}\|_F \tag{191}$$
$$\leq \frac{4\epsilon_0}{K^2} \sum_{l=1}^{L} \lambda_l \frac{\|\boldsymbol{\Sigma}_l^{-1}\|^{-1}}{\eta \tau^K \kappa^2} \rho\Big( \frac{\boldsymbol{W}^{*\top}\boldsymbol{\mu}_l}{\delta_K(\boldsymbol{W}^*)\|\boldsymbol{\Sigma}_l^{-1}\|^{-\frac{1}{2}}}, \delta_K(\boldsymbol{W}^*)\|\boldsymbol{\Sigma}_l^{-1}\|^{-\frac{1}{2}} \Big),$$

where $\epsilon_0 \in (0, \frac{1}{4})$. Then we have

$$\|\nabla^2 \bar{f}(\boldsymbol{W})\| \geq \|\nabla^2 \bar{f}(\boldsymbol{W}^*\boldsymbol{P})\| - \|\nabla^2 \bar{f}(\boldsymbol{W}) - \nabla^2 \bar{f}(\boldsymbol{W}^*\boldsymbol{P})\|$$
$$\geq \frac{4(1 - \epsilon_0)}{K^2} \sum_{l=1}^{L} \lambda_l \frac{\|\boldsymbol{\Sigma}_l^{-1}\|^{-1}}{\eta \tau^K \kappa^2} \rho\Big( \frac{\boldsymbol{W}^{*\top}\boldsymbol{\mu}_l}{\delta_K(\boldsymbol{W}^*)\|\boldsymbol{\Sigma}_l^{-1}\|^{-\frac{1}{2}}}, \delta_K(\boldsymbol{W}^*)\|\boldsymbol{\Sigma}_l^{-1}\|^{-\frac{1}{2}} \Big) \tag{192}$$

$$\|\nabla^2 \bar{f}(\boldsymbol{W})\| \leq \|\nabla^2 \bar{f}(\boldsymbol{W}^*)\| + \|\nabla^2 \bar{f}(\boldsymbol{W}) - \nabla^2 \bar{f}(\boldsymbol{W}^*\boldsymbol{P})\|$$
$$\leq C_4 \cdot \sum_{l=1}^{L} \lambda_l(\|\boldsymbol{\mu}_l\| + \|\boldsymbol{\Sigma}^{\frac{1}{2}}\|)^2 + \frac{4}{K^2} \sum_{l=1}^{L} \lambda_l \frac{\|\boldsymbol{\Sigma}_l^{-1}\|^{-1}}{\eta \tau^K \kappa^2} \rho\Big( \frac{\boldsymbol{W}^{*\top}\boldsymbol{\mu}_l}{\delta_K(\boldsymbol{W}^*)\|\boldsymbol{\Sigma}_l^{-\frac{1}{2}}\|}, \delta_K(\boldsymbol{W}^*)\|\boldsymbol{\Sigma}_l^{-\frac{1}{2}}\| \Big)$$
$$\lesssim C_4 \cdot \sum_{l=1}^{L} \lambda_l(\|\boldsymbol{\mu}_l\| + \|\boldsymbol{\Sigma}_l^{\frac{1}{2}}\|)^2 \tag{193}$$

The last inequality of (193) holds since $C_4 \cdot \sum_{l=1} \lambda_l(\|\boldsymbol{\mu}_l\| + \|\boldsymbol{\Sigma}_l^{\frac{1}{2}}\|)^2 = \Omega(\max_l\{\|\boldsymbol{\Sigma}_l\|\})$, $\frac{4}{K^2} \sum_{l=1}^{L} \lambda_l \frac{\|\boldsymbol{\Sigma}_l^{-1}\|^{-1}}{\eta \tau^K \kappa^2} \rho\big( \frac{\boldsymbol{W}^{*\top}\boldsymbol{\mu}_l}{\delta_K(\boldsymbol{W}^*)\|\boldsymbol{\Sigma}_l^{-1}\|^{-\frac{1}{2}}}, \delta_K(\boldsymbol{W}^*)\|\boldsymbol{\Sigma}_l^{-1}\|^{-\frac{1}{2}} \big) = O(\frac{\max_l\{\|\boldsymbol{\Sigma}_l\|\}}{K^2})$ and $\Omega(\max_l\{\|\boldsymbol{\Sigma}_l\|\}) \geq O(\frac{\max_l\{\|\boldsymbol{\Sigma}_l\|\}}{K^2})$. Combining (192) and (193), we have

$$\frac{4(1 - \epsilon_0)}{K^2} \sum_{l=1}^{L} \lambda_l \frac{\|\boldsymbol{\Sigma}_l^{-1}\|^{-1}}{\eta \tau^K \kappa^2} \rho\Big( \frac{\boldsymbol{W}^{*\top}\boldsymbol{\mu}_l}{\delta_K(\boldsymbol{W}^*)\|\boldsymbol{\Sigma}_l^{-1}\|^{-\frac{1}{2}}}, \delta_K(\boldsymbol{W}^*)\|\boldsymbol{\Sigma}_l^{-1}\|^{-\frac{1}{2}} \Big) \cdot \boldsymbol{I}$$
$$\preceq \nabla^2 \bar{f}(\boldsymbol{W}) \preceq C_4 \cdot \sum_{l=1}^{L} \lambda_l(\|\boldsymbol{\mu}_l\| + \sigma_l)^2 \cdot \boldsymbol{I} \tag{194}$$

### E.7 Proof of Lemma 8

Let $N_\epsilon$ be the $\epsilon$-covering number of the Euclidean ball $\mathbb{B}(\boldsymbol{W}^*\boldsymbol{P}, r)$. It is known that $\log N_\epsilon \leq dK \log(\frac{3r}{\epsilon})$ from Vershynin (2010). Let $\mathcal{W}_\epsilon = \{\boldsymbol{W}_1, ..., \boldsymbol{W}_{N_\epsilon}\}$ be the $\epsilon$-cover set with $N_\epsilon$ elements. For any $\boldsymbol{W} \in \mathbb{B}(\boldsymbol{W}^*\boldsymbol{P}, r)$, let $j(\boldsymbol{W}) = \arg\min_{j \in [N_\epsilon]} \|\boldsymbol{W} - \boldsymbol{W}_{j(\boldsymbol{W})}\|_F \leq \epsilon$ for all $\boldsymbol{W} \in \mathbb{B}(\boldsymbol{W}^*\boldsymbol{P}, r)$.
Then for any $\boldsymbol{W} \in \mathbb{B}(\boldsymbol{W}^*\boldsymbol{P}, r)$, we have

$$\|\nabla^2 f_n(\boldsymbol{W}) - \nabla^2 \bar{f}(\boldsymbol{W})\|$$
$$\leq \frac{1}{n} \|\sum_{i=1}^{n} [\nabla^2 \ell(\boldsymbol{W}; \boldsymbol{x}_i) - \nabla^2 \ell(\boldsymbol{W}_{j(\boldsymbol{W})}; \boldsymbol{x}_i)]\|$$
$$+ \|\frac{1}{n} \sum_{i=1}^{n} \nabla^2 \ell(\boldsymbol{W}_{j(\boldsymbol{W})}; \boldsymbol{x}_i) - \mathbb{E}_{\boldsymbol{x} \sim \sum_{l=1}^{L} \lambda_l \mathcal{N}(\boldsymbol{\mu}_l, \boldsymbol{\Sigma}_l)}[\nabla^2 \ell(\boldsymbol{W}_{j(\boldsymbol{W})}; \boldsymbol{x}_i)]\| \tag{195}$$
$$+ \|\mathbb{E}_{\boldsymbol{x} \sim \sum_{l=1}^{L} \lambda_l \mathcal{N}(\boldsymbol{\mu}_l, \boldsymbol{\Sigma}_l)}[\nabla^2 \ell(\boldsymbol{W}_{j(\boldsymbol{W})}; \boldsymbol{x}_i)] - \mathbb{E}_{\boldsymbol{x} \sim \sum_{l=1}^{L} \lambda_l \mathcal{N}(\boldsymbol{\mu}_l, \boldsymbol{\Sigma}_l)}[\nabla^2 \ell(\boldsymbol{W}; \boldsymbol{x}_i)]\|$$

Hence, we have

$$\mathbb{P}\Big( \sup_{\boldsymbol{W} \in \mathbb{B}(\boldsymbol{W}^*\boldsymbol{P}, r)} \|\nabla^2 f_n(\boldsymbol{W}) - \nabla^2 \bar{f}(\boldsymbol{W})\| \geq t \Big) \leq \mathbb{P}(A_t) + \mathbb{P}(B_t) + \mathbb{P}(C_t) \tag{196}$$

where $A_t$, $B_t$ and $C_t$ are defined as

$$A_t = \{ \sup_{\boldsymbol{W} \in \mathbb{B}(\boldsymbol{W}^*\boldsymbol{P}, r)} \frac{1}{n} \|\sum_{i=1}^{n} [\nabla^2 \ell(\boldsymbol{W}; \boldsymbol{x}_i) - \nabla^2 \ell(\boldsymbol{W}_{j(\boldsymbol{W})}; \boldsymbol{x}_i)]\| \geq \frac{t}{3} \} \tag{197}$$

$$B_t = \{ \sup_{\boldsymbol{W} \in \mathbb{B}(\boldsymbol{W}^*\boldsymbol{P}, r)} \|\frac{1}{n} \sum_{i=1}^{n} \nabla^2 \ell(\boldsymbol{W}_{j(\boldsymbol{W})}; \boldsymbol{x}_i) - \mathbb{E}_{\boldsymbol{x} \sim \sum_{l=1}^{L} \lambda_l \mathcal{N}(\boldsymbol{\mu}_l, \boldsymbol{\Sigma}_l)} [\nabla^2 \ell(\boldsymbol{W}_{j(\boldsymbol{W})}; \boldsymbol{x}_i)]\| \geq \frac{t}{3} \} \tag{198}$$

$$C_t = \{ \sup_{\boldsymbol{W} \in \mathbb{B}(\boldsymbol{W}^*\boldsymbol{P}, r)} \|\mathbb{E}_{\boldsymbol{x} \sim \sum_{l=1}^{L} \lambda_l \mathcal{N}(\boldsymbol{\mu}_l, \boldsymbol{\Sigma}_l)} [\nabla^2 \ell(\boldsymbol{W}_{j(\boldsymbol{W})}; \boldsymbol{x}_i)]$$
$$- \mathbb{E}_{\boldsymbol{x} \sim \sum_{l=1}^{L} \lambda_l \mathcal{N}(\boldsymbol{\mu}_l, \boldsymbol{\Sigma}_l)} [\nabla^2 \ell(\boldsymbol{W}; \boldsymbol{x}_i)]\| \geq \frac{t}{3} \} \tag{199}$$

Then we bound $\mathbb{P}(A_t)$, $\mathbb{P}(B_t)$, and $\mathbb{P}(C_t)$ separately.

1) **Upper bound on** $\mathbb{P}(B_t)$. By Lemma 6 in Fu et al. (2020), we obtain

$$\left\| \frac{1}{n} \sum_{i=1}^{n} \nabla^2 \ell(\boldsymbol{W}; \boldsymbol{x}_i) - \mathbb{E}_{\boldsymbol{x} \sim \sum_{l=1}^{L} \lambda_l \mathcal{N}(\boldsymbol{\mu}_l, \boldsymbol{\Sigma}_l)} [\nabla^2 \ell(\boldsymbol{W}; \boldsymbol{x}_i)] \right\|$$
$$\leq 2 \sup_{\boldsymbol{v} \in \boldsymbol{V}_{\frac{1}{4}}} \left| \left\langle \boldsymbol{v}, (\frac{1}{n} \sum_{i=1}^{n} \nabla^2 \ell(\boldsymbol{W}; \boldsymbol{x}_i) - \mathbb{E}_{\boldsymbol{x} \sim \sum_{l=1}^{L} \lambda_l \mathcal{N}(\boldsymbol{\mu}_l, \boldsymbol{\Sigma}_l)} [\nabla^2 \ell(\boldsymbol{W}; \boldsymbol{x}_i)]) \boldsymbol{v} \right\rangle \right| \tag{200}$$

where $\boldsymbol{V}_{\frac{1}{4}}$ is a $\frac{1}{4}$-cover of the unit-Euclidean-norm ball $\mathbb{B}(\boldsymbol{0}, 1)$ with $\log |\boldsymbol{V}_{\frac{1}{4}}| \leq dK \log 12$. Taking the union bound over $\mathcal{W}_\epsilon$ and $\boldsymbol{V}_{\frac{1}{4}}$, we have

$$\mathbb{P}(B_t) \leq \mathbb{P} \left( \sup_{\boldsymbol{W} \in \mathcal{W}_\epsilon, \boldsymbol{v} \in \boldsymbol{V}_{\frac{1}{4}}} \left| \frac{1}{n} \sum_{i=1}^{n} G_i \right| \geq \frac{t}{6} \right)$$
$$\leq \exp(dK(\log \frac{3r}{\epsilon} + \log 12)) \sup_{\boldsymbol{W} \in \mathcal{W}_\epsilon, \boldsymbol{v} \in \boldsymbol{V}_{\frac{1}{4}}} \mathbb{P}(|\frac{1}{n} \sum_{i=1}^{n} G_i| \geq \frac{t}{6}) \tag{201}$$

where $G_i = \left\langle \boldsymbol{v}, (\nabla^2 \ell(\boldsymbol{W}, \boldsymbol{x}_i) - \mathbb{E}_{\boldsymbol{x} \sim \sum_{l=1}^{L} \lambda_l \mathcal{N}(\boldsymbol{\mu}_l, \boldsymbol{\Sigma}_l)} [\nabla^2 \ell(\boldsymbol{W}, \boldsymbol{x}_i)] \boldsymbol{v}) \right\rangle$ and $\mathbb{E}[G_i] = 0$. Here $\boldsymbol{v} = (\boldsymbol{u}_1^\top, \cdots, \boldsymbol{u}_K^\top)^\top \in \mathbb{R}^{dK}$.

$$|G_i| = \left| \sum_{j=1}^{K} \sum_{l=1}^{K} \left[ \xi_{j,l} \boldsymbol{u}_j^\top \boldsymbol{x} \boldsymbol{x}^\top \boldsymbol{u}_l - \mathbb{E}_{\boldsymbol{x} \sim \sum_{l=1}^{L} \lambda_l \mathcal{N}(\boldsymbol{\mu}_l, \boldsymbol{\Sigma}_l)} (\xi_{j,l} \boldsymbol{u}_j^\top \boldsymbol{x} \boldsymbol{x}^\top \boldsymbol{u}_l) \right] \right|$$
$$\leq C_9 \cdot \left[ \sum_{j=1}^{K} (\boldsymbol{u}_j^\top \boldsymbol{x})^2 + \sum_{j=1}^{K} \mathbb{E}_{\boldsymbol{x} \sim \sum_{l=1}^{L} \lambda_l \mathcal{N}(\boldsymbol{\mu}_l, \boldsymbol{\Sigma}_l)} (\boldsymbol{u}_j^\top \boldsymbol{x})^2 \right] \tag{202}$$

for some $C_9 > 0$. The first step of (202) is by (84). The last step is by (164) and the Cauchy-Schwarz's Inequality.

$$
\begin{aligned}
\mathbb{E}[|G_i|^p] &\le \sum_{l=1}^{p} \binom{p}{l} C_9 \cdot \mathbb{E}_{\boldsymbol{x} \sim \sum_{l=1}^{L} \lambda_l \mathcal{N}(\boldsymbol{\mu}_l, \boldsymbol{\Sigma}_l)} \Big[ \big( \sum_{j=1}^{K} (\boldsymbol{u}_j^\top \boldsymbol{x})^2 \big)^l \Big] \\
&\quad \cdot \Big( \sum_{j=1}^{K} \mathbb{E}_{\boldsymbol{x} \sim \sum_{l=1}^{L} \lambda_l \mathcal{N}(\boldsymbol{\mu}_l, \boldsymbol{\Sigma}_l)} (\boldsymbol{u}_j^\top \boldsymbol{x})^2 \Big)^{p-l} \\
&= \sum_{l=1}^{p} \binom{p}{l} C_9 \cdot \mathbb{E}_{\boldsymbol{x} \sim \sum_{l=1}^{L} \lambda_l \mathcal{N}(\boldsymbol{\mu}_l, \boldsymbol{\Sigma}_l)} \Big[ \sum_{l_1 + \cdots + l_K = l} \frac{l!}{\prod_{j=1}^{K} l_j!} \prod_{j=1}^{K} (\boldsymbol{u}_j^\top \boldsymbol{x})^{2l_j} \Big] \\
&\quad \cdot \Big( \sum_{j=1}^{K} \mathbb{E}_{\boldsymbol{x} \sim \sum_{l=1}^{L} \lambda_l \mathcal{N}(\boldsymbol{\mu}_l, \boldsymbol{\Sigma}_l)} (\boldsymbol{u}_j^\top \boldsymbol{x})^2 \Big)^{p-l} \\
&= \sum_{l=1}^{p} \binom{p}{l} C_9 \cdot \Big[ \sum_{l_1 + \cdots + l_K = l} \frac{l!}{\prod_{j=1}^{K} l_j!} \prod_{j=1}^{K} \mathbb{E}_{\boldsymbol{x} \sim \sum_{l=1}^{L} \lambda_l \mathcal{N}(\boldsymbol{\mu}_l, \boldsymbol{\Sigma}_l)} (\boldsymbol{u}_j^\top \boldsymbol{x})^{2l_j} \Big] \\
&\quad \cdot \Big( \sum_{j=1}^{K} \mathbb{E}_{\boldsymbol{x} \sim \sum_{l=1}^{L} \lambda_l \mathcal{N}(\boldsymbol{\mu}_l, \boldsymbol{\Sigma}_l)} (\boldsymbol{u}_j^\top \boldsymbol{x})^2 \Big)^{p-l} \quad (203) \\
&= C_9 \cdot \sum_{l=1}^{p} \binom{p}{l} \Big( \sum_{j=1}^{K} \mathbb{E}_{\boldsymbol{x} \sim \sum_{l=1}^{L} \lambda_l \mathcal{N}(\boldsymbol{\mu}_l, \boldsymbol{\Sigma}_l)} (\boldsymbol{u}_j^\top \boldsymbol{x})^2 \Big)^l \\
&\quad \cdot \Big( \sum_{j=1}^{K} \mathbb{E}_{\boldsymbol{x} \sim \sum_{l=1}^{L} \lambda_l \mathcal{N}(\boldsymbol{\mu}_l, \boldsymbol{\Sigma}_l)} (\boldsymbol{u}_j^\top \boldsymbol{x})^2 \Big)^{p-l} \\
&= C_9 \cdot \Big( \sum_{j=1}^{K} \mathbb{E}_{\boldsymbol{x} \sim \sum_{l=1}^{L} \lambda_l \mathcal{N}(\boldsymbol{\mu}_l, \boldsymbol{\Sigma}_l)} (\boldsymbol{u}_j^\top \boldsymbol{x})^2 \Big)^p \\
&\le C_9 \cdot \Big( \sum_{j=1}^{K} 1!! \|\boldsymbol{u}_j\|^2 \sum_{l=1}^{L} \lambda_l (\|\boldsymbol{\mu}_l\| + \|\boldsymbol{\Sigma}_l^{\frac{1}{2}}\|)^2 \Big)^p \\
&\le C_9 \cdot \Big( \sum_{l=1}^{L} \lambda_l (\|\boldsymbol{\mu}_l\| + \|\boldsymbol{\Sigma}_l^{\frac{1}{2}}\|)^2 \Big)^p
\end{aligned}
$$

where the first step is by the triangle inequality and the Binomial theorem, and the second step comes from the Multinomial theorem. The second to last inequality in (203) results from Property 8. The last inequality is because $\boldsymbol{v} \in \boldsymbol{V}_{\frac{1}{4}}$, $\sum_{j=1}^{K} \|u_j\|^2 = \|\boldsymbol{v}\|^2 \le 1$.

$$
\begin{aligned}
\mathbb{E}[\exp(\theta G_i)] &= 1 + \theta \mathbb{E}[G_i] + \sum_{p=2}^{\infty} \frac{\theta^p \mathbb{E}[|G_i|^p]}{p!} \\
&\le 1 + \sum_{p=2}^{\infty} \frac{|e\theta|^p}{p^p} C_9 \cdot \Big( \sum_{l=1}^{} \lambda_l (\|\boldsymbol{\mu}_l\| + \|\boldsymbol{\Sigma}_l^{\frac{1}{2}}\|)^2 \Big)^p \quad (204) \\
&\le 1 + C_9 \cdot |e\theta|^2 \Big( \sum_{l=1}^{L} \lambda_l (\|\boldsymbol{\mu}_l\| + \|\boldsymbol{\Sigma}_l^{\frac{1}{2}}\|)^2 \Big)^2
\end{aligned}
$$

where the first inequality holds from $p! \ge (\frac{p}{e})^p$ and (203), and the third line holds provided that

$$
\max_{p \ge 2} \Big\{ \frac{\frac{|e\theta|^{(p+1)}}{(p+1)^{(p+1)}} \cdot \Big( \sum_{l=1}^{L} \lambda_l (\|\boldsymbol{\mu}_l\| + \|\boldsymbol{\Sigma}_l^{\frac{1}{2}}\|)^2 \Big)^{p+1}}{\frac{|e\theta|^p}{p^p} \cdot \Big( \sum_{l=1}^{L} \lambda_l (\|\boldsymbol{\mu}_l\| + \|\boldsymbol{\Sigma}_l^{\frac{1}{2}}\|)^2 \Big)^{p}} \Big\} \le \frac{1}{2} \quad (205)
$$

Note that the quantity inside the maximization in (205) achieves its maximum when $p = 2$, because it is monotonously decreasing. Therefore, (205) holds if $\theta \leq \frac{27}{4e} \sum_{l=1}^{L} \lambda_l(\|\boldsymbol{\mu}_l\| + \|\boldsymbol{\Sigma}_l^{\frac{1}{2}}\|)^2$. Then

$$
\begin{aligned}
\mathbb{P}\Big(\frac{1}{n}\sum_{i=1}^{n} G_i \geq \frac{t}{6}\Big) &= \mathbb{P}\Big(\exp(\theta\sum_{i=1}^{n} G_i) \geq \exp(\frac{n\theta t}{6})\Big) \leq e^{-\frac{n\theta t}{6}}\prod_{i=1}^{n}\mathbb{E}[\exp(\theta G_i)] \\
&\leq \exp(C_{10}\theta^2 n\Big(\sum_{l=1}^{L}\lambda_l(\|\boldsymbol{\mu}_l\| + \|\boldsymbol{\Sigma}_l^{\frac{1}{2}}\|)^2\Big)^2 - \frac{n\theta t}{6})
\end{aligned}
\tag{206}
$$

for some constant $C_{10} > 0$. The first inequality follows from Markov's Inequality. When $\theta = \min\{\frac{t}{12C_{10}\Big(\sum_{l=1}^{L}\lambda_l(\|\boldsymbol{\mu}_l\| + \|\boldsymbol{\Sigma}_l^{\frac{1}{2}}\|)^2\Big)^2}, \frac{27}{4e}\sum_{l=1}^{L}\lambda_l(\|\boldsymbol{\mu}_l\| + \|\boldsymbol{\Sigma}_l^{\frac{1}{2}}\|)^2\}$, we have a modified Bernstein's Inequality for the Gaussian Mixture Model as follows

$$
\begin{aligned}
\mathbb{P}(\frac{1}{n}\sum_{i=1}^{n} G_i \geq \frac{t}{6}) \leq \exp\Big( \max\{ &-\frac{C_{10}nt^2}{144\Big(\sum_{l=1}^{L}\lambda_l(\|\boldsymbol{\mu}_l\| + \|\boldsymbol{\Sigma}_l^{\frac{1}{2}}\|)^2\Big)^2}, \\
&-C_{11}n\sum_{l=1}^{L}\lambda_l(\|\boldsymbol{\mu}_l\| + \|\boldsymbol{\Sigma}_l^{\frac{1}{2}}\|)^2 \cdot t\}\Big)
\end{aligned}
\tag{207}
$$

for some constant $C_{11} > 0$. We can obtain the same bound for $\mathbb{P}(-\frac{1}{n}\sum_{i=1}^{n} G_i \geq \frac{t}{6})$ by replacing $G_i$ as $-G_i$. Therefore, we have

$$
\begin{aligned}
\mathbb{P}(|\frac{1}{n}\sum_{i=1}^{n} G_i| \geq \frac{t}{6}) \leq 2\exp\Big( \max\{ &-\frac{C_{10}nt^2}{144\Big(\sum_{l=1}^{L}\lambda_l(\|\boldsymbol{\mu}_l\| + \|\boldsymbol{\Sigma}_l^{\frac{1}{2}}\|)^2\Big)^2}, \\
&-C_{11}n\sum_{l=1}^{L}\lambda_l(\|\boldsymbol{\mu}_l\| + \|\boldsymbol{\Sigma}_l^{\frac{1}{2}}\|)^2 \cdot t\}\Big)
\end{aligned}
\tag{208}
$$

Thus, as long as

$$
t \geq C_6 \cdot \max\{\sum_{l=1}^{L}\lambda_l(\|\boldsymbol{\mu}_l\| + \|\boldsymbol{\Sigma}_l^{\frac{1}{2}}\|)^2 \sqrt{\frac{dK\log\frac{36r}{\epsilon} + \log\frac{4}{\delta}}{n}}, \frac{dK\log\frac{36r}{\epsilon} + \log\frac{4}{\delta}}{\sum_{l=1}^{L}\lambda_l(\|\boldsymbol{\mu}_l\| + \|\boldsymbol{\Sigma}_l^{\frac{1}{2}}\|)^2 n}\}
\tag{209}
$$

for some large constant $C_6 > 0$, we have $\mathbb{P}(B_t) \leq \frac{\delta}{2}$.

2) **Upper bound on $\mathbb{P}(A_t)$ and $\mathbb{P}(C_t)$.** From Lemma 5, we can obtain

$$
\begin{aligned}
&\sup_{\boldsymbol{W}\in\mathbb{B}(\boldsymbol{W}^*\boldsymbol{P},r)} ||\mathbb{E}_{\boldsymbol{x}\sim\sum_{l=1}^{L}\lambda_l\mathcal{N}(\boldsymbol{\mu}_l,\boldsymbol{\Sigma}_l)}[\nabla^2\ell(\boldsymbol{W}_{j(\boldsymbol{W})};\boldsymbol{x})] - \mathbb{E}_{\boldsymbol{x}\sim\sum_{l=1}^{L}\lambda_l\mathcal{N}(\boldsymbol{\mu}_l,\boldsymbol{\Sigma}_l)}[\nabla^2\ell(\boldsymbol{W};\boldsymbol{x})]|| \\
&\leq \sup_{\boldsymbol{W}\in\mathbb{B}(\boldsymbol{W}^*\boldsymbol{P},r)} \frac{||\mathbb{E}_{\boldsymbol{x}\sim\sum_{l=1}^{L}\lambda_l\mathcal{N}(\boldsymbol{\mu}_l,\boldsymbol{\Sigma}_l)}[\nabla^2\ell(\boldsymbol{W}_{j(\boldsymbol{W})};\boldsymbol{x})] - \mathbb{E}_{\boldsymbol{x}\sim\sum_{l=1}^{L}\lambda_l\mathcal{N}(\boldsymbol{\mu}_l,\boldsymbol{\Sigma}_l)}[\nabla^2\ell(\boldsymbol{W};\boldsymbol{x})]||}{||\boldsymbol{W} - \boldsymbol{W}_{j(\boldsymbol{W})}||_F} \\
&\quad \cdot \sup_{\boldsymbol{W}\in\mathbb{B}(\boldsymbol{W}^*\boldsymbol{P},r)} ||\boldsymbol{W} - \boldsymbol{W}_{j(\boldsymbol{W})}||_F \\
&\leq C_{12} \cdot d^{\frac{3}{2}}K^{\frac{5}{2}}\sqrt{\sum_{l=1}^{L}\lambda_l(\|\boldsymbol{\mu}_l\|_\infty + \|\boldsymbol{\Sigma}_l^{\frac{1}{2}}\|)^2 \sum_{l=1}^{L}\lambda_l(\|\boldsymbol{\mu}_l\|_\infty + \|\boldsymbol{\Sigma}_l^{\frac{1}{2}}\|)^4} \cdot \epsilon
\end{aligned}
\tag{210}
$$

Therefore, $C_t$ holds if

$$
t \geq C_{12} \cdot d^{\frac{3}{2}}K^{\frac{5}{2}}\sqrt{\sum_{l=1}^{L}\lambda_l(\|\boldsymbol{\mu}_l\|_\infty + \|\boldsymbol{\Sigma}_l^{\frac{1}{2}}\|)^2 \sum_{l=1}^{L}\lambda_l(\|\boldsymbol{\mu}_l\|_\infty + \|\boldsymbol{\Sigma}_l^{\frac{1}{2}}\|)^4} \cdot \epsilon
\tag{211}
$$

We can bound the $A_t$ as below.

$$
\begin{aligned}
&\mathbb{P}\Big(\sup_{\boldsymbol{W}\in\mathbb{B}(\boldsymbol{W}^*\boldsymbol{P},r)}\frac{1}{n}||\sum_{i=1}^n[\nabla^2\ell(\boldsymbol{W}_{j(\boldsymbol{W})};\boldsymbol{x}_i)-\nabla^2\ell(\boldsymbol{W};\boldsymbol{x}_i)]||\geq\frac{t}{3}\Big)\\
&\leq\frac{3}{t}\mathbb{E}_{\boldsymbol{x}\sim\sum_{l=1}^L\lambda_l\mathcal{N}(\boldsymbol{\mu}_l,\boldsymbol{\Sigma}_l)}\Big[\sup_{\boldsymbol{W}\in\mathbb{B}(\boldsymbol{W}^*\boldsymbol{P},r)}\frac{1}{n}||\sum_{i=1}^n[\nabla^2\ell(\boldsymbol{W}_{j(\boldsymbol{W})};\boldsymbol{x}_i)-\nabla^2\ell(\boldsymbol{W};\boldsymbol{x}_i)]||\Big]\\
&=\frac{3}{t}\mathbb{E}_{\boldsymbol{x}\sim\sum_{l=1}^L\lambda_l\mathcal{N}(\boldsymbol{\mu}_l,\boldsymbol{\Sigma}_l)}\Big[\sup_{\boldsymbol{W}\in\mathbb{B}(\boldsymbol{W}^*\boldsymbol{P},r)}||\nabla^2\ell(\boldsymbol{W}_{j(\boldsymbol{W})};\boldsymbol{x}_i)-\nabla^2\ell(\boldsymbol{W};\boldsymbol{x}_i)||\Big]\\
&\leq\frac{3}{t}\mathbb{E}\Big[\sup_{\boldsymbol{W}\in\mathbb{B}(\boldsymbol{W}^*\boldsymbol{P},r)}\frac{||\nabla^2\ell(\boldsymbol{W}_{j(\boldsymbol{W})};\boldsymbol{x}_i)-\nabla^2\ell(\boldsymbol{W};\boldsymbol{x}_i)||}{||\boldsymbol{W}-\boldsymbol{W}_{j(\boldsymbol{W})}||_F}\Big]\cdot\sup_{\boldsymbol{W}\in\mathbb{B}(\boldsymbol{W}^*\boldsymbol{P},r)}||\boldsymbol{W}-\boldsymbol{W}_{j(\boldsymbol{W})}||_F\\
&\leq\frac{C_{12}\cdot d^{\frac{3}{2}}K^{\frac{5}{2}}\sqrt{\sum_{l=1}^L\lambda_l(||\boldsymbol{\mu}_l||_\infty+||\boldsymbol{\Sigma}_l^{\frac{1}{2}}||)^2\sum_{l=1}^L\lambda_l(||\boldsymbol{\mu}_l||_\infty+||\boldsymbol{\Sigma}_l^{\frac{1}{2}}||)^4}\cdot\epsilon}{t},
\end{aligned}
\tag{212}
$$

where the first inequality is by Markov's inequality, and the last inequality comes from Lemma 5. Thus, taking

$$
t\geq\frac{C_{12}\cdot d^{\frac{3}{2}}K^{\frac{5}{2}}\sqrt{\sum_{l=1}^L\lambda_l(||\boldsymbol{\mu}_l||_\infty+||\boldsymbol{\Sigma}_l^{\frac{1}{2}}||)^2\sum_{l=1}^L\lambda_l(||\boldsymbol{\mu}_l||_\infty+||\boldsymbol{\Sigma}_l^{\frac{1}{2}}||)^4}\cdot\epsilon}{\delta}
\tag{213}
$$

ensures that $\mathbb{P}(A_t)\leq\frac{\delta}{2}$.

3) **Final step**
Let $\epsilon=\dfrac{\delta}{C_{12}\cdot d^{\frac{3}{2}}K^{\frac{5}{2}}\sqrt{\sum_{l=1}^L\lambda_l(||\boldsymbol{\mu}_l||_\infty+||\boldsymbol{\Sigma}_l^{\frac{1}{2}}||)^2\sum_{l=1}^L\lambda_l(||\boldsymbol{\mu}_l||_\infty+||\boldsymbol{\Sigma}_l^{\frac{1}{2}}||)^4}\cdot ndK}$ and $\delta=d^{-10}$, then from (209) and (213) we need

$$
t>\max\Big\{\frac{1}{ndK},\ \ C_6\cdot\sum_{l=1}^L\lambda_l(||\boldsymbol{\mu}_l||+||\boldsymbol{\Sigma}_l^{\frac{1}{2}}||)^2
$$

$$
\cdot\sqrt{\frac{dK\log(36rnd^{\frac{25}{2}}K^{\frac{7}{2}}\sqrt{\sum_{l=1}^L\lambda_l(||\boldsymbol{\mu}_l||_\infty+||\boldsymbol{\Sigma}_l^{\frac{1}{2}}||)^2\sum_{l=1}^L\lambda_l(||\boldsymbol{\mu}_l||_\infty+||\boldsymbol{\Sigma}_l^{\frac{1}{2}}||)^4})+\log\frac{4}{\delta}}{n}},
$$

$$
\frac{dK\log(36rnd^{\frac{25}{2}}K^{\frac{7}{2}}\cdot\sqrt{\sum_{l=1}^L\lambda_l(||\boldsymbol{\mu}_l||_\infty+||\boldsymbol{\Sigma}_l^{\frac{1}{2}}||)^2\sum_{l=1}^L\lambda_l(||\boldsymbol{\mu}_l||_\infty+||\boldsymbol{\Sigma}_l^{\frac{1}{2}}||)^4})+\log\frac{4}{\delta}}{\sum_{l=1}^L\lambda_l(||\boldsymbol{\mu}_l||+||\boldsymbol{\Sigma}_l^{\frac{1}{2}}||)^2n}\Big\}
\tag{214}
$$

So by setting $t=\sum_{l=1}^L\lambda_l(||\boldsymbol{\mu}_l||+||\boldsymbol{\Sigma}_l^{\frac{1}{2}}||)^2\sqrt{\frac{dK\log n}{n}}$, as long as $n\geq C'\cdot dK\log dK$, we have

$$
\mathbb{P}\Big(\sup_{\boldsymbol{W}\in\mathbb{B}(\boldsymbol{W}^*\boldsymbol{P},r)}||\nabla^2 f_n(\boldsymbol{W})-\nabla^2\bar{f}(\boldsymbol{W})||\geq C_6\cdot\sum_{l=1}^L\lambda_l(||\boldsymbol{\mu}_l||+||\boldsymbol{\Sigma}_l^{\frac{1}{2}}||)^2\sqrt{\frac{dK\log n}{n}}\Big)\leq d^{-10}
\tag{215}
$$

# F  PROOF OF LEMMA 2

We first present a lemma used in proving Lemma 2 in Section F.1 and then prove Lemma 2 in Section F.2.

### F.1 A USEFUL LEMMA USED IN THE PROOF

**Lemma 9.** *If $r$ is defined in (93) for $\epsilon_0 \in (0, \frac{1}{4})$, then with probability at least $1 - d^{-10}$, we have[12]*

$$\sup_{\boldsymbol{W} \in \mathbb{B}(\boldsymbol{W}^*\boldsymbol{P}, r)} ||\nabla \tilde{f}_n(\boldsymbol{W}) - \nabla \tilde{f}(\boldsymbol{W})|| \leq C_{13} \cdot \sqrt{K \sum_{l=1}^{L} \lambda_l (\|\boldsymbol{\mu}_l\| + \|\boldsymbol{\Sigma}_l\|)^2} \sqrt{\frac{d \log n}{n}} (1 + \xi) \quad (216)$$

*for some constant $C_{13} > 0$, where $\boldsymbol{P}$ is a permutation matrix.*

**Proof:**

Note that $\nabla \tilde{f}_n(\boldsymbol{W}) = \nabla f_n(\boldsymbol{W}) + \frac{1}{n} \sum_{i=1}^n \nu_i$, $\nabla \tilde{f}(\boldsymbol{W}) = \nabla \bar{f}(\boldsymbol{W}) + \mathbb{E}[\nu_i] = \nabla \bar{f}(\boldsymbol{W})$. Therefore, we have

$$\sup_{\boldsymbol{W} \in \mathbb{B}(\boldsymbol{W}^*\boldsymbol{P}, r)} ||\nabla \tilde{f}_n(\boldsymbol{W}) - \nabla \tilde{f}(\boldsymbol{W})|| \leq \sup_{\boldsymbol{W} \in \mathbb{B}(\boldsymbol{W}^*\boldsymbol{P}, r)} ||\nabla f_n(\boldsymbol{W}) - \nabla \bar{f}(\boldsymbol{W})|| + \|\frac{1}{n} \sum_{i=1}^n \nu_i\| \quad (217)$$

Then, similar to the idea of the proof of Lemma 8, we adopt an $\epsilon$-covering net of the ball $\mathbb{B}(\boldsymbol{W}^*, r)$ to build a relationship between any arbitrary point in the ball and the points in the covering set. We can then divide the distance between $\nabla f_n(\boldsymbol{W})$ and $\nabla \bar{f}(\boldsymbol{W})$ into three parts, similar to (195). (218) to (220) can be derived in a similar way as (197) to (199), with "$\nabla^2$" replaced by "$\nabla$". Then we need to bound $\mathbb{P}(A_t')$, $\mathbb{P}(B_t')$ and $\mathbb{P}(C_t')$ respectively, where $A_t'$, $B_t'$ and $C_t'$ are defined below.

$$A_t' = \{\sup_{\boldsymbol{W} \in \mathbb{B}(\boldsymbol{W}^*\boldsymbol{P}, r)} \frac{1}{n} || \sum_{i=1}^n [\nabla \ell(\boldsymbol{W}; \boldsymbol{x}_i) - \nabla \ell(\boldsymbol{W}_{j(\boldsymbol{W})}; \boldsymbol{x}_i)]|| \geq \frac{t}{3}\} \quad (218)$$

$$B_t' = \{\sup_{\boldsymbol{W} \in \mathbb{B}(\boldsymbol{W}^*\boldsymbol{P}, r)} ||\frac{1}{n} \sum_{i=1}^n \nabla \ell(\boldsymbol{W}_{j(\boldsymbol{W})}; \boldsymbol{x}_i) - \mathbb{E}_{\boldsymbol{x} \sim \sum_{l=1}^L \lambda_l \mathcal{N}(\boldsymbol{\mu}_l, \boldsymbol{\Sigma}_l)}[\nabla \ell(\boldsymbol{W}_{j(\boldsymbol{W})}; \boldsymbol{x}_i)]|| \geq \frac{t}{3}\} \quad (219)$$

$$C_t' = \{\sup_{\boldsymbol{W} \in \mathbb{B}(\boldsymbol{W}^*\boldsymbol{P}, r)} ||\mathbb{E}_{\boldsymbol{x} \sim \sum_{l=1}^L \lambda_l \mathcal{N}(\boldsymbol{\mu}_l, \boldsymbol{\Sigma}_l)}[\nabla \ell(\boldsymbol{W}_{j(\boldsymbol{W})}; \boldsymbol{x}_i)]$$
$$- \mathbb{E}_{\boldsymbol{x} \sim \sum_{l=1}^L \lambda_l \mathcal{N}(\boldsymbol{\mu}_l, \boldsymbol{\Sigma}_l)}[\nabla \ell(\boldsymbol{W}; \boldsymbol{x}_i)]|| \geq \frac{t}{3}\} \quad (220)$$

(a) Upper bound of $\mathbb{P}(B_t')$. Applying Lemma 3 in Mei et al. (2016), we have

$$||\frac{1}{n} \sum_{i=1}^n \nabla \ell(\boldsymbol{W}_{j(\boldsymbol{W})}; \boldsymbol{x}_i) - \mathbb{E}_{\boldsymbol{x} \sim \sum_{l=1}^L \lambda_l \mathcal{N}(\boldsymbol{\mu}_l, \boldsymbol{\Sigma}_l)}[\nabla \ell(\boldsymbol{W}_{j(\boldsymbol{W})}; \boldsymbol{x}_i)]||$$
$$\leq 2 \sup_{\boldsymbol{v} \in V_{\frac{1}{2}}} \left| \left\langle \frac{1}{n} \sum_{i=1}^n \nabla \ell(\boldsymbol{W}_{j(\boldsymbol{W})}; \boldsymbol{x}_i) - \mathbb{E}_{\boldsymbol{x} \sim \sum_{l=1}^L \lambda_l \mathcal{N}(\boldsymbol{\mu}_l, \boldsymbol{\Sigma}_l)}[\nabla \ell(\boldsymbol{W}_{j(\boldsymbol{W})}; \boldsymbol{x}_i)], \boldsymbol{v} \right\rangle \right| \quad (221)$$

Define $G_i' = \left\langle \boldsymbol{v}, (\nabla \ell(\boldsymbol{W}, \boldsymbol{x}_i) - \mathbb{E}_{\boldsymbol{x} \sim \sum_{l=1}^L \lambda_l \mathcal{N}(\boldsymbol{\mu}_l, \boldsymbol{\Sigma}_l)}[\nabla \ell(\boldsymbol{W}, \boldsymbol{x}_i)]) \right\rangle$. Here $\boldsymbol{v} \in \mathbb{R}^d$. To compute $\nabla \ell(\boldsymbol{W}, \boldsymbol{x}_i)$, we require the derivation in Property 9. Then we can have an upper bound of $\zeta(\boldsymbol{W})$ in (83).

$$\zeta(\boldsymbol{W}) = \begin{cases} \left| -\frac{1}{K} \frac{1}{H(\boldsymbol{W})} \phi'(\boldsymbol{w}_j^\top \boldsymbol{x}) \right| \leq \frac{\phi(\boldsymbol{w}_j^\top \boldsymbol{x})(1 - \phi(\boldsymbol{w}_j^\top \boldsymbol{x}))}{K \cdot \frac{1}{K} \phi(\boldsymbol{w}_j^\top \boldsymbol{x})} \leq 1, & y = 1 \\ \left| \frac{1}{K} \frac{1}{1 - H(\boldsymbol{W})} \phi'(\boldsymbol{w}_j^\top \boldsymbol{x}) \right| \leq \frac{\phi(\boldsymbol{w}_j^\top \boldsymbol{x})(1 - \phi(\boldsymbol{w}_j^\top \boldsymbol{x}))}{K \cdot \frac{1}{K} (1 - \phi(\boldsymbol{w}_j^\top \boldsymbol{x}))} \leq 1, & y = 0 \end{cases} \quad (222)$$

Then we have an upper bound of $G_i'$.

$$|G_i'| = \left| \zeta_{j,l} \boldsymbol{v}^\top \boldsymbol{x} - \mathbb{E}_{\boldsymbol{x} \sim \sum_{l=1}^L \lambda_l \mathcal{N}(\boldsymbol{\mu}_l, \boldsymbol{\Sigma}_l)}[\zeta \boldsymbol{v}^\top \boldsymbol{x}] \right|$$
$$\leq |\boldsymbol{v}^\top \boldsymbol{x}| + \mathbb{E}_{\boldsymbol{x} \sim \sum_{l=1}^L \lambda_l \mathcal{N}(\boldsymbol{\mu}_l, \boldsymbol{\Sigma}_l)}[|\boldsymbol{v}^\top \boldsymbol{x}|] \quad (223)$$

---

[12]$\nabla \tilde{f}_n(\boldsymbol{W})$ is defined as $\frac{1}{n} \sum_{i=1}^n (\nabla l(\boldsymbol{W}, \boldsymbol{x}_i, y_i) + \nu_i)$ in algorithm 1

Following the idea of (203) and (204), and by $\boldsymbol{v} \in V_{\frac{1}{2}}$, we have

$$\mathbb{E}[|G_i'|^p] \leq O\Big(\Big(\sum_{l=1}^{L} \lambda_l (\|\boldsymbol{\mu}_l\| + \|\boldsymbol{\Sigma}_l^{\frac{1}{2}}\|)^2\Big)^{\frac{p}{2}}\Big) \tag{224}$$

$$\mathbb{E}[\exp(\theta G_i')] \leq 1 + O\Big(|e\theta^2| \sum_{l=1}^{L} \lambda_l (\|\boldsymbol{\mu}_l\| + \|\boldsymbol{\Sigma}_l^{\frac{1}{2}}\|)^2\Big) \tag{225}$$

where (225) holds if $\theta \leq \frac{27}{4e}\sqrt{\sum_{l=1}^{L} \lambda_l (\|\boldsymbol{\mu}_l\| + \|\boldsymbol{\Sigma}_l\|)^2}$. Following the derivation of (201) and (206) to (209), we have

$$\mathbb{P}(|\frac{1}{n}\sum_{i=1}^{n} G_i'| \geq \frac{t}{6})$$
$$\leq 2\exp\Big(\max\Big\{-\frac{C_{14}nt^2}{144\sum_{l=1}^{L} \lambda_l (\|\boldsymbol{\mu}_l\| + \|\boldsymbol{\Sigma}_l^{\frac{1}{2}}\|)^2}, -C_{15}n\sqrt{\sum_{l=1}^{L} \lambda_l (\|\boldsymbol{\mu}_l\| + \|\boldsymbol{\Sigma}_l^{\frac{1}{2}}\|)^2 \cdot t}\Big\}\Big) \tag{226}$$

for some constant $C_{14} > 0$ and $C_{15} > 0$. Moreover, we can obtain $\mathbb{P}(B_t') \leq \frac{\delta}{2}$ as long as

$$t \geq C_{13} \cdot \max\Big\{\sqrt{\sum_{l=1}^{L} \lambda_l (\|\boldsymbol{\mu}_l\| + \|\boldsymbol{\Sigma}_l^{\frac{1}{2}}\|)^2}\sqrt{\frac{dK\log\frac{18r}{\epsilon} + \log\frac{4}{\delta}}{n}}, \frac{dK\log\frac{18r}{\epsilon} + \log\frac{4}{\delta}}{\sqrt{\sum_{l=1}^{L} \lambda_l (\|\boldsymbol{\mu}_l\| + \|\boldsymbol{\Sigma}_l^{\frac{1}{2}}\|)^2} \cdot n}\Big\} \tag{227}$$

(b) For the upper bound of $\mathbb{P}(A_t')$ and $\mathbb{P}(C_t')$, we can first derive

$$\mathbb{E}_{\boldsymbol{x} \sim \sum_{l=1}^{L} \lambda_l \mathcal{N}(\boldsymbol{\mu}_l, \boldsymbol{\Sigma}_l)}\Big[\sup_{\boldsymbol{W} \neq \boldsymbol{W}' \in \mathbb{B}(\boldsymbol{W}^*\boldsymbol{P}, r)} \frac{\|\nabla\ell(\boldsymbol{W}, \boldsymbol{x}) - \nabla\ell(\boldsymbol{W}', \boldsymbol{x})\|}{\|\boldsymbol{W} - \boldsymbol{W}'\|_F}\Big]$$

$$\leq \mathbb{E}_{\boldsymbol{x} \sim \sum_{l=1}^{L} \lambda_l \mathcal{N}(\boldsymbol{\mu}_l, \boldsymbol{\Sigma}_l)}\Big[\sup_{\boldsymbol{W} \neq \boldsymbol{W}' \in \mathbb{B}(\boldsymbol{W}^*\boldsymbol{P}, r)} \frac{|\zeta(\boldsymbol{W}) - \zeta(\boldsymbol{W}')| \cdot \|\boldsymbol{x}\|}{\|\boldsymbol{W} - \boldsymbol{W}'\|_F}\Big]$$

$$\leq \mathbb{E}_{\boldsymbol{x} \sim \sum_{l=1}^{L} \lambda_l \mathcal{N}(\boldsymbol{\mu}_l, \boldsymbol{\Sigma}_l)}\Big[\sup_{\boldsymbol{W} \neq \boldsymbol{W}' \in \mathbb{B}(\boldsymbol{W}^*\boldsymbol{P}, r)} \frac{\max_{1 \leq j, l \leq K}\{|\xi_{j,l}(\boldsymbol{W}'')|\} \cdot \|\boldsymbol{x}\|^2 \sqrt{K}\|\boldsymbol{W} - \boldsymbol{W}'\|_F}{\|\boldsymbol{W} - \boldsymbol{W}'\|_F}\Big]$$

$$\leq \mathbb{E}_{\boldsymbol{x} \sim \sum_{l=1}^{L} \lambda_l \mathcal{N}(\boldsymbol{\mu}_l, \boldsymbol{\Sigma}_l)}\Big[\sup_{\boldsymbol{W} \neq \boldsymbol{W}' \in \mathbb{B}(\boldsymbol{W}^*\boldsymbol{P}, r)} \frac{C_9 \cdot \|\boldsymbol{x}\|^2 \sqrt{K}\|\boldsymbol{W} - \boldsymbol{W}'\|_F}{\|\boldsymbol{W} - \boldsymbol{W}'\|_F}\Big]$$

$$\leq C_9 \cdot 3\sqrt{K}d \cdot \sum_{l=1}^{L} \lambda_l (\|\boldsymbol{\mu}_l\|_\infty + \|\boldsymbol{\Sigma}_l^{\frac{1}{2}}\|)^2 \tag{228}$$

The first inequality is by (83). The second inequality is by the Mean Value Theorem. The third step is by (164). The last inequality is by Property 7. Therefore, following the steps in part (2) of Lemma 8, we can conclude that $C_t'$ holds if

$$t \geq 3C_9 \cdot \sqrt{K}d \cdot \sum_{l=1}^{L} \lambda_l (\|\boldsymbol{\mu}_l\|_\infty + \|\boldsymbol{\Sigma}_l^{\frac{1}{2}}\|)^2 \cdot \epsilon \tag{229}$$

Moreover, from (213) in Lemma 8 we have that

$$t \geq \frac{18C_9 \cdot \sqrt{K}d \cdot \sum_{l=1}^{L} \lambda_l (\|\boldsymbol{\mu}_l\|_\infty + \|\boldsymbol{\Sigma}_l\|)^2 \cdot \epsilon}{\delta} \tag{230}$$

ensures $\mathbb{P}(A_t') \leq \frac{\delta}{2}$. Therefore, let $\epsilon = \frac{\delta}{18C_9 \cdot \sqrt{K}d \cdot \sum_{l=1}^{L} \lambda_l (\|\boldsymbol{\mu}_l\|_\infty + \|\boldsymbol{\Sigma}_l\|)^2 \cdot \epsilon \cdot ndK}$, $\delta = d^{-10}$ and $t = C_{13}\sqrt{K\sum_{l=1}^{L} \lambda_l (\|\boldsymbol{\mu}_l\| + \|\boldsymbol{\Sigma}_l\|)^2}\sqrt{\frac{d\log n}{n}}$, if $n \geq C'' \cdot dK \log dK$ for some constant $C'' > 0$, we have

$$\mathbb{P}\Big(\sup_{\boldsymbol{W} \in \mathbb{B}(\boldsymbol{W}^*\boldsymbol{P}, r)} \|\nabla f_n(\boldsymbol{W}) - \nabla\bar{f}(\boldsymbol{W})\|\Big) \geq C_{13} \cdot \sqrt{K\sum_{l=1}^{L} \lambda_l (\|\boldsymbol{\mu}_l\| + \|\boldsymbol{\Sigma}_l\|)^2}\sqrt{\frac{d\log n}{n}} \leq d^{-10} \tag{231}$$

By Hoeffding's inequality in Vershynin (2010) and Property 2, we have

$$\mathbb{P}\Big(\frac{1}{n}\sum_{i=1}^{n}\|\nu_i\|_F \geq C_{13} \cdot \sqrt{\sum_{l=1}^{L}\lambda_l(\|\boldsymbol{\mu}_l\| + \|\boldsymbol{\Sigma}_l^{\frac{1}{2}}\|)^2}\sqrt{\frac{dK\log n}{n}}\xi\Big)$$

$$\lesssim \exp(-C_{13}^2 \cdot \sum_{l=1}^{L}\lambda_l(\|\boldsymbol{\mu}_l\| + \|\boldsymbol{\Sigma}_l^{\frac{1}{2}}\|)^2\frac{\xi^2 dK\log n}{dK\xi^2}) \tag{232}$$

$$\lesssim d^{-10}$$

Therefore,

$$\sup_{\boldsymbol{W}\in\mathbb{B}(\boldsymbol{W}^*\boldsymbol{P},r)}\|\nabla\tilde{f}_n(\boldsymbol{W}) - \nabla\tilde{f}(\boldsymbol{W})\|$$

$$\leq C_{13} \cdot \sqrt{K\sum_{l=1}^{L}\lambda_l(\|\boldsymbol{\mu}_l\| + \|\boldsymbol{\Sigma}_l^{\frac{1}{2}}\|)^2}\sqrt{\frac{d\log n}{n}} + \frac{1}{n}\sum_{i=1}^{n}\|\nu_i\|$$

$$\leq C_{13} \cdot \sqrt{K\sum_{l=1}^{L}\lambda_l(\|\boldsymbol{\mu}_l\| + \|\boldsymbol{\Sigma}_l^{\frac{1}{2}}\|)^2}\sqrt{\frac{d\log n}{n}} + \frac{1}{n}\sum_{i=1}^{n}\|\nu_i\|_F \tag{233}$$

$$\leq C_{13} \cdot \sqrt{K\sum_{l=1}^{L}\lambda_l(\|\boldsymbol{\mu}_l\| + \|\boldsymbol{\Sigma}_l^{\frac{1}{2}}\|)^2}\sqrt{\frac{d\log n}{n}}(1 + \xi)$$

### F.2 PROOF OF LEMMA 2

Following the proof of Theorem 2 in Fu et al. (2020), first, we have Taylor's expansion of $f_n(\widehat{\boldsymbol{W}}_n)$

$$f_n(\widehat{\boldsymbol{W}}_n) = f_n(\boldsymbol{W}^*\boldsymbol{P}) + \Big\langle\nabla\tilde{f}_n(\boldsymbol{W}^*\boldsymbol{P}), \text{vec}(\widehat{\boldsymbol{W}}_n - \boldsymbol{W}^*\boldsymbol{P})\Big\rangle$$

$$+ \frac{1}{2}\text{vec}(\widehat{\boldsymbol{W}}_n - \boldsymbol{W}^*\boldsymbol{P})\nabla^2 f_n(\boldsymbol{W}')\text{vec}(\widehat{\boldsymbol{W}}_n - \boldsymbol{W}^*\boldsymbol{P}) \tag{234}$$

Here $\boldsymbol{W}'$ is on the straight line connecting $\boldsymbol{W}^*\boldsymbol{P}$ and $\widehat{\boldsymbol{W}}_n$. By the fact that $f_n(\widehat{\boldsymbol{W}}_n) \leq f_n(\boldsymbol{W}^*\boldsymbol{P})$, we have

$$\frac{1}{2}\text{vec}(\widehat{\boldsymbol{W}}_n - \boldsymbol{W}^*\boldsymbol{P})\nabla^2 f_n(\boldsymbol{W}')\text{vec}(\widehat{\boldsymbol{W}}_n - \boldsymbol{W}^*\boldsymbol{P}) \leq \Big|\nabla f_n(\boldsymbol{W}^*\boldsymbol{P})^\top\text{vec}(\widehat{\boldsymbol{W}}_n - \boldsymbol{W}^*\boldsymbol{P})\Big| \tag{235}$$

From Lemma 7 and Lemma 9, we have

$$\frac{4}{K^2}\sum_{l=1}^{L}\lambda_l\frac{\|\boldsymbol{\Sigma}_l^{-1}\|^{-1}}{\eta\tau^K\kappa^2}\rho\Big(\frac{\boldsymbol{W}^{*\top}\boldsymbol{\mu}_l}{\delta_K(\boldsymbol{W}^*)\|\boldsymbol{\Sigma}_l^{-1}\|^{-\frac{1}{2}}}, \delta_K(\boldsymbol{W}^*)\|\boldsymbol{\Sigma}_l^{-1}\|^{-\frac{1}{2}}\Big)\|\widehat{\boldsymbol{W}}_n - \boldsymbol{W}^*\boldsymbol{P}\|_F^2$$

$$\leq \frac{1}{2}\text{vec}(\widehat{\boldsymbol{W}}_n - \boldsymbol{W}^*\boldsymbol{P})\nabla^2 f_n(\boldsymbol{W}')\text{vec}(\widehat{\boldsymbol{W}}_n - \boldsymbol{W}^*\boldsymbol{P}) \tag{236}$$

and

$$\Big|\nabla\tilde{f}_n(\boldsymbol{W}^*\boldsymbol{P})^\top\text{vec}(\widehat{\boldsymbol{W}}_n - \boldsymbol{W}^*\boldsymbol{P})\Big|$$

$$\leq\|\nabla\tilde{f}_n(\boldsymbol{W}^*\boldsymbol{P})\| \cdot \|\widehat{\boldsymbol{W}}_n - \boldsymbol{W}^*\boldsymbol{P}\|_F$$

$$\leq(\|\nabla\tilde{f}_n(\boldsymbol{W}^*\boldsymbol{P}) - \nabla\tilde{f}(\boldsymbol{W}^*\boldsymbol{P})\| + \|\nabla\tilde{f}(\boldsymbol{W}^*\boldsymbol{P})\|) \cdot \|\widehat{\boldsymbol{W}}_n - \boldsymbol{W}^*\boldsymbol{P}\|_F \tag{237}$$

$$\leq O\Big(\sqrt{K\sum_{l=1}^{L}\lambda_l(\|\boldsymbol{\mu}_l\| + \|\boldsymbol{\Sigma}_l^{\frac{1}{2}}\|)^2}\sqrt{\frac{d\log n}{n}}(1 + \xi)\Big)\|\widehat{\boldsymbol{W}}_n - \boldsymbol{W}^*\boldsymbol{P}\|_F$$

The second to last step of (237) comes from the triangle inequality, and the last step follows from the fact $\nabla\tilde{f}(\boldsymbol{W}^*\boldsymbol{P}) = 0$. Combining (235), (236) and (237), we have

$$||\widehat{\boldsymbol{W}}_n - \boldsymbol{W}^*\boldsymbol{P}||_F \leq O\Big(\frac{K^{\frac{5}{2}}\sqrt{\sum_{l=1}^{L}\lambda_l(||\boldsymbol{\mu}_l|| + ||\boldsymbol{\Sigma}_l^{\frac{1}{2}}||)^2}(1+\xi)}{\sum_{l=1}^{L}\lambda_l\frac{||\boldsymbol{\Sigma}_l^{-1}||^{-1}}{\eta\tau^K\kappa^2}\rho(\frac{\boldsymbol{W}^{*\top}\boldsymbol{\mu}_l}{\delta_K(\boldsymbol{W}^*)||\boldsymbol{\Sigma}_l^{-1}||^{-\frac{1}{2}}}, \delta_K(\boldsymbol{W}^*)||\boldsymbol{\Sigma}_l^{-1}||^{-\frac{1}{2}})}\sqrt{\frac{d\log n}{n}}\Big)$$
(238)

Therefore, we have concluded that there indeed exists a critical point $\widehat{\boldsymbol{W}}$ in $\mathbb{B}(\boldsymbol{W}^*\boldsymbol{P}, r)$. Then we show the linear convergence of Algorithm 1 as below. By the update rule, we have

$$\boldsymbol{W}_{t+1} - \widehat{\boldsymbol{W}}_n = \boldsymbol{W}_t - \eta_0(\nabla f_n(\boldsymbol{W}_t) + \frac{1}{n}\sum_{i=1}^{n}\nu_i) - (\widehat{\boldsymbol{W}}_n - \eta_0\nabla f_n(\widehat{\boldsymbol{W}}_n))$$
$$= \Big(\boldsymbol{I} - \eta_0\int_0^1 \nabla^2 f_n(\boldsymbol{W}(\gamma))\Big)(\boldsymbol{W}_t - \widehat{\boldsymbol{W}}_n) - \frac{\eta_0}{n}\sum_{i=1}^{n}\nu_i$$
(239)

where $\boldsymbol{W}(\gamma) = \gamma\widehat{\boldsymbol{W}}_n + (1-\gamma)\boldsymbol{W}_t$ for $\gamma \in (0,1)$. Since $\boldsymbol{W}(\gamma) \in \mathbb{B}(\boldsymbol{W}^*\boldsymbol{P}, r)$, by Lemma 1, we have

$$H_{\min} \cdot \boldsymbol{I} \preceq \nabla^2 f_n(\boldsymbol{W}(\gamma)) \leq H_{\max} \cdot \boldsymbol{I}$$
(240)

where $H_{\min} = \Omega\Big(\frac{1}{K^2}\sum_{l=1}^{L}\lambda_l\frac{||\boldsymbol{\Sigma}_l^{-1}||^{-1}}{\eta\tau^K\kappa^2}\rho(\frac{\boldsymbol{W}^{*\top}\boldsymbol{\mu}_l}{\delta_K(\boldsymbol{W}^*)||\boldsymbol{\Sigma}_l^{-1}||^{-\frac{1}{2}}}, \delta_K(\boldsymbol{W}^*)||\boldsymbol{\Sigma}_l^{-1}||^{-\frac{1}{2}})\Big)$, $H_{\max} = C_4 \cdot \sum_{l=1}^{L}\lambda_l(||\boldsymbol{\mu}_l|| + ||\boldsymbol{\Sigma}_l||)^2$. Therefore,

$$||\boldsymbol{W}_{t+1} - \widehat{\boldsymbol{W}}_n||_F = ||\boldsymbol{I} - \eta_0\int_0^1 \nabla^2 f_n(\boldsymbol{W}(\gamma))|| \cdot ||\boldsymbol{W}_t - \widehat{\boldsymbol{W}}_n||_F + ||\frac{\eta_0}{n}\sum_{i=1}^{n}\nu_i||_F$$
$$\leq (1 - \eta_0 H_{\min})||\boldsymbol{W}_t - \widehat{\boldsymbol{W}}_n||_F + ||\frac{\eta_0}{n}\sum_{i=1}^{n}\nu_i||_F$$
(241)

By setting $\eta_0 = \frac{1}{H_{\max}} = O\Big(\frac{1}{\sum_{l=1}^{L}\lambda_l(||\boldsymbol{\mu}_l||+||\boldsymbol{\Sigma}_l||)^2}\Big)$, we obtain

$$||\widehat{\boldsymbol{W}}_{t+1} - \widehat{\boldsymbol{W}}_n||_F \leq (1 - \frac{H_{\min}}{H_{\max}})||\boldsymbol{W}_t - \widehat{\boldsymbol{W}}_n||_F + \frac{\eta_0}{n}\sum_{i=1}^{n}||\nu_i||_F$$
(242)

Therefore, Algorithm 1 converges linearly to the local minimizer with an extra statistical error. By Hoeffding's inequality in Vershynin (2010) and Property 2, we have

$$\mathbb{P}\Big(\frac{1}{n}\sum_{i=1}^{n}||\nu_i||_F \geq \sqrt{\frac{dK\log n}{n}}\xi\Big) \lesssim \exp(-\frac{\xi^2 dK\log n}{dK\xi^2}) \lesssim d^{-10}$$
(243)

Therefore, with probability $1 - d^{-10}$ we can derive

$$||\widehat{\boldsymbol{W}}_t - \widehat{\boldsymbol{W}}_n||_F \leq (1 - \frac{H_{\min}}{H_{\max}})^t||\boldsymbol{W}_0 - \widehat{\boldsymbol{W}}_n||_F + \frac{H_{\max}\eta_0}{H_{\min}}\sqrt{\frac{dK\log n}{n}}\xi$$
(244)

## G PROOF OF LEMMA 3

We need Lemma 10 to Lemma 14, which are stated in Section G.1, for the proof of Lemma 3. Section G.2 summarizes the proof of Lemma 3. The proofs of Lemma 10 to Lemma 12 are provided in Section G.3 to Section G.5. Lemma 13 and Lemma 14 are cited from Zhong et al. (2017b). Although Zhong et al. (2017b) considers the standard Gaussian distribution, the proofs of Lemma 13 and 14 hold for any data distribution. Therefore, these two lemmas can be applied here directly.

The tensor initialization in Zhong et al. (2017b) only holds for the standard Gaussian distribution. We exploit a more general definition of tensors from Janzamin et al. (2014) for the tensor initialization in our algorithm. We also develop new error bounds for the initialization.

### G.1 USEFUL LEMMAS IN THE PROOF

**Lemma 10.** *Let $\boldsymbol{Q}_2$ and $\boldsymbol{Q}_3$ follow Definition 1. Let $S$ be a set of i.i.d. samples generated from the mixed Gaussian distribution $\sum_{l=1}^{L} \lambda_l \mathcal{N}(\boldsymbol{\mu}_l, \boldsymbol{\Sigma}_l)$. Let $\widehat{\boldsymbol{Q}}_2$, $\widehat{\boldsymbol{Q}}_3$ be the empirical version of $\boldsymbol{Q}_2$, $\boldsymbol{Q}_3$ using data set $S$, respectively. Then with a probability at least $1 - 2n^{-\Omega(\delta_1(\boldsymbol{W}^*)^4 d)}$, we have*

$$||\boldsymbol{Q}_2 - \widehat{\boldsymbol{Q}}_2|| \lesssim \sqrt{\frac{d \log n}{n}} \cdot \delta_1(\boldsymbol{W}^*)^2 \cdot \tau^6 \sqrt{D_2(\Psi)D_4(\Psi)} \tag{245}$$

*if the mixed Gaussian distribution is not symmetric. We also have*

$$||\boldsymbol{Q}_3(\boldsymbol{I}_d, \boldsymbol{I}_d, \boldsymbol{\alpha}) - \widehat{\boldsymbol{Q}}_3(\boldsymbol{I}_d, \boldsymbol{I}_d, \boldsymbol{\alpha})|| \lesssim \sqrt{\frac{d \log n}{n}} \cdot \delta_1(\boldsymbol{W}^*)^2 \cdot \tau^6 \sqrt{D_2(\Psi)D_4(\Psi)} \tag{246}$$

*for any arbitrary vector $\boldsymbol{\alpha} \in \mathbb{R}^d$, if the mixed Gaussian distribution is symmetric.*

**Lemma 11.** *Let $\boldsymbol{U} \in \mathbb{E}^{d \times K}$ be the orthogonal column span of $\boldsymbol{W}^*$. Let $\boldsymbol{\alpha}$ be a fixed unit vector and $\widehat{\boldsymbol{U}} \in \mathbb{R}^{d \times K}$ denote an orthogonal matrix satisfying $||\boldsymbol{U}\boldsymbol{U}^\top - \widehat{\boldsymbol{U}}\widehat{\boldsymbol{U}}^\top|| \le \frac{1}{4}$. Define $\boldsymbol{R}_3 = \boldsymbol{Q}_3(\widehat{\boldsymbol{U}}, \widehat{\boldsymbol{U}}, \widehat{\boldsymbol{U}})$, where $\boldsymbol{Q}_3$ is defined in Definition 1. Let $\widehat{\boldsymbol{R}}_3$ be the empirical version of $\boldsymbol{R}_3$ using data set $S$, where each sample of $S$ is i.i.d. sampled from the mixed Gaussian distribution $\sum_{l=1}^{L} \lambda_l \mathcal{N}(\boldsymbol{\mu}_l, \boldsymbol{\Sigma}_l)$. Then with a probability at least $1 - n^{-\Omega(\delta^4(\boldsymbol{W}^*))}$, we have*

$$||\widehat{\boldsymbol{R}}_3 - \boldsymbol{R}_3|| \lesssim \delta_1(\boldsymbol{W}^*)^2 \cdot \left(\tau^6 \sqrt{D_6(\Psi)}\right) \cdot \sqrt{\frac{\log n}{n}} \tag{247}$$

**Lemma 12.** *Let $\widehat{\boldsymbol{Q}}_1$ be the empirical version of $\boldsymbol{Q}_1$ using dataset $S$. Then with a probability at least $1 - 2n^{-\Omega(d)}$, we have*

$$||\widehat{\boldsymbol{Q}}_1 - \boldsymbol{Q}_1|| \lesssim \left(\tau^2 \sqrt{D_2(\Psi)}\right) \cdot \sqrt{\frac{d \log n}{n}} \tag{248}$$

**Lemma 13.** *(Zhong et al. (2017b), Lemma E.6) Let $\boldsymbol{Q}_2$, $\boldsymbol{Q}_3$ be defined in Definition 1 and $\widehat{\boldsymbol{Q}}_2$, $\widehat{\boldsymbol{Q}}_3$ be their empirical version, respectively. Let $\boldsymbol{U} \in \mathbb{R}^{d \times K}$ be the column span of $\boldsymbol{W}^*$. Assume $||\boldsymbol{Q}_2 - \widehat{\boldsymbol{Q}}_2|| \le \frac{\delta_K(\boldsymbol{Q}_2)}{10}$ for non-symmetric distribution cases and $||\boldsymbol{Q}_3(\boldsymbol{I}_d, \boldsymbol{I}_d, \boldsymbol{\alpha}) - \widehat{\boldsymbol{Q}}_3(\boldsymbol{I}_d, \boldsymbol{I}_d, \boldsymbol{\alpha})|| \le \frac{\delta_K(\boldsymbol{Q}_3(\boldsymbol{I}_d, \boldsymbol{I}_d, \boldsymbol{\alpha}))}{10}$ for symmetric distribution cases and any arbitrary vector $\boldsymbol{\alpha} \in \mathbb{R}^d$. Then after $T = O(\log(\frac{1}{\epsilon}))$ iterations, the output of the Tensor Initialization Method 1, $\widehat{\boldsymbol{U}}$ will satisfy*

$$||\widehat{\boldsymbol{U}}\widehat{\boldsymbol{U}}^\top - \boldsymbol{U}\boldsymbol{U}^\top|| \lesssim \frac{||\widehat{\boldsymbol{Q}}_2 - \boldsymbol{Q}_2||}{\delta_K(\boldsymbol{Q}_2)} + \epsilon, \tag{249}$$

*which implies*

$$||(\boldsymbol{I} - \widehat{\boldsymbol{U}}\widehat{\boldsymbol{U}}^\top)\boldsymbol{w}_i^*|| \lesssim (\frac{||\boldsymbol{Q}_2 - \widehat{\boldsymbol{Q}}_2||}{\delta_K(\boldsymbol{Q}_2)} + \epsilon)||\boldsymbol{w}_i^*|| \tag{250}$$

*if the mixed Gaussian distribution is not symmetric. Similarly, we have*

$$||\widehat{\boldsymbol{U}}\widehat{\boldsymbol{U}}^\top - \boldsymbol{U}\boldsymbol{U}^\top|| \lesssim \frac{||\widehat{\boldsymbol{Q}}_3(\boldsymbol{I}_d, \boldsymbol{I}_d, \boldsymbol{\alpha}) - \boldsymbol{Q}_3(\boldsymbol{I}_d, \boldsymbol{I}_d, \boldsymbol{\alpha})||}{\delta_K(\boldsymbol{Q}_3(\boldsymbol{I}_d, \boldsymbol{I}_d, \boldsymbol{\alpha}))} + \epsilon, \tag{251}$$

*which implies*

$$||(\boldsymbol{I} - \widehat{\boldsymbol{U}}\widehat{\boldsymbol{U}}^\top)\boldsymbol{w}_i^*|| \lesssim (\frac{||\boldsymbol{Q}_3(\boldsymbol{I}_d, \boldsymbol{I}_d, \boldsymbol{\alpha}) - \widehat{\boldsymbol{Q}}_3(\boldsymbol{I}_d, \boldsymbol{I}_d, \boldsymbol{\alpha})||}{\delta_K(\boldsymbol{Q}_3(\boldsymbol{I}_d, \boldsymbol{I}_d, \boldsymbol{\alpha}))} + \epsilon)||\boldsymbol{w}_i^*|| \tag{252}$$

*if the mixed Gaussian distribution is symmetric.*

**Lemma 14.** *(Zhong et al. (2017b), Lemma E.13) Let $\boldsymbol{U} \in \mathbb{R}^{d \times K}$ be the orthogonal column span of $\boldsymbol{W}^*$. Let $\widehat{\boldsymbol{U}} \in \mathbb{R}^{d \times K}$ be an orthogonal matrix such that $||\boldsymbol{U}\boldsymbol{U}^\top - \widehat{\boldsymbol{U}}\widehat{\boldsymbol{U}}^\top|| \lesssim \gamma_1 \lesssim \frac{1}{\kappa^2 \sqrt{K}}$. For each $i \in [K]$, let $\widehat{\boldsymbol{v}}_i$ denote the vector satisfying $||\widehat{\boldsymbol{v}}_i - \widehat{\boldsymbol{U}}^\top \bar{\boldsymbol{w}}_i^*|| \le \gamma_2 \lesssim \frac{1}{\kappa^2 \sqrt{K}}$. Let $\boldsymbol{Q}_1$ be defined in Lemma 12 and $\widehat{\boldsymbol{Q}}_1$ be its empirical version. If $||\boldsymbol{Q}_1 - \widehat{\boldsymbol{Q}}_1|| \le \gamma_3 ||\boldsymbol{Q}_1|| \lesssim \frac{1}{4}||\boldsymbol{Q}_1||$, then we have*

$$\left| ||\boldsymbol{w}_i^*|| - \widehat{\alpha}_i \right| \le (\kappa^4 K^{\frac{3}{2}}(\gamma_1 + \gamma_2) + \kappa^2 K^{\frac{1}{2}}\gamma_3)||\boldsymbol{w}_i^*|| \tag{253}$$

### G.2 Proof of Lemma 3

By the triangle inequality, we have

$$
\begin{aligned}
||\boldsymbol{w}_j^* - \widehat{\alpha}_j \widehat{\boldsymbol{U}} \widehat{\boldsymbol{v}}_j|| = \left|\left|\boldsymbol{w}_j^* - ||\boldsymbol{w}_j^*|| \widehat{\boldsymbol{U}} \widehat{\boldsymbol{v}}_j + ||\boldsymbol{w}_j^*|| \widehat{\boldsymbol{U}} \widehat{\boldsymbol{v}}_j - \widehat{\alpha}_j \widehat{\boldsymbol{U}} \widehat{\boldsymbol{v}}_j\right|\right| \\
\leq \left|\left|\boldsymbol{w}_j^* - ||\boldsymbol{w}_j^*|| \widehat{\boldsymbol{U}} \widehat{\boldsymbol{v}}_j\right|\right| + \left|\left|||\boldsymbol{w}_j^*|| \widehat{\boldsymbol{U}} \widehat{\boldsymbol{v}}_j - \widehat{\alpha}_j \widehat{\boldsymbol{U}} \widehat{\boldsymbol{v}}_j\right|\right| \\
\leq ||\boldsymbol{w}_j^*|| \left|\left|\bar{\boldsymbol{w}}_j^* - \widehat{\boldsymbol{U}} \widehat{\boldsymbol{v}}_j\right|\right| + \left|\left|||\boldsymbol{w}_j^*|| - \widehat{\alpha}_j\right|\right| ||\widehat{\boldsymbol{U}} \widehat{\boldsymbol{v}}_j|| \\
\leq ||\boldsymbol{w}_j^*|| \left|\left|\bar{\boldsymbol{w}}_j^* - \widehat{\boldsymbol{U}} \widehat{\boldsymbol{U}}^\top \bar{\boldsymbol{w}}_j^* + \widehat{\boldsymbol{U}} \widehat{\boldsymbol{U}}^\top \bar{\boldsymbol{w}}_j^* - \widehat{\boldsymbol{U}} \widehat{\boldsymbol{v}}_j\right|\right| + \left|\left|||\boldsymbol{w}_j^*|| - \widehat{\alpha}_j\right|\right| ||\widehat{\boldsymbol{U}} \widehat{\boldsymbol{v}}_j|| \\
\leq \delta_1(\boldsymbol{W}^*) \left(\left|\left|\bar{\boldsymbol{w}}_j^* - \widehat{\boldsymbol{U}} \widehat{\boldsymbol{U}}^\top \bar{\boldsymbol{w}}_j^*\right|\right| + \left|\left|\widehat{\boldsymbol{U}}^\top \bar{\boldsymbol{w}}_j^* - \widehat{\boldsymbol{v}}_j\right|\right|\right) + \left|\left|||\boldsymbol{w}_j^*|| - \widehat{\alpha}_j\right|\right|
\end{aligned}
\tag{254}
$$

From Lemma 10, Lemma 13, $\delta_K(\boldsymbol{Q}_2) \lesssim \delta_K^2(\boldsymbol{W}^*)$ and $\delta_K(\boldsymbol{Q}_3(\boldsymbol{I}_d, \boldsymbol{I}_d, \boldsymbol{\alpha})) \lesssim \delta_K^2(\boldsymbol{W}^*)$ for any arbitrary vector $\boldsymbol{\alpha} \in \mathbb{R}^d$, we have

$$
\begin{aligned}
\left|\left|\bar{\boldsymbol{w}}_j^* - \widehat{\boldsymbol{U}} \widehat{\boldsymbol{U}}^\top \bar{\boldsymbol{w}}_j^*\right|\right| \lesssim \frac{||\boldsymbol{Q}_2 - \widehat{\boldsymbol{Q}}_2||}{\delta_K(\boldsymbol{Q}_2)} \lesssim \sqrt{\frac{d \log n}{n}} \cdot \frac{\delta_1(\boldsymbol{W}^*)^2}{\delta_K(\boldsymbol{W}^*)^2} \cdot \tau^6 \sqrt{D_2(\Psi) D_4(\Psi)} \\
= \sqrt{\frac{d \log n}{n}} \cdot \kappa^2 \cdot \tau^6 \sqrt{D_2(\Psi) D_4(\Psi)}
\end{aligned}
\tag{255}
$$

if the mixed Gaussian distribution is not symmetric, and

$$
\left|\left|\bar{\boldsymbol{w}}_j^* - \widehat{\boldsymbol{U}} \widehat{\boldsymbol{U}}^\top \bar{\boldsymbol{w}}_j^*\right|\right| \lesssim \frac{||\boldsymbol{Q}_3(\boldsymbol{I}_d, \boldsymbol{I}_d, \boldsymbol{\alpha}) - \widehat{\boldsymbol{Q}}_3(\boldsymbol{I}_d, \boldsymbol{I}_d, \boldsymbol{\alpha})||}{\delta_K(\boldsymbol{Q}_3(\boldsymbol{I}_d, \boldsymbol{I}_d, \boldsymbol{\alpha}))} = \sqrt{\frac{d \log n}{n}} \cdot \kappa^2 \cdot \tau^6 \sqrt{D_2(\Psi) D_4(\Psi)}
\tag{256}
$$

if the mixed Gaussian distribution is symmetric. Moreover, we have

$$
\left|\left|\widehat{\boldsymbol{U}}^\top \bar{\boldsymbol{w}}_j^* - \widehat{\boldsymbol{v}}_j\right|\right| \leq \frac{K^{\frac{3}{2}}}{\delta_K^2(\boldsymbol{W}^*)} ||\boldsymbol{R}_3 - \widehat{\boldsymbol{R}}_3|| \lesssim \kappa^2 \cdot \left(\tau^6 \sqrt{D_6(\Psi)}\right) \cdot \sqrt{\frac{K^3 \log n}{n}}
\tag{257}
$$

in which the first step is by Theorem 3 in Kuleshov et al. (2015), and the second step is by Lemma 11. By Lemma 14, we have

$$
\left|\left|||\boldsymbol{w}_j^*|| - \widehat{\alpha}_j\right|\right| \leq (\kappa^4 K^{\frac{3}{2}} (\gamma_1 + \gamma_2) + \kappa^2 K^{\frac{1}{2}} \gamma_3) ||\boldsymbol{W}^*||
\tag{258}
$$

Therefore, taking the union bound of failure probabilities in Lemmas 10, 11, and 12 and by $D_2(\Psi) D_4(\Psi) \leq D_6(\Psi)$ from Property 10, we have that if the sample size $n \geq \kappa^8 K^4 \tau^{12} D_6(\Psi) \cdot d \log^2 d$, then the output $\boldsymbol{W}_0 \in \mathbb{R}^{d \times K}$ satisfies

$$
||\boldsymbol{W}_0 - \boldsymbol{W}^*|| \lesssim \kappa^6 K^3 \cdot \tau^6 \sqrt{D_6(\Psi)} \sqrt{\frac{d \log n}{n}} ||\boldsymbol{W}^*||
\tag{259}
$$

with probability at least $1 - n^{-\Omega(\delta_1^4(\boldsymbol{W}^*))}$.

### G.3 Proof of Lemma 10

From Assumption 1, if the Gaussian Mixture Model is a symmetric probability distribution defined in (16), then by Definition 1, we have

$$||\widehat{\boldsymbol{Q}}_3(\boldsymbol{I}, \boldsymbol{I}, \boldsymbol{\alpha}) - \boldsymbol{Q}_3(\boldsymbol{I}, \boldsymbol{I}, \boldsymbol{\alpha})||$$

$$
\begin{aligned}
=\Big|\Big| &\frac{1}{n}\sum_{i=1}^{n}\Big[y_i \cdot p(\boldsymbol{x})^{-1}\sum_{l=1}^{L}\lambda_l(2\pi|\boldsymbol{\Sigma}_l|)^{-\frac{d}{2}}\exp(-\frac{1}{2}(\boldsymbol{x}-\boldsymbol{\mu}_l)\boldsymbol{\Sigma}_l^{-1}(\boldsymbol{x}-\boldsymbol{\mu}_l)) \\
&\cdot\Big(((\boldsymbol{x}-\boldsymbol{\mu}_l)\boldsymbol{\Sigma}_l^{-1})^{\otimes 3}-((\boldsymbol{x}-\boldsymbol{\mu}_l)\boldsymbol{\Sigma}_l^{-1})\widetilde{\otimes}\boldsymbol{\Sigma}_l^{-1})\Big)\Big](\boldsymbol{I},\boldsymbol{I},\boldsymbol{\alpha}) \\
-\mathbb{E}&\Big[y \cdot p(\boldsymbol{x})^{-1}\sum_{l=1}^{L}\lambda_l(2\pi|\boldsymbol{\Sigma}_l|)^{-\frac{d}{2}}\exp(-\frac{1}{2}(\boldsymbol{x}-\boldsymbol{\mu}_l)\boldsymbol{\Sigma}_l^{-1}(\boldsymbol{x}-\boldsymbol{\mu}_l)) \\
&\cdot\Big(((\boldsymbol{x}-\boldsymbol{\mu}_l)\boldsymbol{\Sigma}_l^{-1})^{\otimes 3}-((\boldsymbol{x}-\boldsymbol{\mu}_l)\boldsymbol{\Sigma}_l^{-1})\widetilde{\otimes}\boldsymbol{\Sigma}_l^{-1})\Big)\Big](\boldsymbol{I},\boldsymbol{I},\boldsymbol{\alpha})\Big|\Big|
\end{aligned}
\tag{260}
$$

Following Zhong et al. (2017b), $\widetilde{\otimes}$ is defined such that for any $\boldsymbol{v} \in \mathbb{R}^{d_1}$ and $\boldsymbol{Z} \in \mathbb{R}^{d_1 \times d_2}$,

$$\boldsymbol{v}\widetilde{\otimes}\boldsymbol{Z} = \sum_{i=1}^{d_2}(\boldsymbol{v}\otimes\boldsymbol{z}_i\otimes\boldsymbol{z}_i + \boldsymbol{z}_i\otimes\boldsymbol{v}\otimes\boldsymbol{z}_i + \boldsymbol{z}_i\otimes\boldsymbol{z}_i\otimes\boldsymbol{v}), \tag{261}$$

where $\boldsymbol{z}_i$ is the $i$-th column of $\boldsymbol{Z}$. By Definition 1, we have

$$
\begin{aligned}
\Big|\Big|&\Big[y \cdot p(\boldsymbol{x})^{-1}\sum_{l=1}^{L}\lambda_l(2\pi|\boldsymbol{\Sigma}_l|)^{-\frac{d}{2}}\exp(-\frac{1}{2}(\boldsymbol{x}-\boldsymbol{\mu}_l)\boldsymbol{\Sigma}_l^{-1}(\boldsymbol{x}-\boldsymbol{\mu}_l)) \\
&\cdot\Big(((\boldsymbol{x}-\boldsymbol{\mu}_l)\boldsymbol{\Sigma}_l^{-1})^{\otimes 3}-((\boldsymbol{x}-\boldsymbol{\mu}_l)\boldsymbol{\Sigma}_l^{-1})\widetilde{\otimes}\boldsymbol{\Sigma}_l^{-1})\Big)\Big](\boldsymbol{I},\boldsymbol{I},\boldsymbol{\alpha})\Big|\Big| \\
\lesssim &\Big|\Big|\frac{\sum_{l=1}^{L}\lambda_l(2\pi|\boldsymbol{\Sigma}_l|)^{-\frac{d}{2}}\exp(-\frac{1}{2}(\boldsymbol{x}-\boldsymbol{\mu}_l)\boldsymbol{\Sigma}_l^{-1}(\boldsymbol{x}-\boldsymbol{\mu}_l))\cdot((\boldsymbol{x}-\boldsymbol{\mu}_l)\boldsymbol{\Sigma}_l^{-1})^{\otimes 2}(\boldsymbol{\alpha}^\top\boldsymbol{\Sigma}_l^{-1}(\boldsymbol{x}-\boldsymbol{\mu}_l))}{\sum_{l=1}^{L}\lambda_l(2\pi|\boldsymbol{\Sigma}_l|)^{-\frac{d}{2}}\exp(-\frac{1}{2}(\boldsymbol{x}-\boldsymbol{\mu}_l)\boldsymbol{\Sigma}_l^{-1}(\boldsymbol{x}-\boldsymbol{\mu}_l))}\Big|\Big| \\
\lesssim &||\sigma_{\min}^{-6}(\boldsymbol{x}^\top\boldsymbol{\alpha})\boldsymbol{x}\boldsymbol{x}^\top||
\end{aligned}
\tag{262}
$$

The first step of (262) is because $(\boldsymbol{x}-\boldsymbol{\mu}_l)\boldsymbol{\Sigma}_l)^{\otimes 2}(\boldsymbol{\alpha}^\top\boldsymbol{\Sigma}_l^{-1}(\boldsymbol{x}-\boldsymbol{\mu}_l))$ is the dominant term of the entire expression, and $y \leq 1$. The second step is because the expression can be considered as a normalized weighted summation of $((\boldsymbol{x}-\boldsymbol{\mu}_l)\boldsymbol{\Sigma}_l)^{\otimes 2}(\boldsymbol{\alpha}^\top\boldsymbol{\Sigma}_l^{-1}(\boldsymbol{x}-\boldsymbol{\mu}_l))$ and $(\boldsymbol{x}^\top\boldsymbol{\alpha})\boldsymbol{x}\boldsymbol{x}^\top$ is its dominant term. Define $S_m(\boldsymbol{x}) = (-1)^m\frac{\nabla_{\boldsymbol{x}}^m p(\boldsymbol{x})}{p(\boldsymbol{x})}$, where $p(\boldsymbol{x})$ is the probability density function of the random variable $\boldsymbol{x}$. From Definition 1, we can verify that

$$\boldsymbol{Q}_j = \mathbb{E}[y \cdot S_m(\boldsymbol{x})] \quad j \in \{1, 2, 3\} \tag{263}$$

Then define $Gp_i = \langle\boldsymbol{v}, ([y_i \cdot S_3(\boldsymbol{x}_i)](\boldsymbol{I}_d, \boldsymbol{I}_d, \boldsymbol{\alpha}) - \mathbb{E}[[y_i \cdot S_3(\boldsymbol{x}_i)](\boldsymbol{I}_d, \boldsymbol{I}_d, \boldsymbol{\alpha})]\boldsymbol{v})\rangle$, where $||\boldsymbol{v}|| = 1$, then $\mathbb{E}[Gp_i] = 0$. Similar to the proof of (202), (203), and (204) in Lemma 8, we have

$$|Gp_i|^p \lesssim |\sigma_{\min}^{-6}(\boldsymbol{x}_i^\top\boldsymbol{\alpha})(\boldsymbol{x}_i^\top\boldsymbol{v})^2 + \mathbb{E}_{\boldsymbol{x}\sim\sum_{l=1}^{L}\mathcal{N}(\boldsymbol{\mu}_l,\boldsymbol{\Sigma}_l)}[\sigma_{\min}^{-6}(\boldsymbol{x}_i^\top\boldsymbol{\alpha})(\boldsymbol{x}_i^\top\boldsymbol{v})^2]|^p \tag{264}$$

$$
\begin{aligned}
\mathbb{E}[|Gp_i|^p] &\lesssim (\mathbb{E}_{\boldsymbol{x}\sim\sum_{l=1}^{L}\mathcal{N}(\boldsymbol{\mu}_l,\boldsymbol{\Sigma}_l)}[\sigma_{\min}^{-6}(\boldsymbol{x}_i^\top\boldsymbol{\alpha})(\boldsymbol{x}_i^\top\boldsymbol{v})^2])^p \\
&\leq \sigma_{\min}^{-6p}\mathbb{E}_{\boldsymbol{x}\sim\sum_{l=1}^{L}\mathcal{N}(\boldsymbol{\mu}_l,\boldsymbol{\Sigma}_l)}[(\boldsymbol{x}^\top\boldsymbol{\alpha})^2]^{\frac{p}{2}}\mathbb{E}_{\boldsymbol{x}\sim\sum_{l=1}^{L}\mathcal{N}(\boldsymbol{\mu}_l,\boldsymbol{\Sigma}_l)}[(\boldsymbol{x}^\top\boldsymbol{v})^4]^{\frac{p}{2}} \\
&\leq \tau^{6p}\sqrt{D_2(\Psi)D_4(\Psi)}^p
\end{aligned}
\tag{265}
$$

$$
\begin{aligned}
\mathbb{E}[\exp(\theta Gp_i)] &\lesssim 1 + \sum_{p=2}^{\infty}\frac{\theta^p\mathbb{E}[|Gp_i|^p]}{p!} \lesssim 1 + \sum_{p=2}^{\infty}\frac{|e\theta|^p\tau^{6p}(D_2(\Psi)D_4(\Psi))^{\frac{p}{2}}}{p^p} \\
&\lesssim 1 + \theta^2\tau^{12}D_2(\Psi)D_4(\Psi)
\end{aligned}
\tag{266}
$$

Hence, similar to the derivation of (206), we have

$$\mathbb{P}\Big(\frac{1}{n}\sum_{i=1}^{n}Gp_i \geq t\Big) \leq \exp\Big(-n\theta t + C_{16}n\theta^2(\tau^6\sqrt{D_2(\Psi)D_4(\Psi)})^2\Big) \tag{267}$$

for some constant $C_{16} > 0$. Let $\theta = \frac{t}{2C_{16}\left(\tau^6\sqrt{D_2(\Psi)D_4(\Psi)}\right)^2}$ and $t = \delta_1^2(\boldsymbol{W}^*) \cdot \left(\tau^6\sqrt{D_2(\Psi)D_4(\Psi)}\right) \cdot \sqrt{\frac{d\log n}{n}}$, then we have

$$||\widehat{\boldsymbol{Q}}_3(\boldsymbol{I}_d, \boldsymbol{I}_d, \boldsymbol{\alpha}) - \boldsymbol{Q}_3(\boldsymbol{I}_d, \boldsymbol{I}_d, \boldsymbol{\alpha})|| \le \delta_1(\boldsymbol{W}^*)^2 \cdot \left(\tau^6\sqrt{D_2(\Psi)D_4(\Psi)}\right) \cdot \sqrt{\frac{d\log n}{n}} \qquad (268)$$

with probability at least $1 - 2n^{-\Omega(\delta_1^4(\boldsymbol{W}^*)d)}$.

If the Gaussian Mixture Model is not a symmetric distribution which is defined in (16), we would have a similar result as follows.

$$||\widehat{\boldsymbol{Q}}_2 - \boldsymbol{Q}_2|| = \left|\left|\frac{1}{n}\sum_{i=1}^n [y_i \cdot S_2(\boldsymbol{x})] - \mathbb{E}[y \cdot S_2(\boldsymbol{x})]\right|\right| \qquad (269)$$

$$||y_i \cdot S_2(\boldsymbol{x}_i)|| \lesssim ||\sigma_{\min}^{-4}\frac{1}{K}\sum_{j=1}^K \phi(\boldsymbol{w}_j^{*\top}\boldsymbol{x}_i)\boldsymbol{x}_i\boldsymbol{x}_i^\top|| \qquad (270)$$

Then define $Gp_i' = \left\langle \boldsymbol{v}, ([y_i \cdot S_2(\boldsymbol{x}_i)] - \mathbb{E}[y_i \cdot S_2(\boldsymbol{x}_i)]\boldsymbol{v})\right\rangle$, where $||\boldsymbol{v}|| = 1$, then $\mathbb{E}[Gp_i'] = 0$. Similar to the proof of (202), (203) and (204) in Lemma 8, we have

$$|Gp_i'|^p \lesssim \left|\sigma_{\min}^{-4}(\boldsymbol{x}_i^\top\boldsymbol{v})^2 + \mathbb{E}_{\boldsymbol{x}\sim\sum_{l=1}^L \mathcal{N}(\boldsymbol{\mu}_l, \boldsymbol{\Sigma}_l)}[\sigma_{\min}^{-4}(\boldsymbol{x}_i^\top\boldsymbol{v})^2]\right|^p \qquad (271)$$

$$\mathbb{E}[|Gp_i'|^p] \lesssim \left(\mathbb{E}_{\boldsymbol{x}\sim\sum_{l=1}^L \mathcal{N}(\boldsymbol{\mu}_l, \boldsymbol{\Sigma}_l)}[\sigma_{\min}^{-4}(\boldsymbol{x}_i^\top\boldsymbol{v})^2]\right)^p \le \tau^{4p}D_2(\Psi)^p \qquad (272)$$

$$\mathbb{E}[\exp(\theta Gp_i')] \lesssim 1 + \sum_{p=2}^\infty \frac{\theta^p\mathbb{E}[|Gp_i|^p]}{p!} \lesssim 1 + \sum_{p=2}^\infty \frac{|e\theta|^p\tau^{4p}D_2(\Psi)^p}{p^p} \qquad (273)$$

$$\lesssim 1 + \theta^2\tau^8 D_2(\Psi)^2$$

Hence, similar to the derivation of (206), we have

$$\mathbb{P}\left(\frac{1}{n}\sum_{i=1}^n Gp_i \ge t\right) \le \exp\left(-n\theta t + C_{17}n\theta^2\left(\tau^4 D_2(\Psi)\right)^2\right) \qquad (274)$$

for some constant $C_{17} > 0$. Let $\theta = \frac{t}{2C_{17}\left(\tau^4 D_2(\Psi)\right)^2}$ and $t = \delta_1^2(\boldsymbol{W}^*) \cdot \left(\tau^4 D_2(\Psi)\right) \cdot \sqrt{\frac{d\log n}{n}}$, then we have

$$||\widehat{\boldsymbol{Q}}_2 - \boldsymbol{Q}_2|| \lesssim \delta_1^2(\boldsymbol{W}^*) \cdot \tau^4 D_2(\Psi) \cdot \sqrt{\frac{d\log n}{n}}$$
$$\lesssim \sqrt{\frac{d\log n}{n}} \cdot \delta_1^2(\boldsymbol{W}^*) \cdot \tau^6\sqrt{D_2(\Psi)D_4(\Psi)} \qquad (275)$$

with probability at least $1 - 2n^{-\Omega(\delta_1^4(\boldsymbol{W}^*)d)}$.

### G.4 PROOF OF LEMMA 11

We consider each component of $y = \frac{1}{K}\sum_{i=1}^K \phi(\boldsymbol{w}_i^{*\top}\boldsymbol{x})$.
Define $\boldsymbol{T}_i(\boldsymbol{x}) : \mathbb{R}^d \to \mathbb{R}^{K\times K\times K}$ such that

$$\boldsymbol{T}_i(\boldsymbol{x}) = [\phi(\boldsymbol{w}_i^{*\top}\boldsymbol{x}) \cdot S_3(\boldsymbol{x})](\widehat{\boldsymbol{U}}, \widehat{\boldsymbol{U}}, \widehat{\boldsymbol{U}}) \qquad (276)$$

We flatten $\boldsymbol{T}_i(\boldsymbol{x}) : \mathbb{R}^d \to \mathbb{R}^{K\times K\times K}$ along the first dimension to obtain the function $\boldsymbol{B}_i(\boldsymbol{x}) : \mathbb{R}^d \to \mathbb{R}^{K\times K^2}$. Similar to the derivation of the last step of Lemma E.8 in Zhong et al. (2017b), we can obtain $||\boldsymbol{T}_i(\boldsymbol{x})|| \le ||\boldsymbol{B}_i(\boldsymbol{x})||$. By (260), we have

$$||\boldsymbol{B}_i(\boldsymbol{x})|| \lesssim \sigma_{\min}^{-6}\frac{1}{K}\sum_{j=1}^K \phi(\boldsymbol{w}_j^{*\top}\boldsymbol{x}_i)(\widehat{\boldsymbol{U}}^\top\boldsymbol{x})^3 \qquad (277)$$

Define $Gr_i = \langle \boldsymbol{v}, \boldsymbol{B}_i(\boldsymbol{x}_i)) - \mathbb{E}[\boldsymbol{B}_i(\boldsymbol{x}_i)]\boldsymbol{v}) \rangle$, where $||\boldsymbol{v}|| = 1$, so $\mathbb{E}[Gr_i] = 0$. Similar to the proof of (202), (203) and (204) in Lemma 8, we have

$$|Gr_i|^p \lesssim \left|\sigma_{\min}^{-6}(\boldsymbol{v}^\top \widehat{\boldsymbol{U}}^\top \boldsymbol{x})^3 + \mathbb{E}_{\boldsymbol{x} \sim \sum_{l=1}^L \mathcal{N}(\boldsymbol{\mu}_l, \boldsymbol{\Sigma}_l)}[\sigma_{\min}^{-6}(\boldsymbol{v}^\top \widehat{\boldsymbol{U}}^\top \boldsymbol{x})^3]\right|^p \tag{278}$$

$$\mathbb{E}[|Gr_i|^p] \lesssim \left(\mathbb{E}_{\boldsymbol{x} \sim \sum_{l=1}^L \mathcal{N}(\boldsymbol{\mu}_l, \boldsymbol{\Sigma}_l)}[\sigma_{\min}^{-6}(\boldsymbol{v}^\top \widehat{\boldsymbol{U}}^\top \boldsymbol{x})^3]\right)^p \lesssim \tau^{6p}\sqrt{D_6(\Psi)}^p \tag{279}$$

$$\mathbb{E}[\exp(\theta Gr_i)] \lesssim 1 + \sum_{p=2}^\infty \frac{\theta^p \mathbb{E}[|Gr_i|^p]}{p!} \lesssim 1 + \sum_{p=2}^\infty \frac{|e\theta|^p \tau^{6p} D_6(\Psi)^{\frac{p}{2}}}{p^p} \tag{280}$$
$$\leq 1 + \theta^2(\tau^{12}\sqrt{D_6(\Psi)})^2$$

Hence, similar to the derivation of (206), we have

$$\mathbb{P}\Big(\frac{1}{n}\sum_{i=1}^n Gr_i \geq t\Big) \leq \exp\Big(-n\theta t + C_{18}\theta^2\big(\tau^6\sqrt{D_6(\Psi)}\big)^2\Big) \tag{281}$$

for some constant $C_{18} > 0$. Let $\theta = \frac{t}{C_{18}\big(\tau^6\sqrt{D_6(\Psi)}\big)^2}$ and $t = \delta_1^2(\boldsymbol{W}^*) \cdot \big(\tau^6\sqrt{D_6(\Psi)}\big) \cdot \sqrt{\frac{\log n}{n}}$, then we have

$$||\widehat{\boldsymbol{R}}_3 - \boldsymbol{R}_3|| \lesssim \delta_1(\boldsymbol{W}^*)^2 \cdot \big(\tau^6\sqrt{D_6(\Psi)}\big) \cdot \sqrt{\frac{\log n}{n}} \tag{282}$$

with probability at least $1 - 2n^{-\Omega(\delta_1^4(\boldsymbol{W}^*))}$.

## G.5 PROOF OF LEMMA 12

From Definition 1, we have

$$||\widehat{\boldsymbol{Q}}_1 - \boldsymbol{Q}_1|| = \left\|\frac{1}{n}\sum_{i=1}^n [y_i \cdot S_1(\boldsymbol{x})] - \mathbb{E}[y \cdot S_1(\boldsymbol{x})]\right\|. \tag{283}$$

Based on Definition 1,

$$\left\|[y_i \cdot S_1(\boldsymbol{x}_i)]\right\| \lesssim \left\|\frac{\sum_{l=1}^L \lambda_l \lambda_l (2\pi \prod_{k=1}^d \sigma_{lk}^2)^{-\frac{d}{2}} \exp(-\frac{1}{2}(\boldsymbol{x} - \boldsymbol{\mu}_l)\boldsymbol{\Sigma}_l^{-1}(\boldsymbol{x} - \boldsymbol{\mu}_l)) \cdot (\boldsymbol{x} - \boldsymbol{\mu}_l)\boldsymbol{\Sigma}_l^{-1}}{\sum_{l=1}^L \lambda_l \lambda_l (2\pi \prod_{k=1}^d \sigma_{lk}^2)^{-\frac{d}{2}} \exp(-\frac{1}{2}(\boldsymbol{x} - \boldsymbol{\mu}_l)\boldsymbol{\Sigma}_l^{-1}(\boldsymbol{x} - \boldsymbol{\mu}_l))}\right\|$$
$$\lesssim \left\|\sigma_{\min}^{-2}\frac{1}{K}\sum_{j=1}^K \phi(\boldsymbol{w}_j^{*\top}\boldsymbol{x}_i)\boldsymbol{x}_i\right\| \tag{284}$$

Define $Gq_i = \langle \boldsymbol{v}, ([y_i \cdot S_1(\boldsymbol{x}_i)] - \mathbb{E}[[y_i \cdot S_1(\boldsymbol{x}_i)]]\boldsymbol{v}) \rangle$, where $||\boldsymbol{v}|| = 1$, so $\mathbb{E}[Gq_i] = 0$. Similar to the proof of (202), (203), and (204) in Lemma 8, we have

$$|Gq_i|^p \lesssim \left|\sigma_{\min}^{-2}(\boldsymbol{x}_i^\top \boldsymbol{v}) + \mathbb{E}_{\boldsymbol{x} \sim \sum_{l=1}^L \mathcal{N}(\boldsymbol{\mu}_l, \boldsymbol{\Sigma}_l)}[\sigma_{\min}^{-2}(\boldsymbol{x}_i^\top \boldsymbol{v})]\right|^p \tag{285}$$

$$\mathbb{E}[|Gq_i|^p] \lesssim \left(\mathbb{E}_{\boldsymbol{x} \sim \sum_{l=1}^L \mathcal{N}(\boldsymbol{\mu}_l, \boldsymbol{\Sigma}_l)}[\sigma_{\min}^{-2}(\boldsymbol{x}_i^\top \boldsymbol{v})]\right)^p \leq \tau^{2p}\sqrt{D_2(\Psi)}^p \tag{286}$$

$$\mathbb{E}[\exp(\theta Gq_i)] \lesssim 1 + \sum_{p=2}^\infty \frac{\theta^p \mathbb{E}[|Gq_i|^p]}{p!} \lesssim 1 + \sum_{p=2}^\infty \frac{|e\theta|^p \tau^{2p} D_2(\Psi)^{\frac{p}{2}}}{p^p} \tag{287}$$
$$\leq 1 + \theta^2(\tau^2\sqrt{D_2(\Psi)})^2$$

Hence, similar to the derivation of (206), we have

$$\mathbb{P}\Big(\frac{1}{n}\sum_{i=1}^n Gq_i \geq t\Big) \leq \exp\Big(-n\theta t + C_{19}\theta^2\big(\tau^2\sqrt{D_2(\Psi)}\big)^2\Big) \tag{288}$$

for some constant $C_{19} > 0$. Let $\theta = \frac{t}{C_{19}\left(\tau^2\sqrt{D_2(\Psi)}\right)^2}$ and $t = \left(\tau^2\sqrt{D_2(\Psi)}\right) \cdot \sqrt{\frac{d\log n}{n}}$, then we have

$$\|\widehat{\boldsymbol{Q}}_1 - \boldsymbol{Q}_1\| \lesssim \left(\tau^2\sqrt{D_2(\Psi)}\right) \cdot \sqrt{\frac{d\log n}{n}} \tag{289}$$

with probability at least $1 - 2n^{-\Omega(d)}$.

