# OpenReview forum: "Theoretical  Characterization of Neural Network Generalization with Group Imbalance"
_ICLR.cc/2023/Conference — Submitted to ICLR 2023_

### Official Review · Reviewer_XNZ2 · 2022-10-13

**Confidence:** 1
**Clarity, Quality, Novelty And Reproducibility:** I informed the area chair that I cann…
**Correctness:** 4
**Technical Novelty And Significance:** 4
**Empirical Novelty And Significance:** Not applicable
**Recommendation:** 10

**Strength And Weaknesses:**

I informed the area chair that I cannot review this paper.

**Summary Of The Paper:**

I informed the area chair that I cannot review this paper.

**Summary Of The Review:**

I informed the area chair that I cannot review this paper.

---

### Official Review · Reviewer_n1W9 · 2022-10-20

**Confidence:** 4
**Correctness:** 2
**Technical Novelty And Significance:** 3
**Empirical Novelty And Significance:** 3
**Recommendation:** 5

**Clarity, Quality, Novelty And Reproducibility:**

The writing is clear and easy to follow. There is some technical novelty in the proof of the main result.

**Strength And Weaknesses:**

Strength: The proof of the main result (Theorem 1 and Corollary 1) has some technical novelty, and the result leads to some interesting insights. For example, I think (P4) is a very interesting catch. The authors also validate the result with experiments.

Weaknesses: The main concern I have is that the main result (Theorem 1) only works for a very limited setting, and the insights might not be generalizable to a more general setting. While it is understandable that such a theoretical result requires some strong assumptions, I feel that the assumptions made in Theorem 1 are way too strong, and some insights drawn from the result (especially (P5) and (P6)) are questionable.

Specifically, Theorem 1 assumes that:
- The inputs are sampled from a specific Gaussian mixture model
- The model is a specific 1-hidden-layer neural network
- The labels are generated by a ground truth model of the exact same structure as the model used for training
- $\sigma_{min}$ is bounded away from zero, which is actually a very strong assumption: It is well known that in most tasks, the data lies on a low-dimensional manifold in a high-dimensional space, in which case $\Sigma_l$ could even be non-invertible. And even if it is invertible, $\sigma_{min}$ should be very close to zero.

Furthermore, the main result (Theorem 1) only considers ERM, i.e. minimizing the average risk. It does not discuss any robust training method. One thing I believe the authors should add to the paper is ERM with importance weighting, i.e. minimizing the weighted average of the empirical risk. This should not be too different from ERM. Even better, the authors could also consider DRO, group DRO, etc.

Regarding the insights, (P5) suggests that the performance would be the best if all Gaussian components have zero means, in which case the groups largely overlap with one another (provided that $\tau = \Theta(1)$). This seems rather vacuous. If all groups have the same mean and the same co-variance then of course the performance would be the best given that the groups share the same labeling function. However, in real tasks such as a fairness task, the groups are very different. Plus, I think the real reason why the groups need to have zero means is that the model does not have a bias term: It’s $\phi(w^{\top} x)$ rather than $\phi (w^{\top} x+b)$.


Moreover, I cannot see how (P3) and (P5) lead to (P6). In this paper’s setting, batch normalization mixes the samples from different Gaussian components and potentially messes up the data structure. (P6) is only valid when all the samples in the batch come from the same component, or when the components largely overlap with each other as in (P5).


**Summary Of The Paper:**

This paper proves a group imbalanced generalization result for a one-hidden-layer neural network under the Gaussian mixture model assuming that the labels are generated by a ground truth model of the same structure. With this result, the paper argues that the learning performance is the best when the spectral norm of the co-variance of each Gaussian component is at a medium regime (neither too big nor too small), and increasing the fraction of the minority group data does not improve the test performance if the spectral norm of its co-variance is large. The authors then conduct synthetic and realistic experiments to verify this result.

**Summary Of The Review:**

Overall, I think this paper provides some interesting insights (I especially like (P4)). However, the main result does make a lot of strong assumptions, and some insights drawn from the main result ((P5) and (P6)) are questionable, so I am concerned that these insights might not be generalizable to a more realistic setting. Moreover, it seems to me that all results and discussions in this paper are based on Theorem 1. While the proof of Theorem 1 is indeed not easy, I am concerned that the contributions might be insufficient for ICLR acceptance, especially given that Theorem 1 requires too many strong assumptions. I think the authors should also include robust training methods, such as importance weighting, in their result. I believe that this paper can benefit from another round of edition, so I recommend weak rejection.

---

### Official Review · Reviewer_LoT6 · 2022-10-25

**Confidence:** 2
**Correctness:** 4
**Technical Novelty And Significance:** 3
**Empirical Novelty And Significance:** 3
**Recommendation:** 8

**Clarity, Quality, Novelty And Reproducibility:**

Group imbalance is a crucial issue in deep learning. Investigation of the effect of group imbalance on the generalization ability is interesting and is in high demand. The theoretical insights, particularly the dependency of the covariance norm on the generalization ability, are very interesting. More peculiarly, the insight of (iv) and (v) in Corollary 1 helps to understand the effect of the group imbalance. The analysis techniques also look novel. As claimed by the authors, analyzing the local convex region under the GMM model has a significant difficulty.

Given the results in this paper, a natural question arises do the results generalize for a deeper model? I understand the theoretical analysis for a deeper model is hard. However, empirical evaluations for a deeper model might be carried out easily. It is very helpful if the empirical analyses on when the findings generalize to a deeper model are included.

**Strength And Weaknesses:**

Strength:
-  Well-written and easy to follow.
- The insights regarding the effects of the group-wise covariances help understand the effect of the group imbalance.
- The introduced analysis techniques look novel

Weakness:
- Lack of empirical evaluations with a deeper model.

**Summary Of The Paper:**

The authors theoretically investigate the generalization performance of the one-hidden-layer neural network under a mixture of Gaussian. The main focus of this paper is to analyze the effect of group imbalance on the generalization ability. To simulate the group imbalance, their model can vary the group-wise Gaussian parameters. As a result, they demonstrate that the medium magnitude of the covariance achieves the highest generalization ability. This result also implies that increasing the minority group data does not always improve the generalization ability. The empirical evaluations also demonstrate the claim obtained in the theoretical result.

**Summary Of The Review:**

This paper is well-written and easy to follow. Investigation of group imbalance is in high motivation as group imbalance is a crucial issue in deep learning. The provided insights are interesting and helpful for understanding the effect of the group imbalance. I hence recommend the acceptance.

---

### Official Review · Reviewer_MquM · 2022-10-26

**Confidence:** 3
**Correctness:** 3
**Technical Novelty And Significance:** 3
**Empirical Novelty And Significance:** 2
**Recommendation:** 5

**Clarity, Quality, Novelty And Reproducibility:**

* **Clarity**: The clarity of the paper could be improved. There are parts that are not obvious to me in the proofs and the main text. Also, some of the claims are under-justified or vague.

* **Quality**: The quality of the paper is varying. I believe that it is a good piece of work in most parts. However there are some potential mistakes in some parts of the text (either fundamental, as in claims of this explaining a zero-mean preprocessing or batch normalization; or in some mathematical points).

* **Novelty**: The paper seems novel. I have not seen a study under these conditions where the data is assumed to come from a mixture of Gaussians and the imbalance studied before.

* **Reproducibility**: \
  *Theory*: I reproduced most of the proofs and could follow most of them. There were some parts that were not clear. \
  Overall, I believe that the amount of proofs is too long to have a proper review in the time available for a conference revision.

  *Experiments*: There is no code to reproduce the experiments. Hence, I did not have time to replicate them. In this particular case, where a theoretical analysis is involved, having the code available is paramount, since the empirical match with the theory needs to be checked.

**Strength And Weaknesses:**

**Strengths**

* The paper tackles an important problem: convergence and generalization of GD with imbalanced data.

* The insights from the main theorem are interesting and potentially useful to develop fairer learning algorithms.

* Some new results regarding GMMs' concentration are given in the appendix.

**Weaknesses**

* The learning model assumption seems a little artificial and restrictive.

  * It is mentioned that a one hidden layer model is assumed, but then we are provided with the model in (1), which is standard logistic regression (that is, a linear layer followed by a sigmoid activation function). \
  It is mentioned that the weights in the second layer are assumed to be fixed to facilitate the analysis. This is an artificial setting by itself, which even if other works have employed it, needs some justification as is quite uncommon and its utility is unclear to me. But also the second layer weights are not appearing in (1).

  * The assumption that the real function from features to labels is also strong. Is it also assumed that they come from the arg-max after a linear layer?

* There are some parts of the text where the claims seem incorrect or potentially incorrect:

  * Why do you say that GD converges linearly in the introduction? In the Theorem 1 we see that it converges at a rate $\mathcal{O}(\sqrt{d \log n / n} / K^{1.5})$. That's not linear in any parameter.

  * Why does that group-level mean shifts from zero justify the pre-processing of making the data zero-mean? Note that this pre-processing will make the overall data have a mean of zero. Since it is assumed that the data comes from a mixture of distributions, it is not guaranteed it will have a beneficial effect on a minority group (indeed, it can be harmful if the mean of the group shifts away from zero after the average centering). \
  The same problem occurs with (P6). Unless the batch normalization is applied per group, which is not the case in this paper, it does not necessarily help as the means and variances are updated to normalize the whole batch considered as a unimodal distribution.

  * In Section 3, it is said that $x$ follows a GMM $\sum_{l=1}^L \lambda_l \mathcal{N}(\mu_l, \Sigma_l)$. This assumes that the fraction of the data from each group exactly matches the influence in the real mixture model. In practice, if one has a GMM $\sum_{l=1}^L \lambda_l \mathcal{N}(\mu_l, \Sigma_l)$ one will observe $\tilde{\lambda}_l n$ samples of each group with $\tilde{\lambda}_l \to \lambda_l$ as $n \to \infty$. How does this affect your results?

  * The assumption that $\tau \in \Theta(1)$ seems strong and potentially not true in most cases.  I did some checks with random matrices $A \in \mathbb{R}^d$ generating potential covariances matrices $\Sigma = A^T A$ and it seems that $\tau \in \Theta(d)$. This can be problematic since in the proofs the dependence with $\tau$ (and hence potentially $d$) can be as high as $\tau^{12}$. \
  You could check how this condition holds up taking natural images of different sizes and assuming they are Gaussian. Compute their covariances per group and see if $\tau$ grows or not with the dimension to see if this assumption would hold or not in practice.

  * In Corollary 1, how do conditions (i) and (ii) be true simultaneously? Consider $\Sigma_l^{(1)}$ and $\Sigma_l^{(2)}$. Each is worse than the other since one has a spectral norm closer to zero (condition (i)) and the other has a larger spectral norm (condition (ii)). The idea is further explained in (P3), but needs a more formal statement here.

  * In (P4) it is mentioned that $\lVert \Sigma_l \rVert$ is the smallest among all groups, increasing $\lambda_l$ improves the learning performance since the learning performance is enhanced at a medium regime of group-level co-variance. But if it is smallest, as in the case where it is largest, the medium-level covariance may change, thus also decreasing the performance. It would only increase performance when $\lVert \Sigma_l \rVert$ is close to the medium level.

  * In the appendix, you use that $\nu_i$ ar i.i.d. zero-mean Gaussians but also assume they have bounded magnitude. How is this possible? I believe you can still recover some of your results by saying that this happens with a certain probability using some tail bounds on Gaussian random variables, but as of now, I do not understand how can this happen.

  * When applying l'Hôpital's  rule in (33) if I am not mistaken you are using that both $f(\sigma) = 1 / (u_j^2 / \sigma^2 + 1) \to 0$ and $g(\sigma) = \beta_0(i, u/\sigma, \sigma) - \alpha_0(i, u/\sigma, \sigma) \to 0$ as $\sigma \to 0$. However, when applying the rule $\partial f(\sigma) / \partial \sigma = 2 u_j^2 \sigma / (u_j^2 + \sigma^2)^2$ and not $u_i^2 / (2 \sigma)$. Hence, (33) ends up being a situation of $0 \cdot \infty$ and it is not determined. \
  This affects other properties like Property 3 (4) which builds upon the fact that $\rho > 0$.

  * I don't follow the equality after the inequality in (52). Where did the $e^{-|\mu|^2 / 2}$ term go?

  * Many of the lemmata leading to the main results are based on approximate inequalities ($\gtrsim$). This is concerning since if the approximation error is not quantified it may have important effects on the final results.

* There are some parts of the text that are not clear to me:

  * In Figure 3 (c), it is not clear to me how the convergence rate is calculated.

  * In Figure 4 (b) the test loss of the minority group is lower than the average risk. How is this possible? That would mean that the largest group has a larger loss.

  * Why are the results in Figure 6 consistent with (P4)? In order to have this consistency it would be good to know which are $\lVert \Sigma_{\text{male}} \rVert$ and $\lVert \Sigma_{\text{female}} \rVert$, even if only empirically, and then observe how it is indeed the case that when the performance increases or decreases is for the reasons outlined in (P4).

  * Similarly, the results with increasing $\delta$ or $w$ are not clearly showing what they intended. We would need an estimation of $\lVert \Sigma_{\text{male}} \rVert$ and $\lVert \Sigma_{\text{female}} \rVert$ to see that effect. Similarly, one could also test doing a subtraction of the mean and increasing the mean of these means to see if the results are better when this happens and they get worse as they diverge.

* Sometimes some important details are left:

  * In (29) note that we know that (27) and (28) are positive due to Jensen's inequality.

  * In the proof of Property 3 (3) note that we can use the assumption that the Gaussian behaves like the delta due to the Dominated Convergence Theorem (DCT).

* The bound, though interesting, seems fairly vacuous.

  * Note the potential dependence on $d$ of $\tau$. In this case the number of samples required would explode with increasing dimensions. Even in the artificial case when $\Sigma_1 = \Sigma_2 = I$, for feature dimension 100, 3 units, and 2 classes, one needs more than 150k samples for the results to apply (Figure 1 a). Note that this is far less than the MNIST data samples (60k), where the feature dimension is 256.

**Summary Of The Paper:**

This paper studies the convergence, average population risk, and average per-group population risk of empirical risk minimization (ERM) with gradient descent (GD). They study these phenomena under the assumptions that:

* the input data features come from a Gaussian mixture model (GMM) with mixing coefficients equal to the fraction of the data collected for that group,
* the input data labels are obtained with a neural network of the same architecture and some unknown weights,
* the gradient descent is applied under a neural network with a single hidden layer, and
* the task is a binary classification and the loss is the binary cross-entropy.

Their main result is Theorem 1, which provides some insights into when the performance is best:

* having a group mean close to zero (in fact, low norm $\lVert \mu_l \rVert$) is beneficial for training,
* having the group co-variance close to the medium co-variance is beneficial for training, and
* increasing the fraction of the minority group does not always increase the performance (it can degrade performance when the co-variance is far from the medium co-variance).

Another contribution that is not explicitly in the main text is new results on GMMs' concentration.

**Summary Of The Review:**

This paper studies the convergence and generalization properties of ERM with GD on the expected population and expected per-group population risks for binary classification in logistic regression with the binary-cross entropy loss.

 They consider a rather artificial set-up where the input features data comes from a GMM with mixture representatives equal to the fraction of the observed group data and a balanced spectrum of the co-variances (i.e., $\tau = 1$). Then, the labels come from a linear (or one hidden layer) network with a known number of units and unknown weights.

 Under this setup, they obtain convergence and generalization bounds that provide insights on which are the elements that help or don't help improve the training of unbalanced datasets. In particular, low norm means of each group and covariances close to the medium are key.

 Overall, the insights obtained from the paper are interesting but are very limited in the setting they consider. Unfortunately, as mentioned in the weaknesses, the paper does not successfully try to see if these insights carry on in practice in a convincing way. Also, one can not verify the available experiments with an available code. Moreover, there are some parts of the main text and proofs that seem to have some mistakes that need addressing or explaining.
 Therefore, for these reasons, I must incline towards rejection at the moment.

 On another note, not to the authors fault though, the paper and especially the proofs are very long and having a proper review and proof-check is complicated in the time window given for a conference, and maybe a journal would be a more appropriate venue for this work.


**Some minor comments and nitpicks that did not impact the score of the review**

* Many times you start sentences with a citation using citep, it should be citet.

* In the introduction before the contribution. Please clarify: "In these works, the input features are usually assumed..."

* In the notation, please, clarify "The matrix spectral norm is $\lVert Z \rVert = \delta_1(Z)"$.

* In Section 3. "an unbalanced dataset"

* When you refer to works themselves and not the authors, you should youse citep instead of citet. For instance, right before Section 5.

* The second and third paragraphs in Appendix A.1. are repeating themselves, don't they?

* Before (12), you used one $\times$ instead of $\otimes$.

* The $\Gamma$-function $\Gamma(\Psi)$ should also contain a dependence on $W^*$.

---

> ### Comment · Reviewer_MquM · 2022-11-21
> **Answer to response [1/]**
>
> Thank you for your response. Some of my comments are already addressed. See below for further comments on some parts.
>
> **A to A1:** True, it is not a logistic regression for $K > 1$, I wrote that wrong. My concern here is that this is treated as a neural network when the model in (1) is assuming an average pooling so this basically means that it is an average of $K$ logistic regression models trained simultaneously. Treating that as a "theoretical characterization of neural network generalization with group imbalance" seems an overstatement.
>
> **A to A3:** Thanks for fixing that. However, it is still stated that the results give similar intuition to whitening and making data zero-mean (Le Cun 1998). This is not accurate for the same reason as for batch normalization.
>
> **A to A4:** This does not help me with my question. Yes, you may have a low-rank approximation of your matrix such that $\tau$ is small. But, can you make sure that this approximation is often a good approximation? Can we say that $\tau \in \Theta(1)$ or is it actually $\tau \in \Theta( f(r) )$ where $r$ is the low rank we are using? Does this low-rank approximation $r$ depend on $d$? These are questions that are not addressed and are probably important to understand the applicability of this work. In general, I am still under the impression that the assumption $\tau \in \Theta(1)$ is too strong.
>
> **A to A7:** The answer is not satisfactory. If all $\lVert \Sigma_l \rVert$ are close and then increasing $\lambda_l$ for the smallest $\lVert \Sigma_l \rVert$ increases performance since they are all order close, the same happens for the largest. What is the difference here?
>
> **A to A10**: Sure, it is no longer significant, but it is not an equality, right?
>
> **A to A12:** Thanks. I believe that this procedure should be explained in the paper as well.
>
> **A to A14 and 15:** Including these estimations of $\lVert \Sigma_{\textnormal{male}} \rVert$ and $\lVert \Sigma_{\textnormal{female}} \rVert$ are good for the understanding the claims made. Albeit being close, they still are as they "should" for the claims made to be true so that is good.
>
> **A to A17:** My doubts on the vacuousness of the bound remain. Even with $10^{-2}$ for a simple dataset like MNIST with $d=256$ one would presumably need $75$k samples, which is around $15$k smaller than the samples available and even more than those needed.
>
> **A to A18:** I acknowledge the effort by the authors in their response and their updated version. However, I still can't recommend the paper for acceptance in its current form and believe that a longer reviewing process (like that of a journal) would be most fitting for such a work. I hope the work by the reviewers here were helpful in the improvement of their work.

---

### Official Review · Reviewer_PYh3 · 2022-10-31

**Confidence:** 4
**Correctness:** 3
**Technical Novelty And Significance:** 3
**Empirical Novelty And Significance:** 2
**Recommendation:** 5

**Clarity, Quality, Novelty And Reproducibility:**

The paper is clearly written. There is some novelty in the theoretical model and analysis. The experiments seem to be simple so I think they should be reproducible.

**Strength And Weaknesses:**

Strength:

The paper is clearly written and overall easy to follow. I did not check the details of the proof but the theoretical results seem to be sound.

Weakness:

I think the theoretical model is quite restrictive and unnatural. The input distribution is assumed to follow a GMM, and the labels are generated using a true network, which is a strong assumption. The training algorithm involves a tensor initialization step which is not usually used in practice. These assumptions limit the significance of the theoretical contributions.

The empirical evidence supporting the theoretical results seems to be weak. All the test accuracy numbers in Figure 1(b) and Figure 6 are within 1~2 percentage points. The authors use a very small training dataset so it is likely that these numbers are all within confidence intervals, i.e., the results are not statistically significant. Moreover, the authors did not report error bars in their plots. Therefore, I think the practical implications of the results are also weak.

**Summary Of The Paper:**

This paper proposes a Gaussian mixture model (GMM) for binary classification problems with imbalanced groups. The labels are generated according to a ground truth neural network. Authors mention a few observations according to the theoretical analysis and provided empirical evidence using CelebA and CIFAR datasets.

**Summary Of The Review:**

I think although this paper is well-written, there are some issues with its theoretical model and empirical evidence as mentioned above. Therefore, I don't think its contribution is significant enough to qualify as an ICLR paper.

---

### Comment · Area_Chair_GBMG · 2022-11-22
**Please respond as soon as possible if you still have questions on the paper.**

Please respond as soon as possible if you still have questions on the paper.

---

> ### Comment · Area_Chair_GBMG · 2022-11-29
> **Please respond to the authors by Nov. 30**
>
> Please indicate whether the authors' rebuttal addresses your concerns.
>
> If you still have questions, please ask as soon as possible.

---

> > ### Comment · Area_Chair_GBMG · 2022-12-05
> > **Zoom Meeting**
> >
> > For all reviewers, which have not responded to the authors, I will have to ask you to meet via Zoom. If you want to avoid such an additional step, please respond by Dec. 5.

---

### Decision · Program_Chairs · 2023-01-20

**Decision:**

Reject

**Justification For Why Not Higher Score:**

NA

**Justification For Why Not Lower Score:**

NA

**Metareview: Summary, Strengths And Weaknesses:**

This paper introduces a Gaussian mixture model (GMM) for binary classification tasks involving imbalanced groups. The labels are generated based on a reference neural network. The authors present some observations based on their theoretical analysis and provide empirical evidence using the CelebA and CIFAR datasets.

Though two reviewers gave high ratings, they are not experts on related optics. The other reviewers raised concerns about the restrictive and unnatural assumptions of the theoretical model, which includes a Gaussian mixture model (GMM) for the input distribution and the use of a "true" network for generating labels. The training algorithm involves a tensor initialization step that is not typically used in practice. These assumptions limit the significance of the theoretical contributions. Moreover,  the empirical evidence supporting the theoretical results is weak, as the test accuracy numbers in the plots are not statistically significant and error bars are not included. The practical implications of the results are very limited.

**Summary Of Ac-Reviewer Meeting:**

NA